# An adeno-associated virus variant enabling efficient ocular-directed gene delivery across species

Shuang Luo [1,2,3,6], Hao Jiang [1,3,6], Qingwei Li[1,3], Yingfei Qin[1], Shiping Yang[1], Jing Li[1], Lingli Xu[1], Yan Gou[1], Yafei Zhang[1], Fengjiang Liu [4], Xiao Ke [1,5] ✉, Qiang Zheng [1,3] ✉ & Xun Sun [2] ✉

Recombinant adeno-associated viruses (rAAVs) have emerged as promising gene therapy vectors due to their proven efficacy and safety in clinical applications. In non-human primates (NHPs), rAAVs are administered via suprachoroidal injection at a higher dose. However, high doses of rAAVs tend to increase additional safety risks. Here, we present a novel AAV capsid (AAVv128), which exhibits significantly enhanced transduction efficiency for photoreceptors and retinal pigment epithelial (RPE) cells, along with a broader distribution across the layers of retinal tissues in different animal models (mice, rabbits, and NHPs) following intraocular injection. Notably, the suprachoroidal delivery of AAVv128-anti-VEGF vector completely suppresses the Grade IV lesions in a laser-induced choroidal neovascularization (CNV) NHP model for neovascular age-related macular degeneration (nAMD). Furthermore, cryo-EM analysis at 2.1 Å resolution reveals that the critical residues of AAVv128 exhibit a more robust advantage in AAV binding, the nuclear uptake and endosome escaping. Collectively, our findings highlight the potential of AAVv128 as a next generation ocular gene therapy vector, particularly using the suprachoroidal delivery route.

Adeno-associated viruses (AAVs) are small non-enveloped viruses with a size of approximately 25 nm. They belong to the dependoparvovirus genus of the parvoviridae family and carry a 4.7-kb single-stranded DNA (ssDNA) genome[1]. AAV vectors have gained significant attention as highly effective gene therapy vectors in clinical settings[2]. The unique characteristics of AAV vectors, such as their broad tissue tropisms, low immunogenicity, ease of production, stable, long-term gene expression, non-pathogenic nature, and rarely integrate into the host chromosome[3–5], make them the most promising vector for in vivo gene therapy. Currently, there are six vectorized rAAV serotypes that have

obtained regulatory approval for commercial use in patients. These include AAV1 (Glybera, 2012; uniQure)[6], AAV2 (Luxturna, 2017; Spark Therapeutics)[7], AAV9 (Zolgensma, 2019; Novartis)[8], AAV2 (Upstaza™, 2022; PTC Therapeutics)[9], AAV5 (Roctavian™, 2022; BioMarin Pharmaceutical)[10] and AAV5 (Hemgenix, 2022; uniQure)[11]. As of 25 July 2023, there were 223 interventional clinical trials involving rAAV registered at ClinicalTrials.gov.

Despite the numerous advantages of recombinant adeno-associated viruses (rAAVs), their clinical development is limited by dose-limiting immunotoxicity and poor tissue-specific targeting[5]. To

[1]Chengdu Origen Biotechnology Co. Ltd, Chengdu 610036, China. [2]Key Laboratory of Drug-Targeting and Drug Delivery System of the Education Ministry and Sichuan Province, Sichuan Engineering Laboratory for Plant-Sourced Drug and Sichuan Research Center for Drug Precision Industrial Technology, West China School of Pharmacy, Sichuan University, Chengdu 610041, China. [3]Sichuan Provincial Key Laboratory of Innovative Biomedicine, Chengdu 610036, China. [4]Innovative Center for Pathogen Research, Guangzhou Laboratory, Guangzhou 510005, China. [5]Chengdu Kanghong Pharmaceuticals Group Co Ltd, Chengdu 610036, China. [6]These authors contributed equally: Shuang Luo, Hao Jiang. ✉e-mail: txzz@cnkh.com; zhengqiang@cnkh.com; sunxun@scu.edu.cn

overcome these challenges, scientists have employed various strategies such as capsid engineering and vector genome optimization. One notable example is the use of an AAV2-based engineered AAV2.7m8 capsid, which significantly enhances transgene transduction in retinal photoreceptors and retinal pigment epithelium (RPE) following intravitreal injection compared to wild-type AAV2[12]. In addition to the naturally occurring AAVs, the engineered AAV variants were also investigated in many studies. Multiple engineered AAV variants have gone to therapeutic applications. Among the ongoing clinical trials for retinal diseases, 34% used engineered AAV serotypes[13]. These include AAV2tYF (generated by rational design, clinical trials include NCT02599922, NCT02935517, NCT03316560 and NCT02416622), AAV2.7m8 (generated by directed evolution, clinical trials include NCT04645212, NCT03748784, NCT04418427 and NCT03326336) and 4D-R100 (generated by directed evolution, clinical trials include NCT04483440 and NCT04517149). Encouragingly, more and more engineered AAV variants are explored in clinical applications. To develop novel AAV capsids, researchers have explored four main approaches: natural discovery, directed evolution, rational design, and artificial self-competent AI design[14–20]. Each approach has its own advantages and limitations. In the current study, we focused to modifying and improving the naturally derived AAV8 capsid through rational design for ocular gene delivery.

Neovascular age-related macular degeneration (nAMD) is a leading cause of irreversible vision loss in individuals aged 65 and above, making it an attractive target for ocular gene therapy[21,22]. The pathogenesis of AMD involves choroidal neovascularization (CNV), a process regulated by vascular endothelial growth factor (VEGF)[23]. Current standard treatment for nAMD involves repetitive intravitreal injections of VEGF inhibitors such as anti-VEGF biologics (e.g., Aflibercept or Conbercept), which has significantly improved patient outcomes[24]. However, it is important to note that this approach necessitates frequent injections every 2 to 3 months, which can be inconvenient for patients and may pose certain risks. These risks include subconjunctival and vitreal hemorrhages, corneal edema, conjunctival scars, retinal tears and detachment, lens damage, development of cataracts, choroidal rupture, ocular hypertension, and endophthalmitis[25–28]. In this context, rAAVs present a promising and transformative therapeutic modality for nAMD. By delivering therapeutic genes to target cells, rAAVs have the ability to provide sustained and targeted expression of therapeutic proteins, potentially reducing the need for frequent injections and enhancing treatment convenience and efficacy.

However, high rAAV doses may cause more trouble and health problems. It has been reported that a higher rAAV dose leads to a higher incidence of intraocular inflammation[29]. In the Phase 2 INFINITY trial, one subject receiving a high dose of ADVM-022 encountered a truly unexpected and serious adverse reaction of hypotony with panuveitis and loss of vision following a single intravitreal injection[30]. The subretinal injection is an invasive surgical procedure and requires high technical skills and advanced equipment, and carries the risk of retinal tears, detachments, and macular holes[31]. The suprachoroidal injection is a recent breakthrough in the retinal gene-delivery landscape and may provide a unique opportunity to perform less invasive surgeries[31,32]. In order to effectively reach the target tissues, rAAVs have to cross the Bruch's membrane after injection through the suprachoroidal space (SCS), with a larger titer of $7 \times 10^{12}$ vg/eye[33]. Unfortunately, high doses of rAAVs tend to increase production costs significantly and may increase additional safety risks. To address these critical issues, we aim to engineer the capsid of rAAV and lower the dose of rAAV.

In this study, we present a novel capsid, AAVv128, which was generated through rational design and exhibits significant differences from its parental capsid, AAV8, despite a high sequence similarity of 99%. We demonstrate several key properties that distinguish AAVv128 from AAV8. First, AAVv128 shows improved transduction efficiency specifically targeting photoreceptors and RPE in the retina, regardless of the route of administration (subretinal, intravitreal, or suprachoroidal injection). Second, AAVv128 demonstrates a broader transduction across different layers of retinal tissue following intraocular injections. This wide spread of transduction is observed consistently in various animal models, including mice, rabbits, and non-human primates (NHPs), highlighting the translatability of AAVv128's properties across different species. Most importantly, in the laser-induced CNV NHP model, the administration of AAVv128 via suprachoroidal injection and expressing an anti-VEGF protein results in more effective treatment compared to its parental vector, AAV8. AAVv128 exhibits significantly higher expression of the anti-VEGF protein, indicating its potential as a more potent therapeutic vector for inhibiting the progression of CNV lesions associated with nAMD. Finally, to gain insights into the structural and functional characteristics that define AAVv128, we performed biophysical and cryogenic electron microscopy (cryo-EM) analyses. The 2.1 Å resolution cryo-EM structure of AAVv128 revealed notable differences compared to the structure of AAV8, providing valuable information about the unique features and functional properties of the AAVv128 capsid. This structural feature may create a "sponge effect" and exhibits a more robust advantage in AAV binding, the nuclear uptake and endosome escaping.

## Results

### Rational engineering and screening of AAV8-based capsid variants

Naturally derived AAV8 vector has shown some promises in ocular gene delivery[34]. In this study, we aimed to improve its retina tropism through different routes of administration by employing a rational design approach. Briefly, we used the AAV8 capsid as a scaffold and replaced its amino acid sequences between Q588 ~ A592 or N263 ~ T274 at the VP3 VIII or VP3 VI variable region. To construct these capsid variants, we designed a set of 18 peptides, ranging from 5 to 14 amino acids in length, based on the known structural biology and cellular receptorology of AAV8[35–42] (Supplementary Table 1). Our design approach considered the significance of R585 and R588 as crucial amino acids for the AAV2 cellular receptor heparan sulfate proteoglycan (HSPG)[43]. These amino acids also played a role in interacting with the laminin receptor (LamR)[44]. Furthermore, we acknowledged the importance of positively charged amino acids in enhancing cell binding and facilitating immune evasion post-cell entry[45–47]. These rationally engineered capsid variants were generated by separately packaging a vector genome containing an enhanced green fluorescent protein (eGFP) expression cassette flanked with AAV2 inverted terminal repeats (ITRs) by using HEK293 cell-triple transfection method (Fig. 1a).

In our study, we compared the efficacy of intravitreal injection of different engineered AAV8 variants for intraocular delivery in C57BL/6 J mice. We chose intravitreal injection among the three commonly used intraocular injection methods (intravitreal, suprachoroidal, and subretinal) due to its ease of clinical translatability (Fig. 1b). Four weeks after injection, a careful examination was conducted to assess the expression of eGFP transgene mRNAs in the eyes of mice exposed to different vectors ($6 \times 10^8$ vg/eye, $n = 3$). This assessment was carried out using the quantitative reverse transcription polymerase chain reaction (qRT-PCR). In Fig. 1c, the color depicted represent relative mRNA expression levels from highest to lowest value (red to white, respectively). Among the tested vectors, v121, v123, and v128 exhibited significantly higher mRNA levels compared to AAV8. Notably, AAVv128 demonstrated the highest mRNA expression, with a level that was 75-fold higher than that of AAV8 (Supplementary Table 3). Furthermore, we obtained fluorescent images of the coronal sections of the retina tissues from mice that received the leading vectors, as well as AAV8 control vector (Fig. 1d,e). The images showed that AAVv128, in

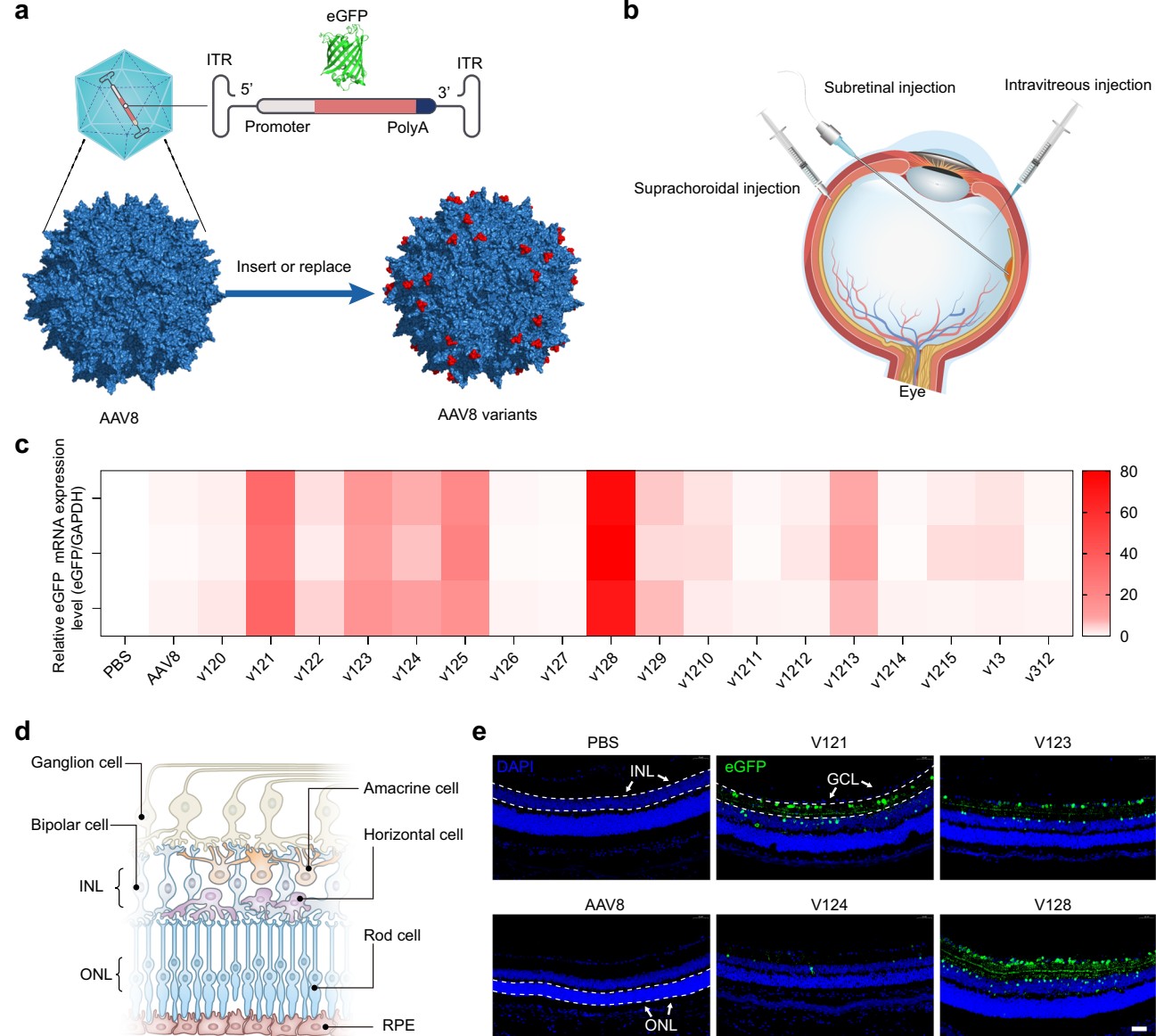

**Fig. 1 | rAAV library screening via intravitreal injection in mice. a** Schematic diagram of rAAV and rAAV variants components. **b** Three injection routes commonly used for AAV ophthalmic gene therapy. **c** Mice were sacrificed at D28 and eyes were harvested. The detection of eGFP mRNA expression levels in AAV8 and AAV8 variants after intravitreal injection in mice using RT-qPCR ($6 \times 10^8$ vg/eye, $n = 3$). The color depicted represent relative mRNA expression levels from highest to lowest value (red to white, respectively). **d** The cell types across the retina were illuminated. **e** Mice were sacrificed at D28 and eyes were harvested. Coronal sections of rAAV-CBA-eGFP transduced mice retina. Immunofluorescence (IF) stained sections (Green) with antibodies against eGFP indicate the positively transduced cell type across the retina. (Scale bar: 50 μm, GCL ganglion cell layer, INL inner nuclear layer, ONL outer nuclear layer;). We conducted three repetitions for each sample, with each assay being operated independently. Source data are provided as a Source Data file.

particular, exhibited clear transduction primarily in the ganglion cell layer (GCL). In contrast, AAV8 showed virtually no transduction in the retina after intravitreal delivery in mice. These findings suggest that the engineered AAV8 variant, AAVv128, achieved enhanced transduction efficiency in the ganglion cell layer following intravitreal injection compared to the native AAV8 vector.

**Subretinal injection of AAVv128 vector produces more robust transduction in different layers and cell types of mouse retinas**
We evaluated the outcomes of subretinal injections in the eyes of mice by collecting optical coherence tomography (OCT) images. Figure 2a illustrated the formation of blebs and the successes of subretinal injections in representative eyes from the groups treated with PBS, AAV8 and AAVv128 vectors, respectively. To assessed eGFP expression, the treated eyes were visualized using fundoscopy at D14 and D28. The

mean pixel intensity per pixel area of AAVv128 was significantly higher compared to AAV8, with a 2-fold increase at Day 14 and a 1.4-fold increase at Day 28 ($1 \times 10^9$ vg/eye, Fig. 2e, f). Similarly, the total pixel intensities were significantly higher in the AAVv128 group, with a 3.7-fold increase at Day 14 and a 3.2-fold increase at Day 28 compared to AAV8. (Fig. 2e, f, Supplementary Fig. 1). It was generally assumed that AAV8 would reach peak expression after 21 days[48–50]. Interestingly, the mean pixel intensity per pixel area of AAVv128 increased significantly from day 0 to day 14, but only slightly between days 14 and 28. This observation suggests that the AAVv128 vector has a faster onset of transgene expression compared to AAV8 (Fig. 2e, f). Additionally, flat mounts of the retina were prepared at Day 28 and evaluated using fluorescence microscopy. The mean pixel intensity per pixel area and total pixel intensity of the AAVv128 group were 1.9-fold and 2-fold higher, respectively, than those of AAV8 (Fig. 2g, i). These findings

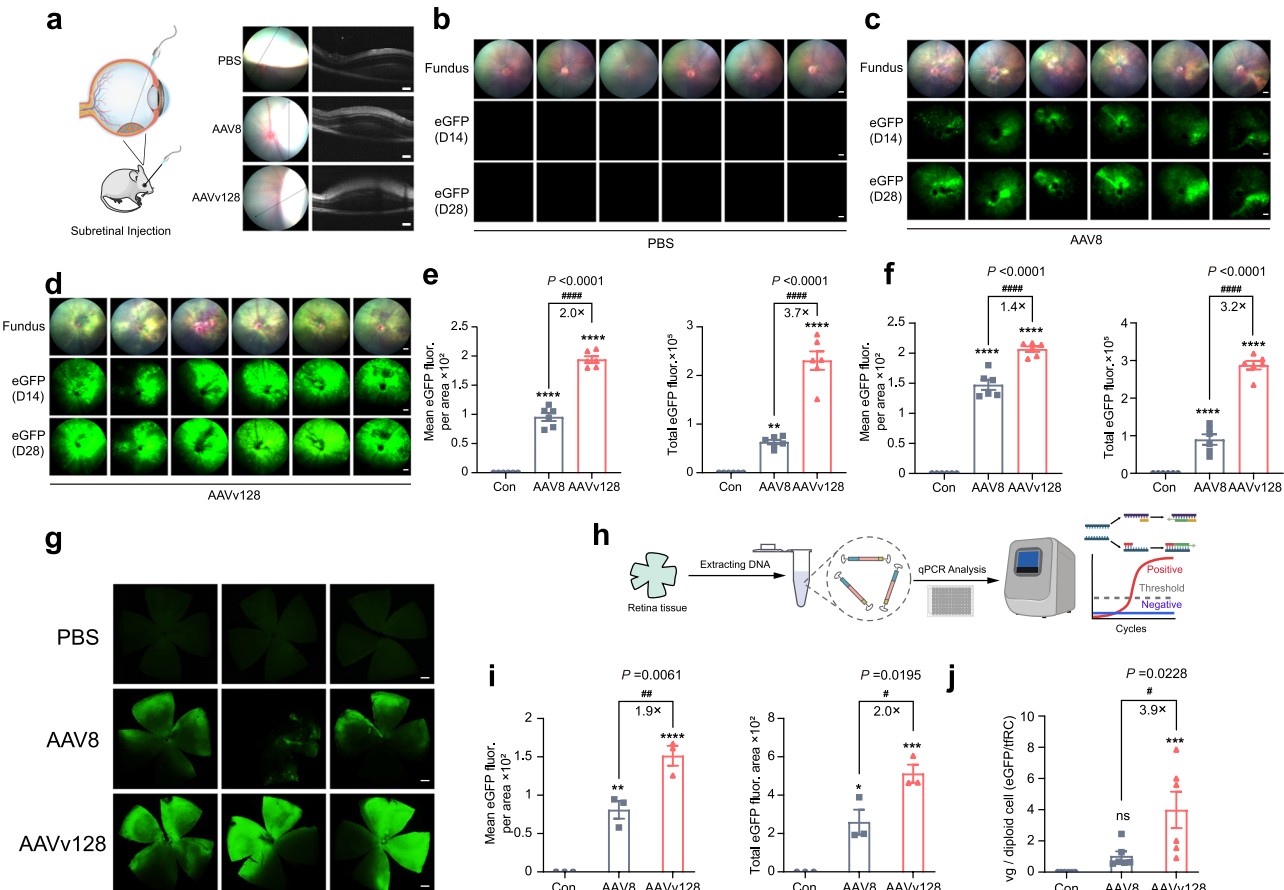

**Fig. 2 | Testing transduction of AAV8 and AAVv128 capsid following subretinal injections. a** OCT images showing retinal blebbing formed at the site of subretinal administration. Representative animal eyes were shown at post-injection at necropsy (1 µL volume, $n = 6$). **b** Fluorescence fundoscopy of mouse eyes treated with PBS (1 µL volume, $n = 6$, scale bar: 500 µm). **c**–**f** Fluorescence fundoscopy of mouse eyes treated with ssAAV-CBA-eGFP vectors packaged with AAV8 (**c**) or AAVv128 (**d**) capsids ($1×10^9$ vg/eye, 1 µL volume, $n = 6$, scale bar: 500 µm). Mice were imaged at D14 and D28 post-injection, and each image of pixel intensity and mean pixel intensity per pixel area was quantified by using Image J (**e**, **f**). **g**, **i** Mice where

sacrificed at D28 and eyes were harvested. Flat mounts of the retina were prepared and imaged by fluorescence microscopy to visualize native eGFP expression (**g**) and quantify by pixel intensity and mean pixel intensity per pixel area (**i**). Scale bar: 500 µm. **h**, **j** Mice retinal tissues were isolated and the relative eGFP copy number per cell were measured by qPCR. Values represent mean ± SD. *P* Values were determined by one-way ANOVA. Compared with PBS, *$P < 0.05$, **$P < 0.01$, ***$P < 0.001$, ****$P < 0.0001$. Compared with AAV8, #$P < 0.05$, ##$P < 0.01$, ###$P < 0.001$, ####$P < 0.0001$. $n = 3$/group (**i**). ns not significant. $n = 6$/group (**e**, **f**, **j**). Source data are provided as a Source Data file.

indicate that the AAVv128 vector leads to significantly higher mean pixel intensity per pixel area and total pixel intensity, reflecting enhanced eGFP expression and distribution in the subretinal space compared to AAV8. The lack of substantial increase in mean pixel intensity per pixel area from Day 14 to Day 28 suggests a faster and more robust transgene expression with AAVv128. The results from the flat mounts further support the superior performance of the AAVv128 vector in terms of eGFP expression compared to AAV8.

In our study, we performed molecular analysis to further validate the superior performance of AAVv128 compared to AAV8. We quantified vector genomes in total cellular DNAs and the relative eGFP copy number per cell extracted from mouse retina tissues. The results showed that the mean eGFP vector genome copies detected in AAVv128-treated retina tissues were 3.7-folds more abundant than those in AAV8-treated tissues. Furthermore, the relative eGFP copy number per cell in AAVv128-treated tissues were 3.9-folds higher than those in AAV8-treated tissues (Fig. 2j). These molecular findings support and reinforce the previous observations of higher mean pixel intensity per pixel area and total pixel intensity in AAVv128-treated retina tissues, indicating a greater abundance of eGFP expression. The quantification of vector genomes and the relative eGFP copy number per cell provides further evidence for the enhanced performance of AAVv128 compared to AAV8.

In addition, we performed co-immunofluorescent staining on coronal sections of retina tissues using antibodies against specific cellular markers. This allowed us to examine the distribution and transduction pattern of eGFP expression in different retinal layers with AAVv128 and AAV8 vectors. The results of the co-immunofluorescent staining revealed that AAVv128 vector exhibited much wider spread and stronger eGFP transduction across various layers of the retina compared to AAV8. Specifically, AAVv128 vector robustly transduced photoreceptors (Rod and Cone), retinal pigment epithelium (RPE) and horizontal cells. However, AAVv128 vector did not show transduction in astrocytes, ganglion cells, anaplastic cells and Müller cells (Fig. 3). In contrast, AAV8 vector showed transduction in photoreceptors, RPE, and horizontal cells, but at lower frequencies compared to AAVv128 vector.

## The thermal stability of AAVv128 is comparable to AAV8 and shows stronger cell transduction in vitro

A good capsid variant possesses immense commercial value and holds promising prospects for application[13,51]. Its significance extends beyond its high transduction efficiency, encompassing attributes such as exceptional stability, ease of purification, and remarkable packaging efficiency. To assess the stability of the AAVv128, we conducted experiments comparing the temperature dependence of AAVv128

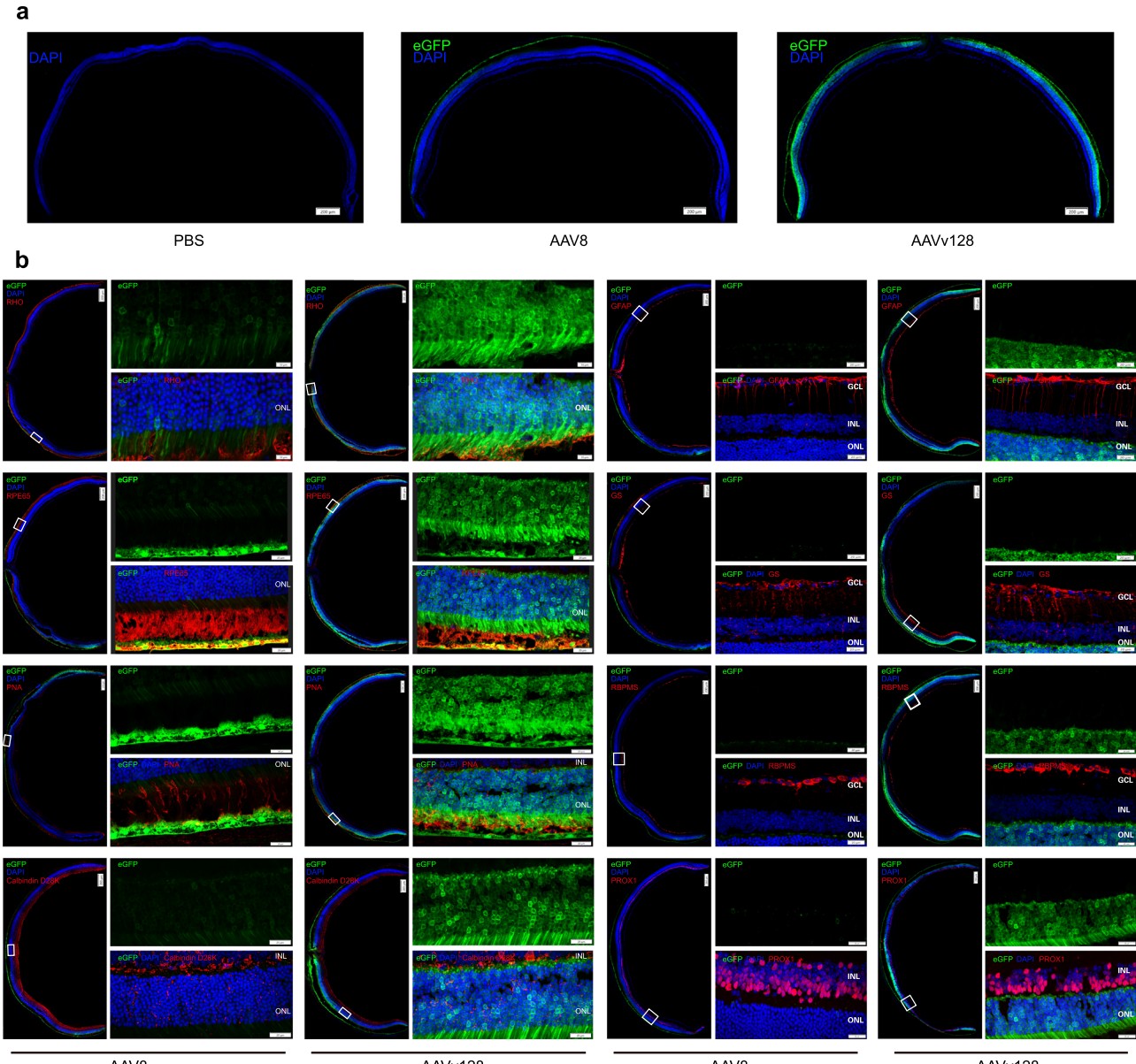

**Fig. 3 | Cross-section and immunofluorescence analyses of mice retina treated with subretinal injection of AAV8 and AAVv128 capsids. a** Coronal sections of mice retina transduced with rAAV-CBA-eGFP. Native eGFP expression (green) shows the positively transduced cell. Scale bar: 200 μm. We conducted four repetitions for each sample, with each assay being operated independently. **b** IF stained sections (red) with antibodies against Rhodopsin (photoreceptors, Rod), Peanut agglutinin (PNA, photoreceptors, Cone), RPE65 (RPE cells), GS (Müller cells), GFAP (astrocytes), RBPMS (ganglion cells), PROX1 (anaplastic cells and ganglion cells) and Calbindin D28K (horizontal cells, anaplastic cells and ganglion cells) indicate the distribution of cell types across the retina. Native eGFP expression (green) that colocalize with IF staining (yellow or white) reveals the positively transduced cell type indicated. Scale bars: 20 μm (right column) or 200 μm (left column). INL inner nuclear layer, ONL outer nuclear layer. We conducted four repetitions for each sample, with each assay being operated independently.

genome release with that of AAV8 at different pH values. We employed a technique called differential scanning fluorimetry (DSF) analysis with SYBR Gold dye by using Uncle, which fluoresces upon binding to DNA[52]. The peak fluorescence observed is an indirect measure of the maximal accessibility of the encapsided genomes to the dye solution[1].

In our study, we found that the release of the capsid genome at pH 7.0 was concurrent with capsid stability. The signal peaks for DNA accessibility were observed at approximately 58.2 °C for AAV8 and 57.4 °C for AAVv128. Compared to AAV8, AAVv128 exhibited more evident DNA accessibility at pH 6.0. The peak DNA accessibility for AAV8 occurred at around 60.5 °C, while AAVv128 showed peak signals at approximately 58.5 °C. At pH 8.0, AAVv128 demonstrated peak signals at approximately 56.8 °C, while the peak DNA accessibility for

AAV8 occurred at around 58.2 °C (Fig. 4a). Those observation indicate that AAVv128 capsid is at least as stable AAV8. The stability of a capsid variant is crucial for its commercial value and application prospects[53]. When moving into human use under the auspices of an FDA Investigational New Drug (IND) application, it is necessary to demonstrate the stability of AAV under various conditions of storage, dilution, and administration when used in humans[53]. The results of our stability analysis suggest that AAVv128 possesses good stability, which is an important characteristic for its potential use as a gene delivery vehicle. Additionally, AAVv128 has demonstrated high transduction efficiency, ease of purification, and high packaging efficiency, further enhancing its potential for commercial applications.

The use of Capillary Isoelectric Focusing (cIEF) allowed us to determine the isoelectric point (pI) of AAV8 and AAVv128. The

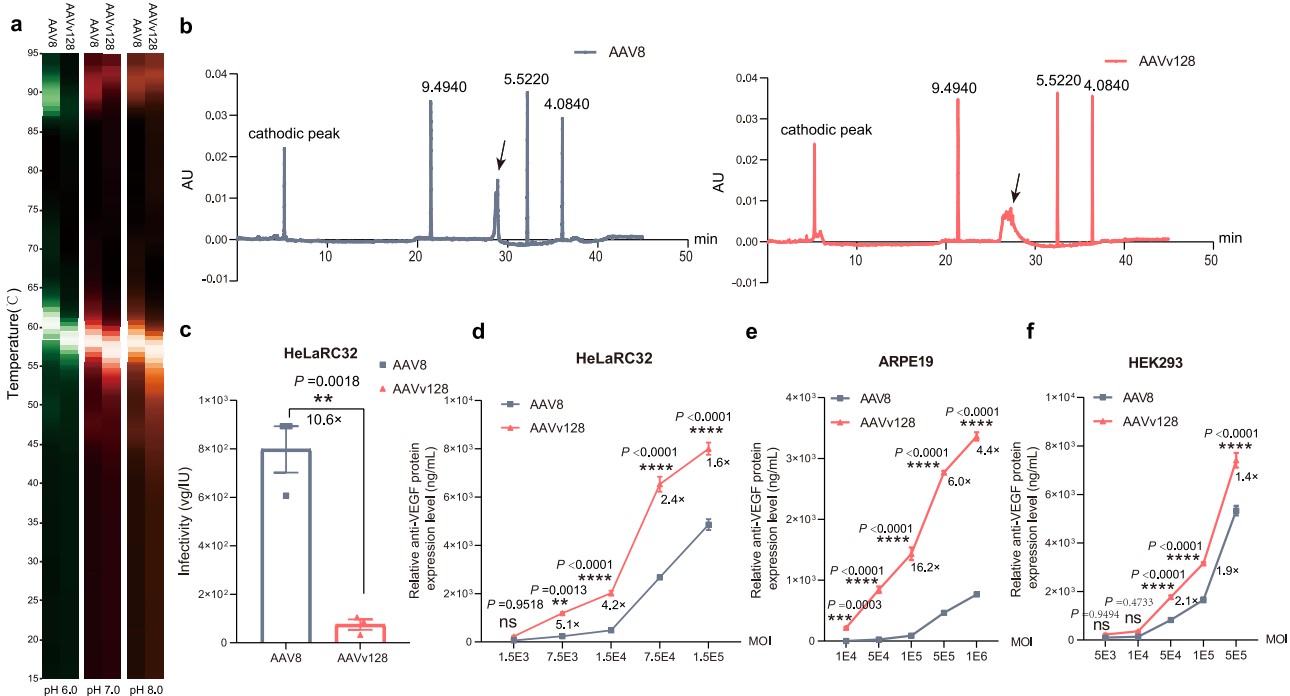

**Fig. 4 | Biochemical and physiological properties analyses of AAVv128 and AAV8. a** Heatmap displays of differential scanning fluorimetry (DSF) analyses used a 473 nm laser to excite SYBR Gold bound to DNA (vector genome extrusion) at pHs 6, 7 and 8. Color scaling depicted represent relative peak signals from highest to lowest value (brightest to dimmest, respectively). **b** The isoelectric point of AAV8 and AAVv128 were detected using capillary isoelectric focusing (cIEF). **c** The infectivity of AAV8 and AAVv128 were detected by a 96-well $TCID_{50}$ format

and quantitative polymerase chain reaction (qPCR). The relative anti-VEGF protein expression levels of AAV8 and AAVv128 on ARPE19 (**e**), HeLaRC32 (**d**) and HEK293 (**f**) cells were detected by ELISA, respectively. Values represent mean ± SD. *P* Values were determined by a two-tailed Student t Test. *$P < 0.05$, **$P < 0.01$, ***$P < 0.001$, ****$P < 0.0001$. ns not significant. $n = 3$/group. Source data are provided as a Source Data file.

measured pI of AAVv128 was 7.6930, which was significantly higher than the pI of AAV8 at 6.8590. This finding further confirms that the insertion of positively charged amino acids in the VP3 VIII region of AAV8 led to a substantial increase in the pI of AAV (Fig. 4b).

The accurate determination of rAAV infectivity is crucial for assessing the activity and ensuring the quality of each rAAV. To determine the infectivity of rAAV, we employed a 96-well $TCID_{50}$ format combined with quantitative polymerase chain reaction (qPCR) detection. The results, as shown in Fig. 4c, revealed that the infectivity of AAVv128 was 75.0 ± 37.5 vg/IU, while that of AAV8 was 797.7 ± 165.1 vg/IU (Fig. 4c). These findings indicate that AAVv128 exhibited a 10-fold greater ability to infect HeLaRC32 cells compared to AAV8, which further supports the superior transduction efficiency of AAVv128 observed in mice (Figs. 2, 3). Interestingly, we observed differential transduction results among ARPE19, HeLaRC32 and HEK293 cells for AAVv128 compared to AAV8. AAVv128 demonstrated more specific transduction in ARPE19, HeLaRC32 and HEK293 cells than AAV8 (HEK293, AAVv128 vs. AAV8 at MOI = 5.0 × 10^4, 1769.17 ± 67.66 ng/mL vs. 824.10 ± 62.21 ng/mL; ARPE19, AAVv128 vs. AAV8 at MOI = 5.0 × 10^4, 843.27 ng/mL vs. 25 ± 9.17 ng/mL; HeLaRC32, AAVv128 vs. AAV8 at MOI = 1.5 × 10^4, 2022.17 ± 17.14 ng/mL vs. 477.93 ± 59.28 ng/mL; Fig. 4d–f). Overall, the combination of enhanced infectivity, higher isoelectric point, and specific transduction patterns makes AAVv128 a highly valuable and promising candidate for gene delivery in ocular therapies.

### AAVv128 exhibits a more robust advantage in AAV binding, the nuclear uptake and endosome escaping

We utilized the positive-AAV8 neutralizing antibody (Nab) serum (Nab blockade, 99.73%) to investigate if the AAVv128 could evade positive-

AAV8 Nab detection (Fig. 5a). The findings demonstrated that the Nab assay for AAVv128 aligned with AAV8 (Fig. 5b).

To compare the cellular transduction efficiencies of AAV8 and AAVv128, we used a virus-cell binding and nuclear uptake assays on cultured HeLaRC32 and ARPE19 cells (Fig. 5c). The results, as shown in Fig. 5, revealed that the AAV binding and cellular uptake of AAVv128 were 5754.0 vg/cell and 123.5 vg/cell in ARPE19, while that of AAV8 were 186.6 vg/cell and 5.0 vg/cell (Fig. 5f, g). These findings indicated that the binding and cellular uptake of AAVv128 were 30.8-fold and 24.7-fold higher than AAV8. What's more, we found that AAVv128 were also 15.6-fold and 64-fold more efficient than AAV8 at the binding and cellular uptake assays in HeLaRC32 cells (Fig. 5d, e). To interrogate nuclear association of AAV variants, HeLaRC32 cells were transduced for 4 h or 6 h with AAV8-eGFP or AAVv128-eGFP and then subjected to nuclear fractionation and quantitative PCR (qPCR) to determine the localization of viral genomes (Fig. 5c). The results showed that the nuclear fraction ratio of AAVv128 was 28.3% and 34.6% at 4 h and 6 h, respectively, which was significantly higher than the nuclear fraction ratio of AAV8 at 7.8% and 15.8% (Fig. 5h, i). Additionally, we found that the relative anti-VEGF protein expression levels of AAVv128 were significantly higher than AAV8 at different time points when transduced with HeLaRC32 and ARPE19 cells. This further supports the higher nuclear fraction ratio of AAVv128 compared to AAV8 in HeLaRC32 cells (Fig. 5h–k).

### Cryo-EM reveals VP3 VIII variable region of AAVv128 form stronger interaction with the surrounding amino acids and may act as a "zipper" for the "pocket" loop

To characterize the structural properties of AAVv128, the purified AAVv128-eGFP vector was subjected to cryo-EM analysis. A total of

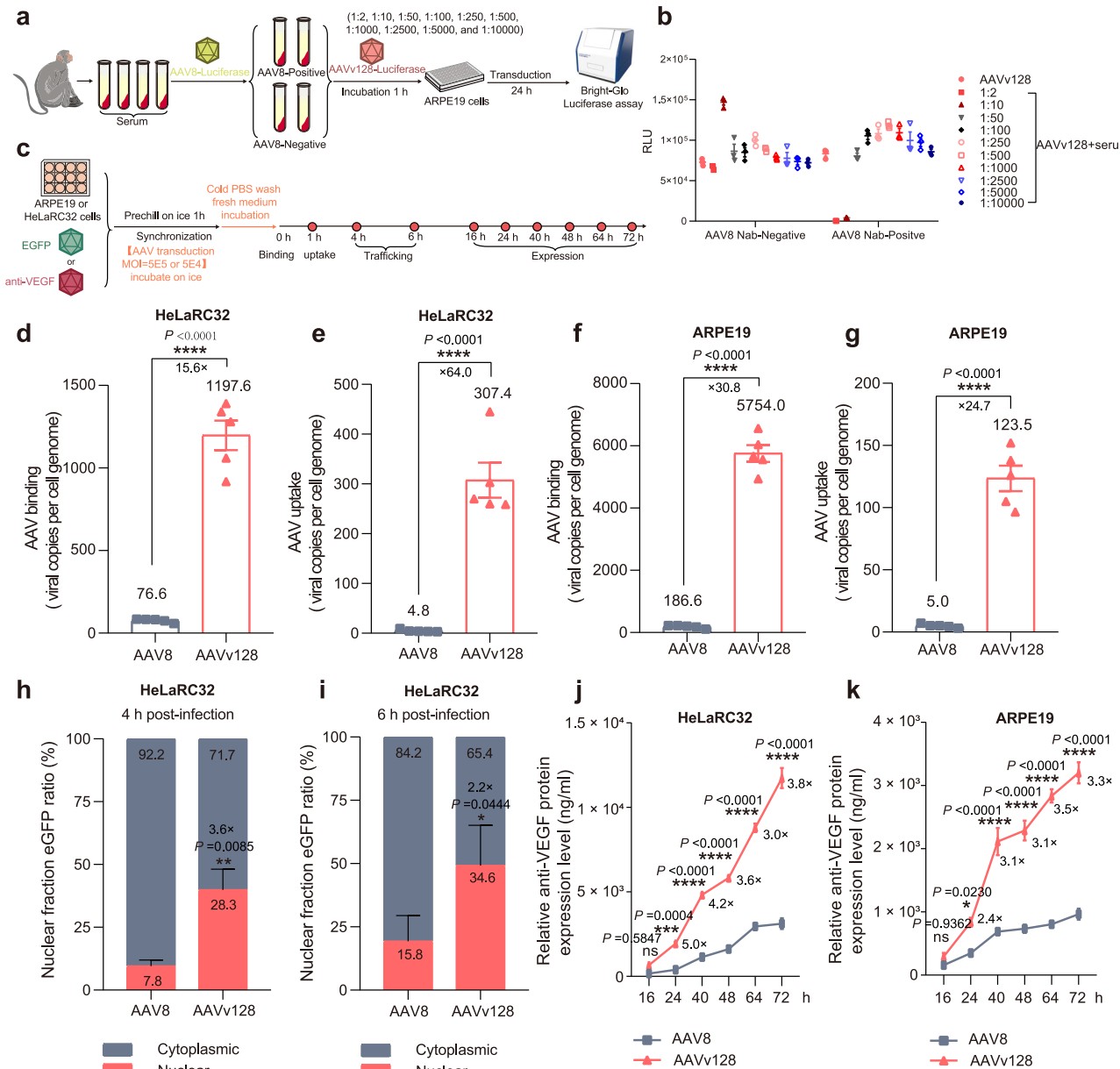

**Fig. 5 | Assessing the performance of the AAVv128 and AAV8 during trafficking steps. a** Schematic diagram for studying the effect of AAV8 Nab-positive monkey serum on AAVv128. **b** Neutralizing antibody (Nab) assay for the determination of the AAVv128 could escape positive-AAV8 Nab recognition. **c** Depicting the process of the AAVv128 and AAV8 during trafficking steps.

**d, e, f, g, h, i, j, k** The AAV binding (**d, f**), cellular uptake (**e, g**), nuclear uptake (**h, i**) and expression (**j, k**) of AAV8 and AAVv128 were detected by qPCR and ELISA, respectively. Values represent mean ± SD. $P$ Values are determined by a two-tailed Student t Test. *$P < 0.05$, **$P < 0.01$, ***$P < 0.001$, ****$P < 0.0001$. $n = 5$/group (**d, e, f, g, h, i**), $n = 3$/group (**j, k**). Source data are provided as a Source Data file.

90,744 particle images were obtained, which allowed the generation of a cryo-EM map at 2.1 Å resolution (Fig. 6, Supplementary Table 7, Supplementary Fig. 3). Using this map, a structural model was constructed, showing optimal real-space fit and stereochemical parameters (Supplementary Table 7). The overall structure of AAVv128 was found to be highly similar to AAV8, both in terms of its overall structure and amino acid sequence (Fig. 6b, c, Supplementary Fig. 4).

The AAVv128 (8JRE) exhibited characteristic features observed in AAV8 capsid, including a depression at the 2-fold symmetry axis, protrusions surrounding the 3-fold symmetry, a channel at the 5-fold symmetry axis that was comprised of five monomers that form the interface and pore for Rep binding, and a 2/5-fold wall separating the depression surrounding the 5-fold axis and at the 2-fold axis (Fig. 6a). While the reconstructed density maps looked similar on the outer capsid surface for the empty/full structures (The results weren't

shown). Of note, VP1u and VP2 domains were not resolved in our cryo-EM map, which were consistent with previous observations in other AAV structures[35–42]. Consequently, only residues 218–743 were definitively resolved within the cryo-EM map (Fig. 6c, Supplementary Fig. 5).

A comparison of the AAVv128 structure with that of AAV8 revealed a distinctive structural feature that may contribute to differences in capsid thermal stability and transduction ability. The AAVv128 and AAV8 structures were similar, except for the 3-fold protrusions (Root-mean-square deviation (RMSD) of 0.507 Å after superimposition of 516 Cα atoms; Fig. 6b, c). The 3-fold protrusions were more pronounced in AAVv128 due to the 5-amino acid insertion, which protruded radially outward from the capsid surface, forming a larger loop that had greater spatial flexibility (Fig. 6b, c). The 3-fold region, including the protrusions, contained receptor binding, infectivity and

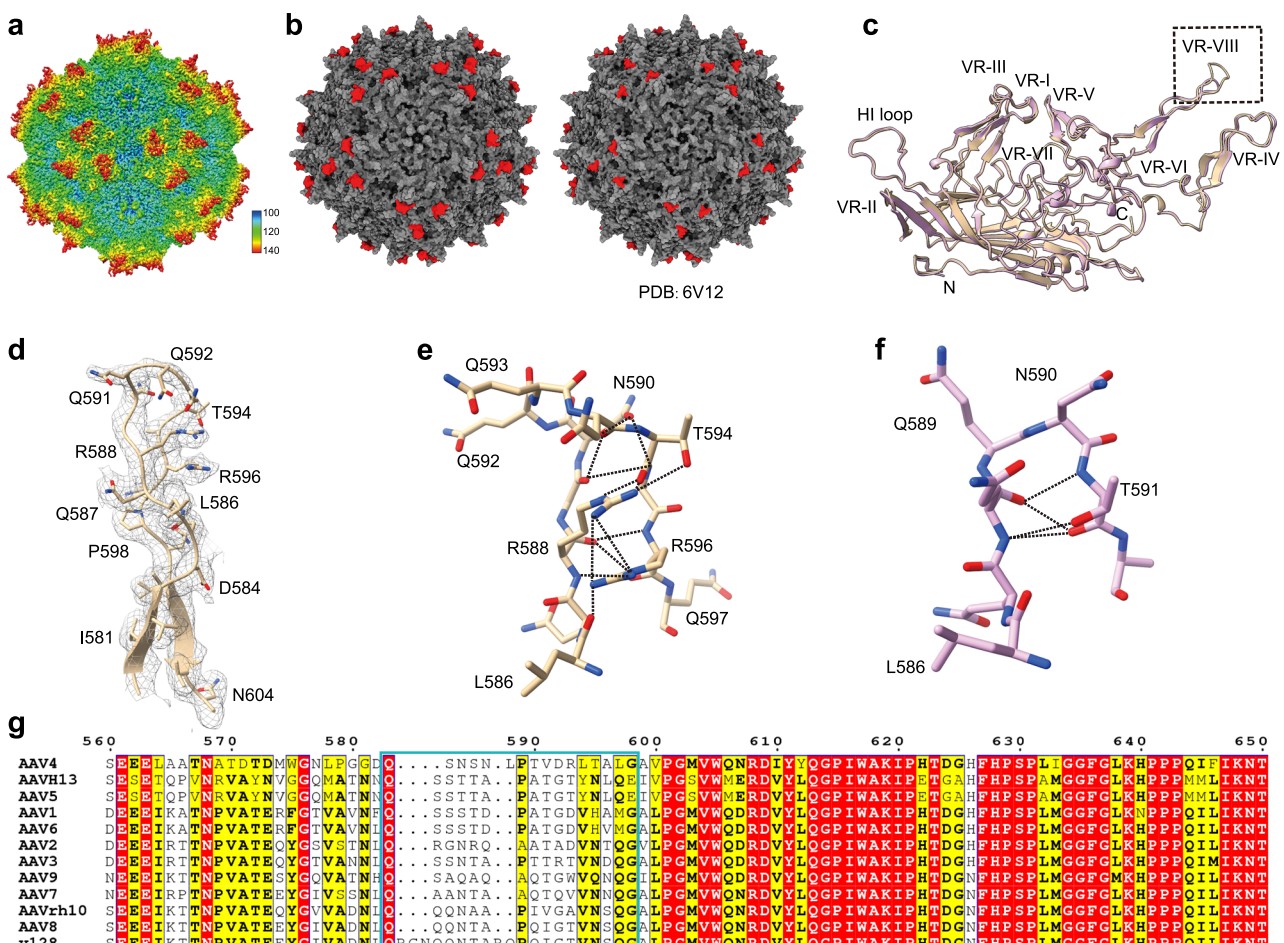

**Fig. 6 | Cryo-EM reveals differences between AAVv128 and AAV8. a** The rendered images of the empty AAVv128 was shown and labeled with the particle dimensions. Depth cueing with colors was used to indicate radius (<100 Å, blue; 100–120 Å, from cyan to green; > 140 Å, red). **b** Comparison of the capsid surfaces of AAV8 and AAVv128. Gray surface representation of the AAV8 (PDB, 6V12) and AAVv128 (PDB, 8JRE) generated from 60 VP monomers. The amino acid differences between with AAV8 and AAVv128 on the capsid surface were colored red. **c** The Structural superposition of AAV8 and AAVv128 VP shown as ribbon diagrams. The conserved β-barrel core motif and the αA helix were indicated, the positions of VR-I to VR-IX were labeled. **d** The residues in a featured fragment from AAVv128 VP3 VIII variable region (D584-N604) were shown as colored sticks and cartoon surrounded by density. **e, f** The residues of main difference between AAV8 (**f**) and AAVv128 (**e**) lied in the VP3 VIII variable region were shown as colored sticks. The residues that the insertion and the 'pocket loop' were labeled in a resplendent shade of red. **g** The identical or conserved residues were shown as letters with red or yellow backgrounds, while the non-conserved residues have a white background. The variable regions between different AAV serotypes were shown in cyan.

antigenicity controlling residues[54]. In AAV8, only a few hydrogen bonds were formed between amino acid sequences in this loop, such as the interaction between the hydroxyl group of T591 and the backbone carbonyl of the symmetry-related Q588 (3.3 Å, Fig. 6f). In contrast, AAVv128 formed a large number of hydrogen bonds between amino acid sequences in this loop, including interactions between R588 to R596, Q593 to A595, among others (Fig. 6e, Supplementary Table 8).

Furthermore, the amino acid sequence of AAVv128 loop was more positively charged, resulting in a 0.83 higher isoelectric point for the entire AAVv128, with a pI of 7.69 (compared to a pI of 6.86 for AAV8, Fig. 4b). Within the amino acid sequence of AAVv128, R588 formed a strong interaction with R596, which appeared to stabilize this loop, acting as a zipper for the "pocket" loop (Fig. 6e). L586 in AAVv128 interacted with R596, whereas L586 in AAV8 did not form such an interaction with the surrounding amino acids (Fig. 6e, f). These observations suggested that the transduction efficiency of AAVv128 at the animal level was substantially better than that of AAV8, potentially due to the charged nature and stability of this loop. The VP3 VIII variable region of AAV2 has been shown to assist AAV2 in entering cells by recognizing the heparan sulfate proteoglycan (HSPG) site on the

cell surface receptor[43,55,56]. However, the VP3 VIII variable region of AAV8 is not involved in HSPG recognition by the cell surface receptor[57]. The data on copy number in mice suggest that AAVv128 is more favorable than AAV8 for recognition by the cell surface receptor and entry into cells (Fig. 2j, Supplementary Fig. 11). This suggests that the more positively charged loop in AAVv128 may play a role in facilitating the recognition of AAV by cell surface receptors, thereby enhancing cellular uptake of the virus.

## AAVv128 outperforms AAV8 in intravitreal and suprachoroidal gene deliveries to rabbits and suprachoroidal gene delivery to NHPs

In an attempt to assess translatability of the strong retina tropism of AAVv128 from murine to large animals, we used the mCherry transgene to perform a comprehensive comparison of three capsids, namely AAV8, AAV2.7m8, and AAVv128, through intravitreal injection and suprachoroidal injection in New Zealand rabbits (1×10$^{11}$ vg/eye). The results showed that AAVv128 exhibited the highest transduction efficiency when administered via suprachoroidal injection, while AAV2.7m8 outperformed other AAV variants when delivered through intravitreal injection (Supplementary Fig. 6). It is worth noting that the

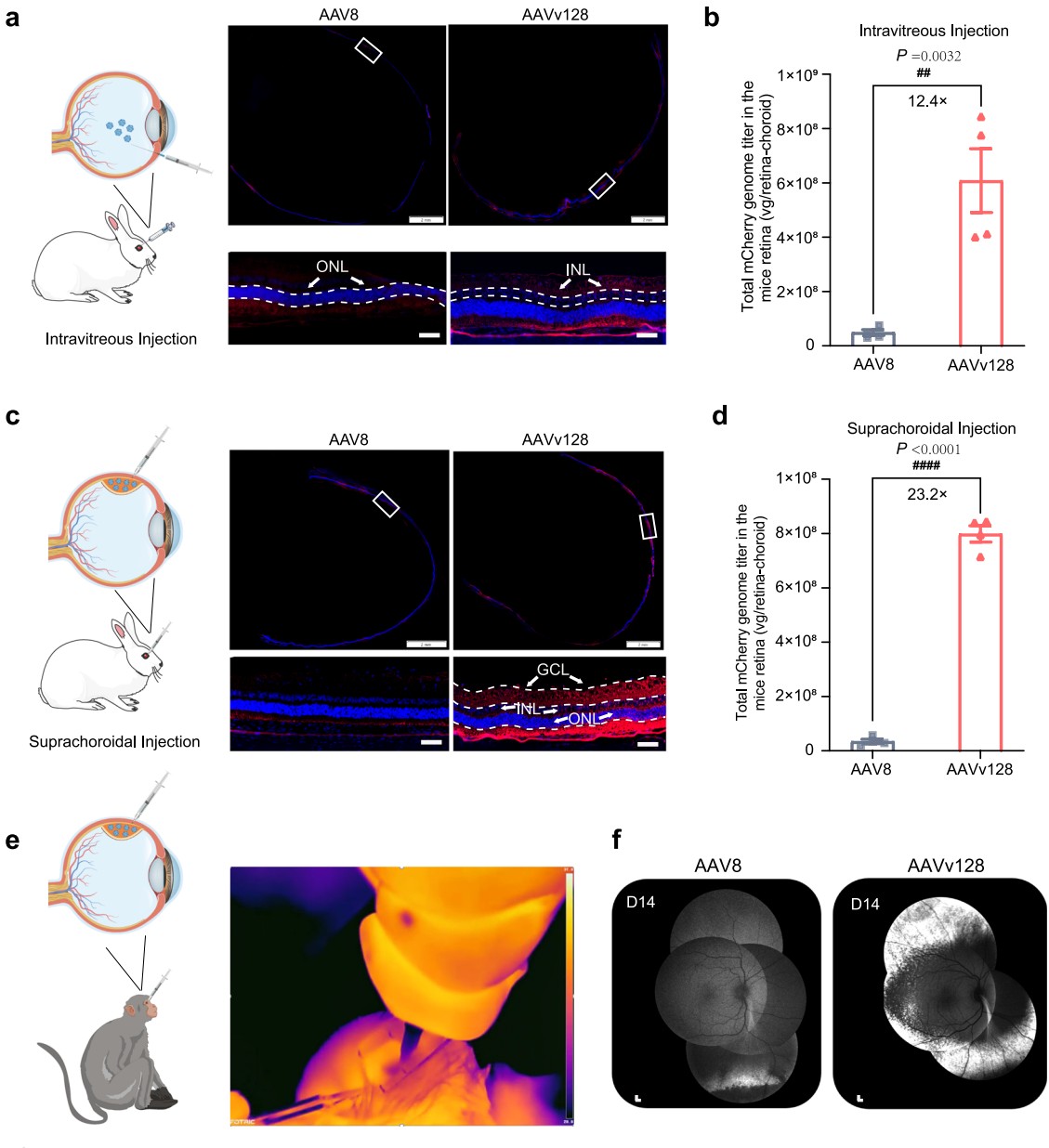

**Fig. 7 | Intraocular injections in large animals to evaluate the transduction efficacy of AAV8 and AAVv128.** The New Zealand rabbit eyes treated by intravitreal injections or suprachoroidal injections of the AAV8 and AAVv128 confers detectable mCherry expression at days 28 post-injection ($1 \times 10^{11}$ vg/eye, 100 µL volume, $n = 4$). **a**, **c** Immunofluorescence analysis of the New Zealand rabbit retinas after intravitreal injection (**a**) and suprachoroidal injection (**c**). **b**, **d** The retina-choroid tissues of the New Zealand rabbit retinas after intravitreal injection (**b**) and suprachoroidal injection (**d**) were isolated and their viral genomic DNA content were measured by qPCR ($n = 4$). Scale bars: 50 µm (down column) or 2 mm (up column). **e** Infrared imaging analysis of suprachoroidal injections in cynomolgus monkeys. NHPs eyes treated by suprachoroidal injection of the AAV8 and AAVv128 capsid confers detectable eGFP expression at days 14 post-injection ($3.5 \times 10^{12}$ vg/eye, 100 µL volume, $n = 1$). **f** Transduction validation of NHPs eyes treated with AAV8 and AAVv128 by Scanning laser ophthalmoscopy (SLO); scale bars: 200 µm. GCL ganglion cell layer, INL inner nuclear layer, ONL outer nuclear layer. Values represent mean ± SD. $P$ values were determined by a two-tailed Student $t$ Test. #$P < 0.05$, ##$P < 0.01$, ###$P < 0.001$, ####$P < 0.0001$. Source data are provided as a Source Data file.

AAVv128 capsid was developed based on the AAV8 capsid, using it as a scaffold and introducing specific amino acid substitutions in the variable region of VP3 VIII (Q588 ~ A592). In subsequent experiments, we focused on the use of AAVv128 compared to AAV8.

For intravitreal injections, the superior transduction of AAVv128 was supported by quantitative data, as the vector genome abundance detected by qPCR in total cellular DNAs from the whole retina-choroid tissues was 12.4-fold higher for AAVv128 compared to AAV8 (Fig. 7b). In the case of suprachoroidal injections, the superiority of AAVv128 over AAV8 in retinal transduction was even more apparent.

The red fluorescence signal was much stronger in the retina sections of rabbits injected with AAVv128 compared to AAV8 (Fig. 7c). Consistently, the qPCR analysis revealed that AAVv128 had 23.2-fold more vector genome copies in the whole retina-choroid tissues compared to AAV8 (Fig. 7d). Noticeably, AAVv128 predominantly transduced the photoreceptor inner and outer segments (IS/OS) and GCL after both intravitreal and suprachoroidal injections, whereas AAV8 primarily transduced the IS/OS (Fig. 7a, c). This indicates that AAVv128 exhibits a broader range of transduction across different layers of the retina compared to AAV8. These findings suggest that

AAVv128 maintains its strong retina tropism in large animals, as observed in murine models.

We also assessed the retina transduction of AAV8 and AAVv128-eGFP vectors in NHPs through suprachoroidal injection ($3.5\times10^{12}$ vg/eye). The injection process was monitored using an infrared camera, allowing us to observe the successful infusion of the vectors into the suprachoroidal space. The injection sites were confirmed by the infusion of low-temperature testing articles, which changed color spectra from blue to red as the vectors gradually reached body temperature (Fig. 7e, Supplementary Fig. 6, see video in Supplementary Movie.1). Animals were scheduled for in vivo fluorescence retinal imaging by scanning laser ophthalmoscopy (SLO) to measure eGFP expression at Day14 after AAV8 and AAVv128 injection, followed by immunofluorescently analysis. Fluorescence imaging revealed a significant increase in eGFP fluorescence in the peripheral fundus when using the AAVv128 vector compared to AAV8 in cynomolgus monkey eyes. This effect was observed when the vectors were delivered via suprachoroidal injections at a dose of $3.5 \times 10^{12}$ vg/eye (Fig. 7f, Supplementary Fig. 9). Collectively, these findings indicated that AAVv128 outperformed AAV8 in delivering the transgene to the fundus more efficiently through the suprachoroidal space in both rabbits and NHPs. The stronger and more widespread fluorescence signal observed in AAVv128-treated eyes suggests its potential as a highly effective vector for ocular gene delivery in translational and clinical applications.

In the co-immunofluorescently analyses of coronal sections of the NHPs retinas, the results further supported the superior transduction efficiency of AAVv128 compared to AAV8 (Supplementary Fig. 8). The images revealed that the transduction of AAVv128 was much more efficient, showing stronger eGFP fluorescence compared to AAV8. In the case of AAVv128, the transduction was predominantly observed in photoreceptors and RPE. On the other hand, AAVv128 showed limited transduction in astrocytes, horizontal cells, bipolar cells, Müller cells and ganglion cells (Supplementary Fig. 8). In contrast, AAV8 primarily targeted photoreceptors and RPE, but with lower efficiencies compared to AAVv128. These findings highlight the differential transduction patterns of AAVv128 and AAV8 in NHP retinas. This information is crucial for understanding the cellular tropism and potential applications of these vectors in ocular gene delivery.

### Suprachoroidal injection of AAVv128 vector can prevent nAMD by Laser-induced CNV and produces more robust transduction and inhibition of VEGF in NHP retinal tissues

Before conducting the animal efficacy studies, we thoroughly validated the quality of the AAV8-anti-VEGF vector and AAVv128-anti-VEGF vector (Supplementary Fig. 2). The results showed comparable purity and empty and full ratios for both vectors. In summary, based on our comprehensive evaluation of the rAAV samples, we have confidence in the satisfactory quality and reliability of our experimental materials. The therapeutic potential of AAVv128-anti-VEGF vector efficacy was performed using a Laser-induced choroidal neovascularization (CNV) NHPs model. Briefly, AAVv128-anti-VEGF vector was injected into suprachoroidal space at Day 0, and six laser spots were applied around the macula of each eye using laser at 28-days ($1\times10^{12}$ vg/eye). At the evaluation timepoint of 56-days post injection (28-days post Laser-induced CNV model), the NHPs were sacrificed and eyes were harvested. The retinal tissues of the NHPs retinas after suprachoroidal injection were isolated and the anti-VEGF protein expression level were measured. A higher concentration of the anti-VEGF protein was found in the AAVv128-treated NHP retinal tissue and other tissues, with levels of 2112 ng/g (OS, oculus sinister) and 4056 ng/g (OD, oculus dextrus) in retinal, respectively, compared to the prototype AAV8. In the control AAV8 group, transgene expression in NHP retinal tissue was 39.12 ng/g (OS) and 52.8 ng/g (OD), respectively (Fig. 8a, Supplementary Table 4). Other tissues including choroidal, sclera, aqueous humor, conjunctiva and vitreous body etc.

In order to further evaluate the treatment efficacy for nAMD, fluorescein fundus angiograph (FFA) was used to determine the number of Grade IV lesions (the gold standard for evaluating the efficacy of treatment for nAMD) at 35-days and 49-days (Fig. 8b, $2 \times 10^{12}$ vg/eye). The anti-VEGF protein level of AAV8 and AAVv128 in the aqueous humor were measured at 35-days. A higher concentration of the anti-VEGF protein was found in the AAVv128-treated NHP aqueous humor, compared to the prototype AAV8 ($n = 8$, Supplementary Table 5). In the Vehicle group, the percentage of Grade IV lesions was 43.75% and 70.83% at 35-days and 49-days (14-days and 28-days post Laser-induced CNV model), respectively, indicating the formation of new vessels. In the AAVv128-treated group, the percentage of Grade IV lesions was 0% and 0% at 35-days and 49-days (Fig. 8c−e, Supplementary Table 6, Supplementary Fig. 10). However, in the AAV8-treated group, the percentage of Grade IV lesions was 31.25% and 41.67% at 35-days and 49-days, respectively. These initial findings indicate that the AAVv128, compared to its prototypical capsid AAV8, demonstrated much higher efficacy for treating nAMD, particularly using the route of suprachoroidal injection. These results suggest that the engineered AAVv128 capsid holds promise as a potential gene delivery vehicle for treating ocular disorders of the retina.

## Discussion

AAV has shown great potential as a gene therapy vector due to its broad host tropism, long-expression, and low-immunogenicity. However, there have been concerns regarding the specificity, potency, and off-target effects of some first-generation AAV products[58]. To address these challenges, researchers have developed new AAV variants through capsid modification to improve transduction efficiency and specificity in on-target cells or tissues. Rational design approaches, based on knowledge about potential receptor interactions, have been employed to create capsid variants with desired features and to reduce the size of the AAV capsid variants library[59]. In addition to the naturally occurring AAVs, multiple engineered AAV variants have gone to clinical trials for retinal diseases, including AAV2tYF, AAV2.7m8 and 4D-R100[13].

AAV8 has been shown to effectively infect photoreceptors and RPE when administrated via subretinal injection[34]. In fact, the transduction efficiency of AAV8 in photoreceptors and RPE was found to be superior to that of AAV2[60]. In this study, we designed and constructed the AAVv128 variant using a rational design approach. Despite there are only 10 amino acid differences form AAV8, AAVv128 exhibited higher transduction efficiencies, a larger transduction area, faster expression rates, and the enhanced ability to enter cells when targeting retinal tissues. These improvements highlight the potential of the AAVv128 variant in gene therapy applications, particularly in the context of retinal diseases.

In the field of ophthalmology, different routes of administration, such as subconjunctival, intravitreal, suprachoroidal and subretinal injections, are commonly used to treat various diseases[61]. Each route may require the use of specific AAV serotypes or variants for effective gene delivery. For example, AAV2 or AAV2 variants are often used to for intravitreal injection[62–64], while AAV8 is commonly used for subretinal injection[65,66]. Subconjunctival injection often utilize AAV8 and AAV6 for gene delivery[67], and AAV8 is considered a reliable choice for suprachoroidal injections[33,68,69]. Compared with subretinal and intravitreal injections, suprachoroidal injections is a recent breakthrough in the retinal gene-delivery landscape and may provide a unique opportunity to perform less invasive surgeries and less likely to cause side effects such as inflammation[33]. However, rAAVs have to cross the Bruch's membrane after injection through the suprachoroidal space (SCS) in order to effectively reach the target tissues, with a larger titer of $7\times10^{12}$ vg/eye[33]. Unfortunately, high doses of rAAVs tend to increase production costs significantly and may introduce additional safety risks.

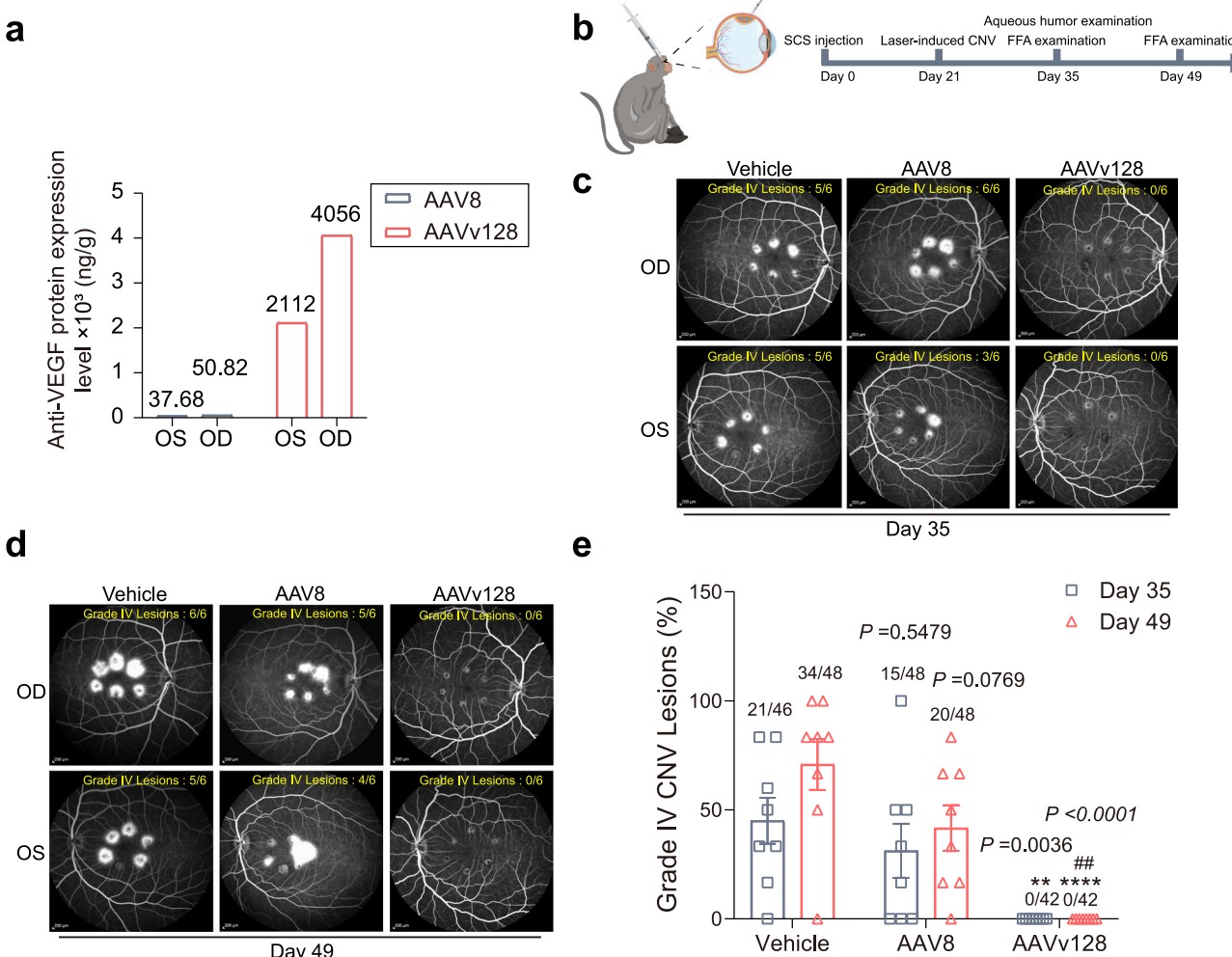

**Fig. 8 | Laser-induced CNV model in NHPs are used to evaluate the efficacy of nAMD by AAVv128-anti-VEGF vector. a** The anti-VEGF protein levels of monkey treated with suprachoroidal injection of AAV8-anti-VEGF vector and AAVv128-anti-VEGF vector (1×10[12] vg/eye, 100 μL volume, *n* = 1) were measured at Day 56. AAV8-anti-VEGF vector and AAVv128-anti-VEGF vector were injected into suprachoroidal space at Day0, six laser spots were applied around the macula of each eye using laser at 28-days. The NHPs were sacrificed at Day 56 (after induction of choroidal neovascularization by Laser-induced CNV model) and eyes were harvested. The retinal tissues of the NHP retinas after suprachoroidal injection were isolated and the anti-VEGF protein expression level of AAV8-treated group and AAVv128-treated group were measured by ELISA. **b** Timeline of studying AAV-anti-VEGF vector to evaluate the efficacy of nAMD by using Laser-induced CNV model in NHP eyes.

**c, d** Fluorescein fundus angiograph (FFA) was used to determine the number of Grade IV lesions. Representative FFA of NHP eyes treated with suprachoroidal injection of Vehicle, AAV8-anti-VEGF vector and AAVv128-anti VEGF vector at 35-days (**c**) and 49-days (**d**) (2×10[12] vg/eye, 100 μL volume, *n* = 8). **e** Percentage of Grade IV lesions with suprachoroidal injection of Vehicle (formulation buffer), AAV8-anti-VEGF vector and AAVv128-anti-VEGF vector. Numbers on the top of bars show the number of Grade IV lesions scored over the total number of assessable lesions (six laser spots/eye, *n* = 8). OD oculus dextrus, OS oculus sinister, FFA fundus fluorescence angiography. Values represent mean ± SD. *P* values were determined by one-way ANOVA. Compared with vehicle, *$P$ < 0.05, **$P$ < 0.01, ***$P$ < 0.001, ****$P$ < 0.0001. Compared with AAV8, #$P$ < 0.05, ##$P$ < 0.01 (*P* = 0.0074). *n* = 8/group. Source data are provided as a Source Data file.

In our study, we found that AAVv128 exhibited stronger transduction efficiency than AAV8 when delivered via subretinal injection in mice. AAVv128 showed a 4-fold increase in its ability to enter cells and expanded transduction to a wider range of cell types, including photoreceptors, RPE and horizontal cells (Figs. 2 and 3). Additionally, we investigated the transduction of AAVv128 and AAV8 in New Zealand rabbit eyes. We observed significantly higher levels of target gene expression in retinal-choroidal tissues with AAVv128 compared to AAV8 when administered via intravitreal or suprachoroidal injections (Fig. 7a, c). Encouragingly, similar results were observed in monkey eyes (Supplementary Table 4, Supplementary Table 5). In a study, it was observed that the administration of a lower dose of AAV8 (7×10[11] vg/eye, rhesus 01) did not result in any detectable expression at any time points. However, when a higher dose of 7×10[12] vg/eye (rhesus 02) was injected, scattered areas of punctate eGFP fluorescence were observed bilaterally in the peripheral fundus[33]. In contrast, the use of the AAVv128 vector led

to a significant increase in eGFP fluorescence in the peripheral fundus compared to AAV8 (dose: 3.5×10[12] vg/eye). Additionally, it was reported that suprachoroidal delivery of AAV in monkey might have a greater tendency to distribute circumferentially along the periphery rather than posteriorly toward the macula[70,71]. Suprachoroidal AAV delivery might be better suited for gene therapies that target peripheral or RPE diseases, for example the production of anti-angiogenesis agents for neovascular retinal conditions, rather than macular diseases[33]. The transduction cell types of AAVv128 in these eyes were mainly photoreceptors and RPE (Supplementary Fig. 8). It was shown that suprachoroidal delivery of AAV8 in Sprague-Dawley rats mainly resulted a larger average transduction diameter and an abundance of transduced cells within the outer nuclear layer (ONL)[72]. This could be due to the structural differences between species that result in different transduction cell types, including Bruch's membrane (BrM) thickness, the thickness or number of cells within the retinal layers[73,74].

Traditional treatment for nAMD often involves frequent intravitreal injections every 2-3 months[75]. However, AAV gene therapy has the potential to provide long-lasting effectiveness with a single injection. In our study, using NHPs with a laser-induced CNV model, the AAVv128-anti-VEGF vector demonstrated promising efficacy. All Grade IV lesions were effectively suppressed using AAVv128-anti-VEGF vector via suprachoroidal injection, while the percentage of Grade IV lesions was 31.25% and 41.67% at 35-days and 49-days in the AAV8-anti-VEGF vector, respectively (Fig. 8c–e, Supplementary Table 6). Most importantly, those results were the first to report complete inhibition of Grade IV lesions in the laser-induced CNV NHP model using AAVv128 delivered by suprachoroidal injection. Moreover, the AAVv128 group exhibited significantly higher levels of anti-VEGF protein expression in retinal tissue, surpassing the AAV8 group by several folds (Fig. 8a, Supplementary Table 4, Supplementary Table 5). These findings suggest that the AAVv128 variant may hold promise for ophthalmic treatments, particularly using the route of suprachoroidal injection.

To better understand the high transduction capacity of the AAVv128 variant, we resolved its structure using cryo-electron microscopy (cryo-EM) with a resolution of 2.1 Å. In comparison to AAV8, we observed the presence of positively-charged amino acids and hydrogen bonds in the AAVv128 variant's variable region of VP3 VIII. This structural feature likely facilitates the formation of a pocket structure in the VP3 VIII loop, which may enhance the interaction with cell surface receptors (Fig.6c-f). This finding could explain the higher levels of viral genome detected in target tissues 28 days after administration of the AAVv128 variant (Figs. 2j and 7b, d). Despite the formation of a larger loop in the variable region of VP3 VIII, the AAVv128 variant exhibited comparable stability to AAV8, as determined by the Uncle assay. These results suggest that the AAVv128 variant possesses favorable characteristics for ophthalmic gene therapy, including enhanced transduction efficiency and stable capsid structure (Figs. 4 and 5). The detection of the isoelectric point (pI) of the AAVv128 variant using cIEF indicated that it contained a higher number of positively charged amino acids (Fig. 4b). This characteristic may create a "sponge effect" and enhance endosome escaping[47]. Consequently, more viral particles may be able to reach the nucleus, as suggested by the results shown in Figs. 2j and 5d–i. It exhibited a greater propensity for cellular recognition, particularly in ARPE19 cells. This increased recognition was observed at a minimum infecting dose (MOI) of 50,000 viral particles per cell. Notably, AAVv128 achieved the same protein expression level as AAV8 while requiring only 1/39th of the titer of AAV8 (Fig. 4e). The infectivity of AAVv128 was assessed using $TCID_{50}$ format and qPCR, which confirmed that AAVv128 had a 10-fold higher ability to infect cells compared to AAV8 (Fig. 4c).

In order to understand the molecular mechanism of AAVv128, the Nab assay for AAVv128 was consistent with AAV8 by using the positive-AAV8 Nab serum. This result suggested that the epitope of Nab of AAVv128 was consistent with AAV8 (Fig. 5b). We also used a virus-cell binding and nuclear uptake assays to explore the cellular transduction efficiencies of AAV8 and AAVv128. The results showed that the AAV binding, cellular uptake, nuclear uptake and expression of AAVv128 were significantly higher than AAV8 (Fig. 5d–k). This is a strong indication that AAVv128 has tremendous advantages over AAV8 in the treatment of ophthalmic diseases.

In conclusion, the rational design of the AAVv128 variant has resulted in significant improvements in its biophysical properties and in vivo tropism compared to AAV8. The enhanced transduction efficiency, stability, cellular recognition, and tissue specificity make AAVv128 a promising gene therapy vector, particularly in the field of ophthalmology. These findings demonstrate the potential for engineering novel and improved viral vectors for gene therapy based on the rational design approach. The development of such vectors holds great promise for advancing the field of gene therapy and expanding the range of diseases that can be effectively treated.

## Methods

### Viral vector production and purification

To generate recombinant AAV8 and AAV8 variants, a triple-plasmid transfection approach was employed[76,77]. Briefly, the transfection was conducted in HEK293T cells (ATCC, #CRL-3216) using the following plasmids: pAAV-Transgene, pAAV-luciferase, pAAV-mCherry, pAAV-anti-VEGF (The coding region of the anti-VEGF protein includes domain 2 of human VEGFR1, domain 3 of human VEGFR2, and the Fc portion of human IgG1.) or pAAV-eGFP; pRC encoding AAV2 Rep and AAV8 Cap proteins or variants RC plasmids carrying various VP variants; and pHelper. The AAV8 variants were generated through rational design strategies, incorporating knowledge about AAV structure and immunology to enhance their properties and functionalities[36,40,78,79]. All transgenes were driven by the CMV early enhancer/chicken β actin (CBA) ubiquitous promoter[80].

The HEK293T cells used for transfection were cultured in Dulbecco's modified essential medium (DMEM; Gibco) supplemented with 10% fetal bovine serum (Gibco) and 1% penicillin–streptomycin antibiotics (Gibco). The cells were maintained at 37 °C and 5% $CO_2$ incubator until they reached approximately 80% confluence. For transfection, the cells were seeded in 150-mm plates and transfected with a mixture of plasmids, including the pHelper plasmid, pRC plasmid, and pAAV-Transgene plasmid, in a molar ratio of 1:1:1. The total amount of plasmid DNA used was 60 µg. At 72 h of transfection, the cells were harvested by centrifugation at 1,500 × g at 4 °C for 15 minutes. The cell pellet was collected and resuspended in a buffer containing 150 mM NaCl, 20 mM EDTA, and 20 mM Tris-HCl. The resuspended lysate was subjected to three freeze-thaw cycles to disrupt the cells and release the viral particles. To remove unpackaged rAAV DNA, the crude lysate was treated with Benzonase and 10% deoxycholic acid at 37 °C for 1 hour. The lysate was then centrifuged at 10,000×g for 30 minutes to separate the packaged rAAV particles from the cellular debris. Simultaneously, rAAV particles released into the culture medium were recovered by adding a PEG/NaCl solution (10% polyethylene glycol 8,000(w/v), 0.5 M NaCl) and precipitation at 4°C for 16 hours. Purification of rAAV was performed using affinity chromatography (Thermo, USA, POROS™ CaptureSelect™ AAVX Affinity Resin, #A36745) and subsequent anion exchange chromatography. The purified rAAV particles were then dialyzed in stock buffer (20 mM Tris-Cl pH8.0, 150 mM NaCl, 1 mM $MgCl_2$, 0.003% Poloxamer 188) using Slide-A-Lyzer™ Dialysis Cassettes (Thermo, 20 K MWCO, 12 mL, #66012).

### ddPCR for viral titer

The rAAV samples were treated with DNase I (200U, Thermo, #18047019) at 37 °C for 1 hour. DNase I degrades any residual unpackaged DNA in the sample. After the digestion, the DNase I activity was terminated by adding 20 µL of 0.1 M EDTA (ethylenediaminetetraacetic acid), and heated in a water bath at 85 °C for 20 minutes. The high temperature inactivates the DNase I enzyme. Proteinase K (TakaRa, #9034) was added to the reaction mixture and incubated at 55 °C for 2 hours. To stop the Proteinase K activity, the reaction mixture was heated in a boiling water bath for 15 minutes. The treated rAAV samples were subjected to PCR analysis using PerfecTa® Multiplex PCR Tough Mix® (Quanta bio, #95147-250). The PCR reaction was set up in a 25 µL reaction system with the following protocol: Partition at 40 °C "Sapphire V1", 95 °C for 10 minutes, followed by 38 cycles of denaturation at 95 °C for 20 seconds, annealing at 63 °C for 20 seconds, and extension at 72 °C for 20 seconds. Release P "Sapphire V1". After completing the PCR reaction, the chip containing the PCR products was removed and placed in a microdrop analyzer. The Crystal Reader software was then opened to collect and process the PCR data. The eGFP primers are as follows: Forward, 5′-CACATGAAGCAGCACGACTT-3′, Reverse, 5′-TCGTCCTTGAAGAAGATGGT-3′, Probe, 5′-FAM-AGTCCGCCATGCCCGAAGGCT-TAMRA-3′; anti-VEGF protein

primers are as follows: Forward, 5′-AAAGGCTTCTATCCCAGCGA-3′, Reverse, 5′-CGGAGCATGAGAAGACGTTC-3′, Probe, 5′-FAM-CAAGG CCACGCCTCCCGTGC-BHQ1-3′; mCherry primers are as follows: Forward, 5′-CACTACGACGCTGAGGTCAA-3′, Reverse, 5′-TAGTCCTCGTTG TGGGAGGT-3′, Probe: FAM-5′-AAGAAGCCCGTGCAGCTGCC-BHQ1-3′.

## AAV capsid mutant construction

The Mutation Generation System kit (Thermo, USA, #F701) was used to generate mutations in the AAV8 capsid ORF. Supplementary Table 1 listed all AAV8 variants. Primer pairs for mutagenesis displayed in Supplementary Table 2.

## Animal use

Six- to eight-week-old male C57BL/6 J mice, twelve- to thirteen-week-old male New Zealand White rabbits with weight ranging from 2.0 to 2.5 kg (Dashuo Laboratory Animal Technology Co., Ltd., China), and two-year-old Cynomolgus monkeys with weight ranging from 2.5 to 5.0 kg (monkeys were purchased from Hubei Topgene Biotechnology Co., Ltd., and housed in West China-Frontier PharmaTech Co., Ltd. (WCFP), Tianfu Drug Research Center, SYXK(Chuan)2021-238) were used in this study. The mice and rabbits were housed under specific pathogen-free (SPF) conditions (70 ~ 74 F with humidity at 35 ~ 45%) at West China school of Pharmacy, Sichuan University, with a 12-hour light/12-hour darkness cycle. Mice and rabbits were fed normal chow (#1010088, #1010070). At the end of the study, the animals were deeply anesthetized and euthanized by carbon dioxide exposure. The animal experiments were approved by the Committee on the Ethics of Animal Experiments of Sichuan University and were conducted in compliance with the recommendations in the Guide for the Care and Use of Laboratory Animals of Sichuan University Ethics Committee.

The cynomolgus monkeys were housed in West China-Frontier PharmaTech Co., Ltd., Tianfu Drug Research Center. The institution is accredited by AAALAC International (Association for Assessment and Accreditation of Laboratory Animal Care International). The experiments conducted with cynomolgus monkeys followed the regulations of the Institutional Animal Care and Use Committee (IACUC) and the Guide for the Care and Use of Laboratory Animals (8th Edition)[81]. The animal experiments were designed in accordance with the appropriate guidelines. This experiment has been approved by the Institutional IACUC under the approval number: IACUC-SW-A2021046-P004-01.

Cynomolgus monkeys were housed individually in double-layer monkey cages (L× W × H: 800 mm × 1000 mm × 2080 mm) in a conventional animal room. The environmental conditions in the animal room were maintained as follows: temperature range of 16 ~ 26 °C with a daily fluctuation of ≤4 °C, relative humidity between 40 ~ 70%, 12 hours of artificial light (07:30 to 19:30) and 12 hours of darkness (19:30 to 07:30) daily. Illumination was provided during the dark period as needed for the study. Cynomolgus monkeys were fed a maintenance diet for monkeys *ad libitum*, with nutrient testing reports provided by the supplier for crude protein, crude fat, crude ash, crude fiber, water, calcium, phosphorus, and amino acids (cystine + methionine and lysine), as well as contaminants such as arsenic, lead, cadmium, mercury, benzene hexachloride, dichlorodiphenyltrichloroethane, aflatoxin B1, bacterial colony count, coli group, mycetes, yeast count, and pathogenic bacteria (Salmonella). Fresh fruit was provided twice a week with *ad libitum* access for the animals.

Prior to sacrifice, the Cynomolgus monkeys underwent a fasting period of at least 12 hours while having *ad libitum* access to water. Following the AVMA Guidelines for the Euthanasia of Animals (2020 Edition), the animals were anesthetized with ketamine 10 mg/kg + xylazine 1.25 mg/kg (a mixed solution of ketamine 40 mg/mL + xylazine 5 mg/mL for intramuscular injection at 0.25 mL/kg), with dosage adjustments based on the animals' condition. Tissue samples were collected following euthanasia, performed by i.v. injection of pentobarbital sodium.

## Intraocular rAAV vector injections and clinical assessment

For subretinal injections, mice were anesthetized by intraperitoneal injections of ketamine (0.1 mg/g, Sigma, #K1884) and xylazine (0.02 mg/g, MedChemExpress, #HY-B0443A) mixed with sterile water in the ratio of 0.6:1:8.4, respectively. Mice received local anesthesia in the eyes using 0.5% proparacaine hydrochloride (TCI, #P2156), and placed on the operating table after anesthesia, with the left hand holding the head and the right hand holding the microinjector. The microinjector was inserted into the vitreous at 1 mm from the posterior edge of the cornea. The tip of the 34-gauge beveled needle was initially inserted vertically and then tilted up towards the contralateral retina. The needle was slowly pushed until resistance was encountered. Then, 1 μL of rAAV vector solution with a titer of $1 \times 10^{12}$ vg/mL was injected using the UMP3T-1 Microinjection Syringe Pump and the Nanofil Sub-Microliter Injection System (World Precision Instruments). The injection was performed under an Opmi 1 FR pro surgical microscope (Carl Zeiss)[82]. The Micron IV retinal imaging microscope (Phoenix Research Laboratories) was used to capture OCT scans, as well as color and fluorescence fundal images. And the fluorescence pixel intensity and mean pixel intensity per pixel area were quantified using Image J (National Institutes of Health, NIH)[82].

For intravitreal injections, mice were anaesthetized by intraperitoneal injections of ketamine (0.1 mg/g) and xylazine (0.02 mg/g) mixed with sterile water in the ratio of 0.6:1:8.4, respectively. The needle bevel was turned downward while penetrating the sclera, choroid and retina. The positioning of the needle in the vitreous was confirmed using a stereo microscope (Olympus, #SZX16). Using a microscope and a 33-gauge needle on a microsyringe (Hamilton, #715265), 2 μL of rAAV vector solution (titer: $3 \times 10^{11}$ vg/mL) was injected into the vitreous of the mice under direct visualization. After the injection, 1% Chloramphenicol ointment (Adooq Bioscience, #A10200) was applied topically to the eye. The mice were placed on a heat-pad to keep them warm during the recovery period.

For suprachoroidal injections, rabbits were given an intramuscular injection of a mixture of ketamine hydrochloride (50 mg/kg) and xylazine hydrochloride (10 mg/kg) for general anesthesia. A drop of topical ophthalmic 0.5% proparacaine hydrochloride was applied for local anesthesia. Using a self-researched suprachoroidal chamber syringe, a 100 μL of rAAV sample (AAV8 or AAVv128) with a titer of $1 \times 10^{12}$ vg/mL was injected into the suprachoroidal space (SCS) along 4 mm of the corneal limbus of rabbit eyes. For the suprachoroidal injection, the device utilized for suprachoroidal injection was a customized combination of an adapter, a 1 mL syringe, and a needle. The adapter incorporated a unique internal design that enabled the syringe and needle to lock after insertion, allowing control over the exposed length of the needle. The length of the needle tip was adjusted by turning the fitting at the top of the adapter. The device was manufactured by Chengdu Origen Biotechnology and currently not commercially available. For suprachoroidal injection in monkeys, a needle tip length of 0.7 mm was used. Additionally, we have uploaded a video capturing the process of suprachoroidal injection in a monkey, recorded using an infrared thermal camera (Supplementary Movie.1). After the injection, 1 to 2 drops of oxyfloxacin eye ointment were applied to prevent infection.

For suprachoroidal injections in cynomolgus monkeys, monkeys were anesthetized with 2.5% sodium pentobarbital intravenously at a dose of 25-30 mg/kg. A 100 μL of rAAV sample (AAV8 or AAVv128) (titer: $3.5 \times 10^{13}$ vg/mL) was injected into the suprachoroidal space (SCS) along 4 mm of the corneal limbus of the cynomolgus monkey's eyes. After injection, each group of monkeys received about 1 to 2 drops of oxyfloxacin eye ointment in both eyes to maintain corneal wetting and prevent infection.

## Quantitative RT-PCR (qRT-PCR) and qPCR

The eGFP mRNA expression levels in AAV8 and AAV8 variants after intravitreal injection in mice were assessed by qRT-PCR[83,84].

Specifically, gene expression of the eGFP was analyzed, indicating the different transduction efficiency of AAV8 and AAV8 variants. Briefly, mice where sacrificed at D28 and retinal tissues were harvested. Total RNA was extracted from retinal tissues using a Macherey-Nagel RNA extraction kit (MN, #740984.50) according to the manufacturer's protocol. RNA quality and quantity were assessed using a DeNovix DS-11 FX Series of Spectrophotometers/Fluorometers (DeNovix) and 25 μL of RNA was then reverse transcribed to cDNA by using High Capacity cDNA reverse transcriptase kit (Thermo, #4368814). And TaqMan gene assays was used to quantify the mRNA expression level on a Real-time PCR System (Roche). Using the TaKaRa MiniBEST Viral DNA Extraction Kit (TaKaRa, #9766), the genome copies in the retinas of mice or rabbits treated with AAV8 and AAVv128 were extracted and quantified using qPCR. The eGFP and mCherry primers were described above. Relative gene expression to endogenous control was calculated using the formula $2^{-\Delta Ct}$, where $\Delta Ct$ represented the magnitude of the difference between cycle threshold (Ct) values of the target and endogenous control (GAPDH, tfRC), and the result expressed as a percentage of the mean value of the control group.

### Optical Coherence Tomography (OCT) and Fundus Fluorescence angiography (FFA) Examination

In the experimental procedure, mice were first anesthetized with 2.5% pentobarbital sodium (approximately 25 mg/kg), administered via intraperitoneal injection. To facilitate examination, their pupils were dilated using Mydrin-p (Compound Tropicamide Eye Drops, eye instillation, 1-2 drops/eye). OCT imaging was performed using the Micron IV retinal imaging system (phoenix micron IV). A rapid linear scan centered on the injection site was conducted to obtain images confirming the completion of dose administration. The OCT and fundus photographic camera system were operated simultaneously during the examination to capture comprehensive retinal information. After the procedure, to prevent infection, 1-2 drops of ofloxacin eye ointment were administered to each animal.

Non-human primates (NHPs) underwent SLO using the Spectralis HRA OCT (Heidelberg) at Day14 after rAAV injections. Confocal SLO was used to capture fluorescence images using 488-nm excitation light. Images were captured from the central macula and from the peripheral retina in regions of visible eGFP fluorescence by manually steering the Spectralis HRA OCT. For FFA examination, NHPs were anesthetized and their pupils were dilated using Mydrin-p. Imgae acquisition was performed by Spectralis HRA (Heidelberg), and monkeys were quickly injected with fluorescein sodium (20 mg/kg) via saphenous veins of lower limb (or other suitable sites). Both early-phase (within approximately 1 min) and late-phase (after approximately 5 min) fundus angiograms were acquired. Pentobarbital sodium (i.v., approximately 10-80 mg/monkey) was given to animals to maintain anesthesia during the study process. After the operation, 1-2 drops of Ofloxacin Eye Ointment were given to animals for anti-infection.

### Laser-induced CNV model in NHPs

NHPs were anesthetized with 2.5 % pentobarbital (25 mg/kg) intravenously. To maintain corneal moisture during anesthesia, a small amount of oxyfloxacin eye ointment was occasionally applied. After the NHPs were adequately anesthetized and the pupils were dilated using tropicamide drops, the head of the monkey was positioned in front of an ophthalmic laser photocoagulator. The laser (Vitra 532 nm) was mounted on a slit lamp using an adapter and the beam was directed onto the retina using a contact lens (Mainster wildfield laser lens). Laser photocoagulations were performed in the perimacular area of the monkey eyes. Lesions were placed in the macula with six spots. Laser lesions were placed in a circular fashion around the macula about one disk diameter from the foveal center. Care was taken to avoid lasering the fovea. Photocoagulation used laser energy to create controlled burns in specific areas of the retina. The laser parameters used for photocoagulation typically included a spot diameter of 50 μm, an energy range of 0.5 to 0.7 W, and an exposure time of 0.05 seconds. The criteria for successful photocoagulation are the presence of visible bubbles, indicating that Bruch's membrane (a layer of the retina) has been disrupted.

### The purity of rAAV capsids was measured by SDS-PAGE

60 μL of rAAV samples at a concentration of $1.0 \times 10^{12}$ vg/mL were mixed with 20 μL of 4× loading buffer containing 10% 2-mercaptoethanol. These samples were then placed in a 90 °C water bath for 10 minutes and then centrifuged at $15,000 \times g$ for 5 minutes. For SDS-PAGE analysis, 10 μL of samples were loaded onto a 10% gel and electrophoresed at 80 V for 30 minutes. The voltage was then increased to 120 V. Electrophoresis was stopped when the bromophenol blue dye migrated to the bottom of the gel. The gel was then placed in a destaining dish. Then, 50 - 60 mL of SYPRO Ruby staining solution (Invitrogen™, # S12000) was added to the dish and the gel was stained for 5 hours. After staining, 50 - 100 mL of an aqueous solution containing 10% ethanol and 7% formic acid was added to the dish for decolorization. The gel was destained for 30 - 60 minutes and then imaged with a gel imaging system.

### Sedimentation velocity analytical Ultracentrifugation (SV-AUC) and TEM assays

SV-AUC experiments were carried out at Tsinghua University using Beckman Coulter cells equipped with 12-mm, two-sector centerpieces and Sapphire windows. Before analysis, the sample and reference sectors were filled with 390 μL of the sample and 410 μL of buffer (pH 6.0, same as sample buffer), respectively. The cell was then fixed into an eight-cell rotor and mounted in the instrument, allowing it to equilibrate for 1 hour after reaching vacuum and the set temperature of 20°C. The absorbance signal was recorded at 230 nm for 100 scans within the radius range of 6.15 - 7.25 cm, and the sample was centrifuged at a speed of 35,500 ×g. The wavelength of 230 nm was chosen for better sensitivity and to minimize response bias between empty and full AAV capsids. SEDFIT software was used to process the SV-AUC data.

For transmission electron microscope (TEM) assays, 20 μL of AAV samples with a concentration of $1.0 \times 10^{13}$ vg/mL were applied to a carbon support membrane. After a 3-minute incubation, excess liquid at the membrane's edge was absorbed using filter paper, leaving a small amount of sample on the membrane. Next, 50 μL of a 3% phosphotungstic acid (Thermo, # 040116) negative staining solution was applied to the membrane and allowed to stand for another 3 minutes. Any excess liquid was absorbed using filter paper, and if needed, the membrane was placed under an irradiation lamp to aid drying. Finally, the treated membrane was fixed using specialized equipment and examined using a Tecnai G2 F20 transmission electron microscope (TEM).

### Heparan sulfate proteoglycan (HSPG) binding assays

200 μL of rAAV samples at a concentration of $1 \times 10^{10}$ vg/mL were added to 0.2 g of HSPG medium (Cytiva, #17-0407-01). After vortexing the mixture every ten minutes, it was allowed to stand at room temperature for thirty minutes. After 30 minutes of incubation with the above complexes, the PVDF membrane (Millipore, #IEVH85R) was rinsed twice with 0.5% PBST for five minutes each. The membrane was then incubated with a 5% skim milk solution for 60 minutes at room temperature and then incubated overnight at 4 °C with HSPG2-specific polyclonal antibody (Proteintech, #19675-1-AP). The membrane was then incubated with HRP-conjugated goat anti-rabbit antibody (Beyotime, #A0208, 1:5000) for 60 minutes at room temperature. The PVDF membrane was then recoated with 0.5% PBST three times for 5 minutes each. Finally, a Bio-Rad gel imager was used to visualize the membrane.

## Cell transduction and Enzyme-linked immunosorbent assays (ELISAs)

Depending on genomic titer, the rAAV sample were aspirated into sterile low adsorption Ep tubes (eppendorf), and then $3.6\times10^5$ ARPE19 cells (ATCC, #ATCC®CRL2302TM), HeLaRC32 cells (ATCC, #CRL-2972™) or HEK293 cells, 10% FBS, supplemented with DMEM, MEMα or DMEM /F12 complete medium (Gibco, # 11320-033) to 600 μL were mixed with rAAV gently. In a 12-well plate, 500 μL of a mixture of rAAV and cells was added to each well (i.e. $3\times10^5$ total cells, $6\times10^{10}$ vg total virus, $1.2\times10^{11}$ vg/mL virus concentration, MOI = total virus/number of cells = $2\times10^5$ in each well) and three replicates of each sample were prepared. The cell cultures were placed in culture for $72\pm1$ hour. At the incubation, the cell plates were frozen and thawed at $-65\,°C$ for 3 times, and ELISA assays were performed to measure the levels of anti-VEGF protein.

Briefly, high-binding 96-well ELISA plates (Costar, #42592) were coated overnight at 4 °C with purified recombinant human VEGF (R&D, #293-VE-050) diluted in DPBS. Then, VEGF165 was removed and wells were washed 3 times with 1×PBS followed by blocking for 2 hours at 37 °C with 5% BSA in PBS. Blocking solution was removed and ARPE19, HeLaRC32 or HEK293 cell supernatants and standards containing anti-VEGF protein were added to the wells and incubated for 1 hour at 37 °C. Following incubation, wells were washed 5 times with 1×PBS. Bound anti-VEGF protein was detected using an HRP-conjugated anti-human VEGFR1, anti-VEGFR1 antibody (R&D, #MAB321) and incubated at 37 °C for 1.5 hours. The plates were washed 5 times with 1×PBS and 100 μL TMB peroxidase substrate solution (R&D, #DY999) was added. The reaction was stopped by the addition of 50 μL of 2 mol/L $H_2SO_4$ and then measured optical density (OD) at 450 nm (TECAN, Switzerland). The blood VEGF expression level of NHP was detected by commercial V-PLEX Plus NHP VEGF Kit (MSD, #K156RHG-1).

## Immunofluorescence analysis

Mice, rabbit or non-human primate where sacrificed at the designated time, and their eyes were collected for further analysis. The harvested eyes were fixed in FAS Eye fixative (Servicebio, #G1109) overnight at 4 °C. The fixed eyes were embedded in a 1:1 mixture of optimal cutting temperature (OCT) compound (Tissue-Tek, #4583) and 30% sucrose. The embedded eyes were cryo-sectioned at a thickness of 5μm using a Leica EM UC7 cryostat. The cryo-sectioned retina tissue sections were blocked with a blocking solution consisting of 5% goat serum in PBS with 0.1% Triton X-100 for 1 hour at room temperature. The tissue sections were incubated overnight at 4 °C in blocking solution containing primary antibodies (anti-eGFP, 1:800, Invitrogen, #A11122; anti-Rhodopsin, 1:200, Invitrogen, #PA5-85608; anti-Arcchis hypogaea, 1:100, Invitrogen, #L32460; anti-Calbindin D28K, 1:100, Invitrogen, #PA1-931; anti-GFAP, 1:200, Invitrogen, #MA5-12023; anti-CHX10, 1:100, Invitrogen, #PA5-85404; anti-RPE65, 1:200, Invitrogen, #PA5-110315; anti-RBPMS, 1:200, Invitrogen, #PA5-31231; anti-PROX1, 1:200, Invitrogen, #PA5-85552; anti-GS, 1:200, Invitrogen, #MA5-27749). The stained retina tissue sections were washed three times in 1× PBS to remove any unbound primary antibodies and reduce background signal. Following the washing steps, the tissue sections were incubated with a secondary antibody (goat anti-rabbit, 1:2000, Invitrogen, #A32740) for 2 hours at room temperature. After incubation with the secondary antibody, the tissue sections were washed three times to remove any unbound secondary antibody. The sections were then stained with 4′,6-diamidino-2-phenylindole (DAPI, Thermo, # D3571), a fluorescent dye that binds to DNA and highlights cell nuclei. The stained slides were examined using digital scanning and viewing software, such as the Olympus SpinSR10 spinning disc confocal super-resolution microscope (JAPAN, OlyVIA). Global retinal images (10× tiled retinal sections) and high-magnification images (60× region specific areas) were collected at the same intensity and exposure thresholds for each respective magnification. For high-magnification images, the parameter was as follows: OBIS Laser 405, 50%; OBIS Laser 488, 30%; OBIS Laser 561, 30%. And the fluorescence pixel intensity and mean pixel intensity per pixel area were quantified using Image J (National Institutes of Health, NIH).

## The thermal stability of rAAV vector was determined by differential scanning fluorimetry (DSF) analysis using Uncle

Thermal denaturation of rAAV involves genome ejection and capsid disruption[85]. Uncle (Unchained Labs) can track genome ejection with DNA binding fluorescent dyes to determine a melting temperature ($T_m$) based on DNA release[52]. For capsid stability experiments, SYBR™ Gold 10,000× in DMSO (Thermo, #S11494) was diluted to a 400× working stock in PBS (pH 7.4, Corning). rAAV samples (titer: $5\times10^{12}$ vg/mL) were adjusted to different pH levels ranging from pH 7 to pH 4. The samples were then centrifuged for 60 seconds at $14,000\times g$ in a benchtop centrifuge to remove any large particles. 9 μL of each rAAV sample, along with 20× SYBR™ Gold, a set of standards, and a well of buffer alone, were tested using the Capsid Stability & DLS (Dynamic Light Scattering) application on Uncle. SYBR™ Gold was excited with a 473 nm laser. DLS readings involved acquiring four acquisitions of 5 seconds each. The samples were heated from 25 °C to 95 °C at a rate of 0.5 °C/minute while continuously monitoring the fluorescence. At 95 °C, a final DLS reading was taken. The samples were cooled back to 25 °C, and a final fluorescence measurement was taken. The area under the fluorescence intensity curves from 500–650 nm was calculated using Uncle Analysis software. The Tm values was defined as the maximum Δsignal/Δtemp detected between 25 and 95 °C. Tm represents the temperature at which significant genome ejection and capsid disruption occur.

## The infectious titer of rAAV was assessed by a 96-well $TCID_{50}$ format and qPCR

HeLaRC32 cells ($2\times10^4$ cells/well) were seeded into 96-well plates. After an incubation period of 18-24 hours, the cells were infected with human adenovirus 5 (Ad5, ATCC, #VR-1516™) at the saturating concentration of $8\times10^6$ vg per well[86]. Following the infection, the cells were further incubated for 72 hours to allow for viral replication and expression. After this incubation period, the cells were lysed, and 2.5 μL of lysate from each well was analyzed by using a real-time quantitative PCR (qPCR) assay. The qPCR primers used in the assay were as follows: Forward, 5′-GCTACTGTCAATGGCCACCT-3′, Reverse, 5′-TCCCACAGACAGCTCAATGC-3′, Probe, 5′-FAM-TGTGGTGCTGAG CCCATCCCA-BHQ1-3′. Based on the qPCR results, the wells inoculated with the serial dilutions of rAAV samples were scored for the presence or absence of the virus. The $TCID_{50}$ (infectivity) was then calculated using the Spearman-Karber's method.

## The pI of the AAV capsid was measured by Capillary Isoelectric Focusing (cIEF)

To prepare the rAAV sample for desalting, the following steps were performed[87]. Mix 100 μL of rAAV sample which has a titer of $1.0\times10^{13}$ vg/mL, with 400 μL of ddH$_2$O. Centrifuge at 9,000×g for 10 minutes using an Amicon® Ultra-0.5 device (Merck, 10 K, #UFC501096). Then, 400 μL of ddH$_2$O was added and centrifuged at 9,000×g for 10 minutes. The centrifuged sample was collected and adjust its volume to 100 μL using ddH$_2$O. Next, for the cIEF analysis, the following steps were performed: the master-mix solution was prepared by combining 200 μL of 3 M urea-cIEF gel solution, 12 μL of ampholytes (cytiva, #17045601, pH 3 ~ 10), 20 μL of cathodic stabilizer (500 mM Arginine), 2 μL of anodic stabilizer (200 mM iminodiacetic acid) and 2 μL of each pI marker (such as pI Marker 9.5, pI marker 5.5, pI marker 4.1 from Beckman, #A58481). 10 μL of desalted rAAV samples were mixed with 240 μL of master-mix solution. And transferred into a PA800 Plus instrument (Sciex) for cIEF analysis.

## Neutralizing antibody assays

Neutralizing antibody (Nab) assays were performed on cynomolgus monkey sera using a cell-based AAV-CBA-luciferase or AAV-CBA-eGFP transduction inhibition assay[88,89]. Briefly, the NHPs venous blood samples were collected into serum-separating vacutainer tubes (BD, #367820) and serum supernatant was isolated using centrifuging at 1,500×g for 10 minutes. The study samples were heat-inactivate serum by heating at 56 °C for 30 minutes and then were diluted 10-fold with DMEM/F12 complete medium. 60 μL of diluted samples mixed with 60 μL of $2×10^{11}$ vg/mL AAV-CBA-Luciferase at a MOI of $1.2×10^6$ and then co-incubated at 37 °C for 1 hour. 50 μL of co-incubated solution was transferred into the ARPE19 cells ($1×10^4$ cells/well) in 96-well plate (Corning, #9018). Twenty-four hours later, the Bright-Glo Luciferase assay was measured according to the manufacturer's protocol (Promega, #E2620). Reading RLU values used the chemiluminescence detection module of the multimode microplate reader (Cytation 5). Nab blockade (Blockade, %) is a signal inhibition ratio, a decision value to determine whether the serum samples are Nab-positive. Nab blockade greater than or equal to 50% is defined as Nab-positive; otherwise, it is Nab-negative.

We used serum samples from NHPs that were proven to be positive or negative for AAV8 Nab to test the AAVv128 capsid for Nab escape. Briefly, ARPE19 cells were seeded at $1×10^4$ per well with DMEM/F12 containing 10% fetal bovine serum (FBS) at 37 °C with 5% $CO_2$ in 96-well plates. The following day, the serum samples were diluted in DMEM/F12 + heat inactivated serum using the following dilutions of primate serum: 1:2, 1:10, 1:50, 1:100, 1:250, 1:500, 1:1000, 1:2500, 1:5000, and 1:10000, and mixed with AAV8-CBA-Luciferase or AAVv128-CBA-Luciferase vector ($1.2×10^{10}$ vg/cell). The mixtures were incubated at 37 °C for 1 hour before adding to the prepared ARPE19 cells. And then, the Bright-Glo Luciferase assay was measured after incubating 24 hours.

## AAV binding, trafficking and expression assay

For AAV binding studies, ARPE19 and HeLaRC32 cells were seeded at $3×10^5$ cells/well in 12-well plates and incubated 16 ~ 24 h. Cells were placed to prechilled at 4 °C for 30 min and then the medium was then changed to 1000 μL ice-cold serum-free medium containing rAAVs-CBA-eGFP at 4 °C for 1 hour (capsid: AAV8 or AAVv128, rAAV titer, $1.5×10^{10}$ vg/well for HeLaRC32 cells, MOI = $5×10^4$; $1.5×10^{11}$ vg/well for ARPE19 cells, MOI = $5×10^5$), followed by three washes with ice-cold phosphate-buffered saline (PBS). After gathering cells that had bound rAAVs, viral genomic DNA was isolated and subjected to qPCR measurement.

For cellular uptake studies, following removal of unbound rAAVs, cells were incubated in MEMα or F12 medium containing 10% fetal bovine serum at 37 °C and 5% $CO_2$ for 1 hour. And then, medium was removed, and cells were treated with 0.05% trypsin to dissociate cell-surface associated rAAVs. Cells were transferred to Eppendorf tubes and washes three times with cold PBS. Total viral genomic DNA for quantification was then extracted as described above.

For interrogation of nuclear uptake, HeLaRC32 cells were seeded at $1.0×10^6$ cells/well in 6-well plates and incubated 16 ~ 24 hours. Cells were placed to prechilled at 4 °C for 30 min and then the medium was then changed to 1000 μL ice-cold serum-free medium containing rAAVs-CBA-eGFP at 4 °C for 1 hour (capsid: AAV8 or AAVv128, rAAV titer, $5×10^{11}$ vg/well for HeLaRC32 cells, MOI = $5×10^5$), followed by three washes with ice-cold PBS. Cells were harvested after 4 hours, 6 hours of incubation. Cells which cytoplasmic and nuclear fractions were extracted with the NE-PER™ Nuclear and Cytoplasmic Extraction Reagents kit (Thermo, #78833). Total viral genomic DNA for quantification was then extracted as described above.

For AAV expression assay, ARPE19 and HeLaRC32 cells were seeded at $3.0×10^5$ cells/well in 12-well plates and incubated 16 ~ 24 h.

$1.5×10^{11}$ vg rAAV samples or $1.5×10^{10}$ vg rAAV samples were added to the ARPE19 or HeLaRC32 cells, respectively. Cells were harvested after 16 hours, 24 hours, 40 hours, 48 hours, 60 hours and 72 hours of incubation. The protein expression levels were determined by ELISA, as described above.

## Cryo-EM

In the described experiment, the following steps were taken to obtain a cryo-electron microscopy (cryo-EM) map of the AAVv128 sample: a total of 2.5 μL of AAVv128 (titer,$1.0×10^{13}$ vg/mL) was applied to $H_2/O_2$ glow-discharged 300-mesh Quantifoil R 1.2/1.3 grids (Quantifoil, Micro Tools GmbH, Germany), and subsequently blotted using a FEI Vitrobot and then frozen in liquid ethane. The grids were imaged on a Thermo Fisher Krios G4 microscope (Thermo Fisher Scientific, USA) equipped with a cold field-emission gun, a Selectris X energy filter and a Falcon 4 detector. The energy filter was operated with a slit width of 10 e⁻V to remove inelastically scattered electrons. Image stacks were collected using EPU software at a pixel size of 0.73 Å/pixel with a total dose of 50 e⁻/Å². 2,041 image stacks were subjected to beam-induced motion correction and CTF estimation using cryoSPARC[90]. 728,557 particles were picked automatically and were extracted with a box size of 448 pixels. After several rounds of 2D and 3D classifications, 135,554 particles were used to perform Ab-initio Reconstruction and Heterogenous Refinement. Afterward, the candidate model and 90,744 particles were selected and refined using Homogeneous Refinement with I symmetry applied to generate the final cryo-EM map at a 2.08 Å resolution. Local resolution ranges were also analyzed within cryoSPARC according to the gold-standard Fourier shell correlation (FSC) cut-off of 0.143.

## Structure refinement

The atomic model of AAVv128 was built based on the structure of AAV8 (PDB ID: 6V12). The structure of AAV8 was docked into the EM density map by using UCSF Chimera[91], this was then followed by iterative manual fitting adjustment in Coot[92] and real space refinement in PHENIX[93]. The VR-VIII loop was manually mutated to the corresponding amino acids in the model. All figures were made using UCSF ChimeraX[94] and PyMOL (www.pymol.org)[95].

## Statistical analysis

All statistical analyses were performed by using GraphPad Prism 8.0 (Graphpad). Two-sided Student's $t$-test and one-way ANOVA were used for data comparison. $P$ values less than 0.05 were considered statistically significant.

## Reporting summary

Further information on research design is available in the Nature Portfolio Reporting Summary linked to this article.

## Data availability

The mode and map of AAVv128 underlying Fig. 6, Supplementary Table 7 and Supplementary Fig. 5 is available at the RCSB Protein Data Bank PDB: 8JRE and the Electron Microscopy Database: EMD-36594, respectively. PDB codes of previously published structures used in this study are 6V12 (AAV8). All other data associated with this study are present in the paper or the Supplementary Materials. Source data for each relevant figure is provided in a Source Data file. Source data are provided with this paper.

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

## Acknowledgements

We acknowledge the support by National Natural Science Foundation of China (No. 81925036, 82320108020) and the Fundamental Research Funds for the Central Universities. We would like to thank the Advanced Bio-imaging Technology Platform of Guangzhou National Laboratory for our Cryo-EM data collection. We acknowledge the support by Chengdu Origen Biotechnology Co., Ltd for providing funds for NHPs research.

## Author contributions

S Luo designed, executed, interpreted all the experiments and drafted the paper; FJ. Liu executed and interpreted experiments relate to cryo-EM; S Luo, H Jiang, QW Li and YF Qing, designed, executed, interpreted experiments related to the rabbit and monkey pharmacodynamic tests; S Luo, YF Qing executed and interpreted experiments related to intra-vitreal injections in mice and rabbits; SP Yang, J Li, LL Xu, Y Gou and YF Zhang performed biochemical and physiological properties of rAAV; X Ke, Q Zheng and X Sun supervised the research and prepared the paper.

## Competing interests

X Ke, Q Zheng, S Luo, H Jiang are inventor on patents with potential royalties licensed to Chengdu Origen Biotechnology Co Ltd. Remaining authors declare no competing interests.
