## [Peer Review File · Nature Communications]

REVIEWER COMMENTS

Reviewer #1 (Remarks to the Author):

This study by Luo et al. evaluated their newly developed recombinant adeno-associated virus (rAAV) named AAVv128, adopted based on AAV8 by rational design. AAVs has been used extensively as a delivery vector for various ocular diseases, and having a more effective rAAV capsid with higher infectivity can be beneficial from a translational perspective as it would allow reducing vector dose, where high levels may be associated with ocular inflammation. The novel capsid is highly innovative, and the focus here on suprachoroidal delivery overcomes the surgical complexity of subretinal delivery. The manuscript was clear and logically presented, including the design of the vector, demonstration of efficacy with intravitreal, subretinal, and suprachoroidal injections across mice, rabbits, and nonhuman primates (NHPs), as well as structural analysis to speculate on its improved efficacy. Testing using an anti-VEGF agent to suppress laser-induced CNV provides a nice translational application for this novel platform.

There are several elements that would strengthen this study:

- The rationale for modifying the specific region of VP3 is not well explained. The authors should explain how the modified residues on VP3 impact the binding of the laminin receptor for AAV8.
- The authors switch between different modes of delivery (intravitreal, subretinal, and suprachoroidal) as well as transgene (eGFP, mCherry), but the rationale is not clear. For example, why GFP in mouse and monkeys, but mCherry in rabbits, but not vice versa?
- Because the superior infectivity of v128 can be explained just by different vector dose, rigorous verification of titer equivalence, purity, and empty capsid ratios are helpful to support superior infectivity. For example, check ddPCR and qPCR for titer, SDS-PAGE for purity, and TEM for empty capsid testing. The use of a single viral dose in most experiments, without evidence of dose-dependent changes in efficacy, limits the interpretability of the studies. Finally, the vector dose for experiment should be added to figures and body of Results.
- The robust but peripheral transduction (Fig 4) in the cynomolgus macaque eye is unique, and aligns more with the peripheral transduction reported by PMID 32055646 than PMID 31408444 in rhesus macaques, or PMID 32948113 in rats. This should be elaborated upon in the Discussion section, where the authors should address why the biodistribution of suprachoroidal AAV in macaques may differ from rodents, and how v128 may provide better translatability. The authors should also compare the vector dose used in this study with prior studies to support its superior performance with lower viral titers.
- Given the challenge of suprachoroidal delivery in comparison to subretinal or intravitreal injections, the custom device or needle used should be clearly explained. The use of the infrared thermal camera is strongly supportive, although the video was not available in the review packet download.
- The identity of the anti-VEGF agent in the AAV vector is not clear. The Methods suggest it's a decoy VEGFR1 like sFlt but this was never clarified. If this is similar to the anti-VEGF FAb fragment similar to ranibizumab, or a VEGF-TRAP similar to aflibercept, the comparator pharmacologic agent can be a good positive control.
- The measurement of blood VEGF levels is highly irregular, as intraocular delivery should presumably reduce intraocular VEGF levels without affecting systemic VEGF levels, which could be dangerous. Aqueous or vitreous sampling to quantify levels of VEGF could be considered, although suprachoroidal

delivery may not impact aqueous/vitreous VEGF levels as strongly as intravitreal delivery.

- Why is the anti-VEGF effective on laser CNV when the expression is largely peripheral? If possible, local measurements of anti-VEGF protein levels from retinal and choroidal tissues in the macula vs. periphery would be extremely insightful.

I have more specific comments on the text below:

- Lines 59-61, the authors state, "However, this treatment approach requires frequent injections every 2 to 3 months, posing inconvenience and potential risks to patients." Please elaborate on the potential risks such as endophthalmitis or retinal tears.

- Lines 116-117, what viral dose was used, and why choose only one dosage? Did the authors see a dosage dependent effect in the eGFP mRNA level when they increase or decrease the dosage? This could lay the foundation for the vector dose for rabbit and NHP studies.

- Line 149, the authors claim that subretinal AAV8 delivered to mouse eyes reached its peak eGFP expression at Day 28. However, the data shown in Fig. 2 ended at Day 28. How did the authors know that the eGFP expression did not increase past 28 days?

- Fig. 3: eGFP expression was not consistent within group (v128 vs AAV8). For example, in the first row of v128, eGFP expression could be seen in the ONL but not in other rows? Similarly for AAV8, only the first row showed eGFP expression in the RPE while the other showed eGFP more in the photoreceptor layer.

- Fig. 4a/c: the larger view pictures do not show any DAPI staining as compared to the insets.

- Fig 4e: more magnified views of the injection area showing the spread of the injectate is preferred

- How many eyes per group were used for the NHP blood VEGF study? Based on the paragraph and Fig. 5, only seemed like one animal was reported.

- Fig. 5b: beside measuring anti-VEGF protein from retinal tissues, measurement in choroidal tissues could be more informative for suprachoroidal delivery

- Line 338-339: Clarify what is meant by "...in the VP3 VIII variable region, which exhibited greater spatially flexible."

- Lines 476-477: this statement is false; suprachoroidal injections are rarely used in clinical settings, whereas intravitreal injections are very common, and subretinal injections are less common; suprachoroidal delivery is a relatively novel mode of delivery that is not yet well characterized

- Lines 651-653: more details are needed to clarify what the "self-researched suprachoroidal chamber syringe". If this is not commercially available, the authors should include a photo and explain its design, as compared to other suprachoroidal AAV studies.

- Line 695: Heidelberg Spectralis is not a fundus camera, it's an SLO for near-infrared reflectance.

- Line 708: More details are needed for the laser procedure. Was a macular contact lens used? Is the Vitra a slit lamp, rather than a "total retinoscope" which is not an appropriate term.

- Minor: ARPE19 was spelled incorrectly throughout manuscript.

- Minor: grammatical errors throughout (Lines 376, 383)

- Minor: line 442: did the authors mean "superior nuclear fraction"?

- Ext Fig. 1: Why are the CHX10 and PROX1 expression different between v128 and AAV8 groups?

- Ext Fig. 2: Some additional eyes had local hemorrhages (e.g. last OS eye of vehicle) that should be excluded also.

Reviewer #2 (Remarks to the Author):

In this paper, Luo et al exploit rational design to identify novel adeno-associated viral (AAV) vectors-based capsids, with improved retinal tropism. From an initial in vivo screening, the authors isolate the AAVv128 capsid, which they further characterize both at the structural and functional level. This capsid presents 10 amino acids of difference from the parental AAV8, which the authors propose to critically influence capsid structure and stability and, consequently, interaction with cell surface receptors. To investigate the improved transduction abilities of this novel AAV vectors, the authors explore 3 relevant intraocular delivery routes (i.e. intravitreal, subretinal and suprachoroidal), using as models both mice, rabbits, and non-human primates (NHP). To finally define the therapeutic potential of the novel v128 vector, they explore its use to deliver anti-VEGF in a laser-induced choroidal neovascularization (CNVs) NHP model of age-related macular degeneration.

Data in the CNVs NHP model presented in the manuscript appear promising, showing higher efficiency of the v128 vector, compared to AAV8, in preventing development of neovascular AMD phenotypes when delivered via suprachoroidal injection.

However, despite exploitation of different injection routes and animal models, the manuscript lacks in several instances of a comprehensive characterization of the transduction properties of the vector, to support the claim of v128 being a “highly potent variant enabling efficient ocular-directed gene delivery”, to relevant cell types involved in retinal diseases, as detailed below:

- Intravitreal injection: v128 is found to significantly outperform AAV8 in mice and rabbit; yet, AAV8 very poorly transduces the retina from the vitreous, so it is difficult to understand whether the transduction levels achieved with v128 are therapeutically meaningful. Comparison with AAV serotypes known to result in higher transduction levels from the vitreous (i.e. AAV7m8 or AAV2) appears required. Also, in Fig. 4a, transduction from vitreous appears to result in mCherry expression more efficient in the outer retina and RPE than in the INL/GCL; this is in contrast with what shown in mice (Fig. 1e), where transduction from v128 is exclusively limited to the GCL. Can the author explain this discrepancy?
- Subretinal injection: Fig. 2 convincingly shows overall higher levels of retinal transduction from v128 compared to AAV8; however, given the experimental setting used (inclusion of a ubiquitous promoter in the EGFP expression cassette and harvesting of the whole retina for DNA analysis) it is not clear the transduction levels achieved by v128 in relevant cell types for treatment of retinal diseases (RPE and, more importantly, photoreceptors). Also, evaluation of EGFP genome copy number in retinal tissues (panels j) shows that only half (3 out of 6) of the retinas injected with v128 clearly outperform AAV8. Data in Fig. 3 are included to support increased transduction of several different cell types in the retina from v128, including photoreceptors and ganglion cells. However, data appear not consistent enough to support the claims. Photoreceptors nuclei are clearly transduced only in 1 (first row on the left) out of 7 v128-injected eyes shown in Fig. 3. All the other eyes show green signal only at the level of OS, which is unexpected considering that the EGFP reporter protein should localize evenly in photoreceptors. Indeed, even in the presence of low levels of transduction, as in one of the AAV8-injected eyes (see fifth row of the right part of Fig. 3), EGFP positive nuclei in the ONL are clearly detectable. Eyes injected with vectors encoding for EGFP from a photoreceptor-specific promoter are required to definitively assess levels of transduction from v128 in photoreceptors. Similarly, EGFP positive GC are evident only in the last row of v128-injected eyes (Fig. 3, left part). Therefore, the claim that “v128 vector robustly transduce ganglion cells” appears not supported. The author should explain this apparent lack of consistency. Lastly, the authors should include representative pictures of PBS-injected eyes, as negative controls of the

immunostainings.

- Suprachoroidal injection: as for subretinal injections, more accurate dissection of which cell layers are efficiently transduced from v128 is required. In Extended Data Fig. 1 no clear transduction is seen in both the ONL and IS. In Fig. 4c (and a), the upper panels with low magnification images of the retina are barely visible and, as such, not enough informative on the extent of retinal transduction (including ONL) achieved. The authors comment “v128 predominantly transduced the photoreceptor inner and outer segments (...) after both intravitreal and suprachoroidal injections”, however it is unclear why expression of reporter proteins (as EGFP and mCherry) should be limited to IS/OS in photoreceptors, and not visible in the whole photoreceptor body. Representative pictures of PBS-injected eyes should be included here too. Also, it would be highly informative to add Scanning laser ophthalmoscopy (SLO) images, as in Fig. 4f, for all the NHP eyes injected with EGFP.

Lastly, as a note, reading of the manuscript has been complicated by the not always accurate use of the English language, as well as several typos (as an examples, ARPE19 cells are misspelled as either APRE19 or AREP19 cell throughout the text) and non-sequential order of citation of the figure panels (see Fig.6b-c, cited in line 324, before Fig. 6a, cited in line 330; or Fig. 8f, g, cited in lines 432, before Fig.8d, e, cited in line 435). Also, in this reviewer opinion, the manuscript would be easier to be followed if the characterization of the novel AAV vector (fig.6, 7 and 8) is presented before investigation of its retinal transduction properties and therapeutic efficacy.

Reviewer #3 (Remarks to the Author):

The manuscript by Luo et al describes the rational design, the production, and the characterization of an AAV variant AAVv128. This variant is rationally designed based on the sequence and structure of AAV8. When compared to AAV8, AAVv128 exhibits significantly greater transduction of all ocular cells tested, independent of the mode of application, or the size of the test model ranging from mice to non-human primates. To provide an explanation for the significantly higher transduction phenotype of AAVv128 compared to AAV8, the authors present an extensive comparative biophysical characterization of the AAVv128 and AAV8 by cryo-EM, DSF, and Capillary Isoelectric focusing (cIEF). This manuscript provides valuable information for the AAV gene therapy community, specifically for the treatment of ocular genetic disorders and neovascular age related macular degeneration. The manuscript provides substantial experimental evidence to support the conclusion that AAVv128 is superior to AAV8 in targeting and transducing ocular tissue. There are however several issues that need to be addressed:

1. There are numerous publications citing the use of different AAV2 based vectors for the treatment of different ocular disorders. One of these AAV2 based vectors (Upstaza) have been approved for the treatment of RPE, however all the comparisons have focused on the use of AAVv128 compared to AAV8. Although AAVv128 is superior to AAV8 there is no discussion of its performance compared to other established serotypes used in the treatment of ocular disorders. In the discussion, there is an extensive description of other AAV serotypes and variants that are optimized for treatment in the eye, and while this is important to the discussion it needs to be mentioned in the introduction.

2. Section 3.5 The description of the icosahedral 2-fold, 3-fold and 5-fold axis should be labeled on figure

6A. The secondary structural elements of the model should also be described or referenced. The variable regions should be introduced and rmsd values shown for the superposition of the monomers in 6C. The colors selected for the AAV8 and AAVv128 models are too close and need to be more distinct. The stick models in 6E and 6F should have all the residues labeled and the broken lines representing the interactions removed. The interactions can be shown either in a table or as a table in the figure. It is also not clear from the figure which residues are the insertion and the 'pocket loop'.

3.

4. The statement in Section 3.6 line 374 - 376 is not grammatically correct and needs a reference. The same statement is repeated in 387 – 390 which also needs a reference and proper explanation as to why these physical properties are important for commercial applications.

Minor issues

1. Several grammatical errors, and the sentences need further clarification. A few examples are listed below:

(i) Line 68, ...suspected unexpected serious adverse reaction

(ii) Line 113 – 114, Four weeks after the injection, the expression of eGFP transgene mRNAs in the mouse eyes received different vectors were quantified by qRT-PCR.

(iii) Line 118, ... with a level 75-folds higher than that of AAV8

(iv) Line 376 – 377, To assess the stability of the v128, we conducted experiments compared the temperature 377 dependence of v128 genome release with that of AAV8 at different pHs levels.

(v) Line 605, The study involved the use of Six- to eight-week-old male

2. The term AAVv128 and v128 is used interchangeable, please select one term and use it consistently.

3. It is not clear why the Capillary isoelectric focusing (cIEF) was done. The high resolution Cryo-EM structure show the location of the positive residues in VRIII. It should also be noted that the 588-RGNQQ-592 motif in AAVv128 is most similar to 585-RGNRQ-589 motif in AAV2. The AAV2 residues R585 and R588 have been shown to be critical for the heparin binding phenotype of AAV2. Consequently, a heparin binding assay would potentially provide more informative than a cIEF assay.

Reviewer #4 (Remarks to the Author):

The paper describes the intraocular transduction characteristics of the AAV8-derived capsid variant v128 which outperforms the parental vector upon intravitreal, subretinal and suprachoroidal administrations. v128 does not cross the retinal barriers following intravitreal administration back to the outer retina, however, in general, cells that are AAV8 target upon intraocular injections are transduced at higher levels by v128 across species up to non-human primates. This results in better therapeutic outcome of v128 than AAV8 in a primate model of choroidal neovascularization upon suprachoroidal administration. The cryo EM analysis of v128 highlights differences with AAV8 which could explain its higher transduction characteristics.

The experiments are overall well designed with AAV8 as constant comparison for v128. The evidence of

v128 higher potency than AAV8 is supported by an extensive and comprehensive set of data both in vitro and in the retina, leaving little doubt that v128 should be preferred in the retina to AAV8. It would be interesting to test if this holds true also in liver, another traditional target of AAV8. This could importantly expand v128 therapeutic applications. But of course this goes beyond the scope of the current manuscript.

My only comment is that the authors should explain in detail the rationale for the selection of the regions Q588~A592 or N263~T274 at AAV8 VP3 VIII or VP3 VI variable regions for the mutagenesis. The current explanation is quite vague and provides references other papers but since the whole work starts from there it would be nice to expand this part.

Responses to Reviewers

Reviewer #1 (Remarks to the Author)

This study by Luo et al. evaluated their newly developed recombinant adeno-associated virus (rAAV) named AAVv128, adopted based on AAV8 by rational design. AAVs has been used extensively as a delivery vector for various ocular diseases, and having a more effective rAAV capsid with higher infectivity can be beneficial from a translational perspective as it would allow reducing vector dose, where high levels may be associated with ocular inflammation. The novel capsid is highly innovative, and the focus here on suprachoroidal delivery overcomes the surgical complexity of subretinal delivery. The manuscript was clear and logically presented, including the design of the vector, demonstration of efficacy with intravitreal, subretinal, and suprachoroidal injections across mice, rabbits, and nonhuman primates (NHPs), as well as structural analysis to speculate on its improved efficacy. Testing using an anti-VEGF agent to suppress laser-induced CNV provides a nice translational application for this novel platform.

There are several elements that would strengthen this study:

Comment 1: The rationale for modifying the specific region of VP3 is not well explained. The authors should explain how the modified residues on VP3 impact the binding of the laminin receptor for AAV8.

Response: We appreciate the reviewer's suggestion, we have included more information about the changes made to the VP3 VIII region. Our design approach took into account the importance of R585 and R588, which are key amino acids for the AAV2 cellular receptor heparan sulfate proteoglycan (HSPG)^[1]. These amino acids play a role in interacting with the laminin receptor (LamR)^[2]. Additionally, we consider the role of positively charged amino acids in promoting cell binding and improving immune evasion after cell entry^[3-5]. These factors are carefully considered in our rational design strategy.

The initial receptor responsible for AAV8 binding is currently unknown. However, it has been reported that AAV8 utilizes the 37/67-kDa LamR for cellular transduction^[2]. This interaction is mediated by two stretches of amino acids in AAV8: residues 491 to 557 and 593 to 623^[2]. In the structure of AAVv128, these residues were located at the threefold region of the capsid. Residues 491 to 557 were located within the protrusions surrounding this axis, while residues 593 to 623 were near the threefold axis, consistent with AAV8^[6]. These regions contacted the heparin sulfate binding region^[2]. They were part of the large loop between β G and β H, which contained the AAV variable regions V, VI, VII, and VIII. Additionally, variable regions I and IV were structurally adjacent to these two stretches and had been shown to play a role in tissue transduction and antibody recognition^[7-9].

In addition, AAVv128 incorporated two crucial amino acids, R588 and R596, into the VP3 VIII region. Our findings indicated that these amino acids could potentially enhance the recognition of the cellular receptor HSPG (Extended Fig.11, Extended Table.8) and increase the overall capsid charge (Fig.4b). Moreover, as shown in Fig.5, these amino acids exhibited a significant improvement in cell binding ability (Fig.5d, Fig.5f) and facilitated post-entry cell escape (Fig.5e, Fig.5g-i). While the specific amino acids of AAVv128 that interact with LamR remain unknown, we hypothesize that modifying these amino acids in the VP3 VIII region could potentially enhance the interaction with LamR. The conjecture is supported by the significant improvement in the transduction of cells and tissues by AAVv128, as shown in Fig.2, Fig.3, Fig.5, Fig.7, and Fig.8 of the manuscript. It is also possible that differences in amino acids between the AAV8 and AAVv128 in the two mapped amino stretches also play a role in their different LamR binding phenotypes. To gain deeper insights into their interaction mechanism, we plan to determine the structure of the AAVv128-LamR complex in the future.

To better illustrate the rationale for modifying the specific region of VP3 VIII, the corresponding descriptions were added in the revised manuscript.

Briefly, we used the AAV8 capsid as a scaffold and replaced its amino acid sequences between Q588~ A592 or N263~T274 at the VP3 VIII or VP3 VI variable region. To construct these capsid variants, we designed a set of 18 peptides, ranging from 5 to 14 amino acids in length, based on the known structural biology and cellular receptorology of AAV8³⁵⁻⁴² (Extended Data Table.1). Our design approach considered the significance of R585 and R588 as crucial amino acids for the AAV2 cellular receptor heparan sulfate proteoglycan (HSPG)⁴³. These amino acids also played a role in interacting with the laminin receptor (LamR)⁴⁴. Furthermore, we acknowledged the importance of positively charged amino acids in enhancing cell binding and facilitating immune evasion post-cell entry⁴⁵⁻⁴⁷. These rationally engineered capsid variants were generated by separately packaging a vector genome containing an enhanced green fluorescent protein (eGFP) expression cassette

flanked with AAV2 inverted terminal repeats (ITRs) by using HEK293 cell-triple transfection method (Fig.1a). (**Section**
**Results Line 106-116, Page 5**)

**Extended Data Table.8** | The hydrogen bonding interactions within VR-III loop.

AAVv128	
L586	R596
R587	R596
R588	T594, R596
G589	N593, A595
N590	A595
AAV8	
Q587	T591
Q588	T591

**Reference:**

- [1] Crosson SM, Bennett A, Fajardo D, et al. Effects of Altering HSPG Binding and Capsid Hydrophilicity on Retinal Transduction by AAV. *J Virol.* 2021 Apr 26;95(10):e02440-20.
- [2] Akache B, Grimm D, Pandey K, et al. The 37/67-kilodalton laminin receptor is a receptor for adeno-associated virus serotypes 8, 2, 3, and 9. *J Virol.* 2006 Oct;80(19):9831-6.
- [3] Patel SG, Sayers EJ, He L, et al. Cell-penetrating peptide sequence and modification dependent uptake and subcellular distribution of green fluorescent protein in different cell lines. *Sci Rep.* 2019 Apr 18;9(1):6298.
- [4] Lussi C, de Martin E, Schweizer M. Positively Charged Amino Acids in the Pestiviral Erns Control Cell Entry, Endoribonuclease Activity and Innate Immune Evasion. *Viruses.* 2021 Aug 10;13(8):1581.
- [5] Saadat M, Zahednezhad F, Zakeri-Milani P, et al. Drug Targeting Strategies Based on Charge Dependent Uptake of Nanoparticles into Cancer Cells. *J Pharm Pharm Sci.* 2019;22(1):191-220.
- [6] Nam HJ, Lane MD, Padron E, et al. Structure of adeno-associated virus serotype 8, a gene therapy vector. *J Virol.* 2007 Nov;81(22):12260-71.
- [7] Kern A, Schmidt K, Leder C, et al. Identification of a heparin-binding motif on adeno-associated virus type 2 capsids. *J Virol.* 2003 Oct;77(20):11072-81.
- [8] Lochrie MA, Tatsuno GP, Christie B, et al. Mutations on the external surfaces of adeno-associated virus type 2 capsids that affect transduction and neutralization. *J Virol.* 2006 Jan;80(2):821-34.
- [9] Opie SR, Warrington KH Jr, Agbandje-McKenna M, et al. Identification of amino acid residues in the capsid proteins of adeno-associated virus type 2 that contribute to heparan sulfate proteoglycan binding. *J Virol.* 2003 Jun;77(12):6995-7006.

**Comment 2:** The authors switch between different modes of delivery (intravitreal, subretinal, and suprachoroidal) as well as transgene (eGFP, mCherry), but the rationale is not clear. For example, why GFP in mouse and monkeys, but mCherry in rabbits, but not vice versa?

**Response:** We deeply appreciate the reviewer's suggestion. In our study, we alternate different injection routes for the following reasons. First, in the field of ophthalmology, the AAV8 capsid has demonstrated remarkable transduction capabilities in both photoreceptors and RPE cells, surpassing other AAV serotypes. Leveraging this knowledge, our capsid variants were thoughtfully designed based on the AAV8 framework. To streamline the process of identifying AAV variants with superior transduction capabilities, we chose the simplest and most convenient method of intravitreal injection. After successful screening of the AAV8v128 capsid, we decided to evaluate the transduction potential of the AAV8 variant in photoreceptors and RPE cells by subretinal injection. This approach allowed us to gain valuable insights into the effectiveness and feasibility of this particular variant. However, as our research progressed, we found that suprachoroidal injection offered significant advantages over subretinal injection. Recognizing the potential of this alternative approach, we proceeded to investigate the feasibility and effectiveness of suprachoroidal injection in large animal models.

We used different transgenes, namely eGFP and mCherry, to conduct our experiments in several animal models for the
reasons listed below.

First, we embarked on a separate experiment involving mice, where we specifically selected mCherry as a transgene.
The transduction results obtained with mCherry were comparable to those achieved with eGFP (Extended Date Fig.1). In the
case of monkeys, we chose eGFP as the transgene primarily due to the limitations imposed by the wavelength capabilities of
the equipment (Spectralis HRA OCT, Heidelberg). To acquire in vivo fundus fluorescence data, eGFP was found to be an
optimal choice.

Another crucial aspect of our decision-making process was the selection of mCherry as a transgene for rabbit-based
testing. We employed the mCherry transgene to perform a comprehensive comparison of three different AAV capsids, namely
AAV8, AAV2.7m8, and AAV8v128, through intravitreal and suprachoroidal injection (Extended Date Fig.6). Consequently,
the paper only included the mCherry data, while the eGFP data was not included.

**Extended Date Fig. 1 | Evaluating the transduction of the AAV8 and AAVv128 capsids with different transgenes after subretinal injection.**

Note: **a** Fluorescence funduscopy of mouse eyes treated with ssAAV-CBA-eGFP/mCherry vectors packaged with AAV8 or AAVv128 capsids
(1×10^9 vg/eye, 1 μ L volume, n=4). Mice were imaged at D28 post-injection, and mice were sacrificed at D28 and eyes were harvested. **b** Coronal
sections of mice retina transduced with rAAV-CBA-eGFP/mCherry. Native eGFP expression (green) or mCherry expression (Red) shows the
positively transduced cell. Scale bar= 200 μ m

**Extended Date Fig.6 | Intraocular injections in New Zealand rabbits to evaluate the transduction efficacy of AAV8, AAVv128 and**
**AAV2.7m8**

**Note:** The New Zealand rabbit eyes treated by intravitreal injections or suprachoroidal injections of the AAV8 and AAVv128 confers detectable
mCherry expression at days 28 post-injection (1×10^{11} vg/eye, 100 μ L volume). Scale bars= 100 μ m

**Comment 3:** Because the superior infectivity of v128 can be explained just by different vector dose, rigorous verification of
titer equivalence, purity, and empty capsid ratios are helpful to support superior infectivity. For example, check ddPCR and
qPCR for titer, SDS-PAGE for purity, and TEM for empty capsid testing. The use of a single viral dose in most experiments,
without evidence of dose-dependent changes in efficacy, limits the interpretability of the studies. Finally, the vector dose for
experiment should be added to figures and body of Results.

**Response:** Our deepest gratitude goes out to you for your thoughtful suggestions, which have helped to significantly improve
this paper substantially. As already mentioned, we have carefully incorporated the necessary information as explained
previously. In this context, we examined the results of the ddPCR assay for genomic titers. To ensure accuracy and reliability,
we conducted three replicates for each sample, with each test performed independently. The genomic titers of the AAV8-anti-
VEGF vector were determined to be 2.36×10^{13} vg/mL, 2.23×10^{13} vg/mL, and 2.17×10^{13} vg/mL, respectively. Similarly, the
genomic titers of the AAVv128-anti-VEGF vector were measured to be 2.18×10^{13} vg/mL, 2.07×10^{13} vg/mL, and 2.16×10^{13}
vg/mL, respectively (Extended Date Fig.2). These measurements and repetitions increased confidence in the reliability of our
ddPCR assay results.

Furthermore, we conducted an SDS-PAGE analysis, the results of which were shown in Extended Data Fig.2a. The purity
of the AAV8 and AAVv128 vectors was found to be comparable, further confirming the integrity and quality of our
experimental samples. To gain further insight into the properties of the viruses, we carried out further experiments using
Transmission Electron Microscopy (TEM) and Sedimentation Velocity Analytical Ultracentrifugation (SV-AUC). The results
of these analyzes were presented in Extended Data Fig.2b-c, which allowed us to determine the empty-to-full capsid ratio of
the virus. For the AAV8-anti-VEGF vector, the empty and full ratios were calculated to be 27.79% and 63.18%, respectively.
Similarly, for the AAVv128-anti-VEGF vector, the empty and full ratios were determined to be 32.13% and 60.85%,
respectively. These results further strengthened the comparability and consistency of our results. In summary, based on the
comprehensive evaluations of the rAAV samples we carried out, we are convinced of the satisfactory quality and reliability
of our experimental materials. We have considered the reviewers' comment regarding the inclusion of the vector dose for the
experiment in the figures and the main body of the Results section.

To better illustrate the results, the corresponding descriptions were added in the revised manuscript.

Before conducting the animal efficacy studies, we thoroughly validated the quality of the AAV8-anti-VEGF vector and
AAVv128-anti-VEGF vector (Extended Date Fig.2). The results showed comparable purity and empty and full ratios for both
vectors. In summary, based on our comprehensive evaluation of the rAAV samples, we have confidence in the satisfactory
quality and reliability of our experimental materials. (*Section Results Line 321-324, Page 12*)

Extended Date Fig.2 | The quality of AAV8 and AAVv128 samples for evaluating the treatment efficacy for nAMD studies. **a**, SYPRO Ruby-stained SDS-PAGE analysis of anion exchange chromatography-purified AAV8 and AAVv128. **b**, Transmission electron microscopy (TEM) micrographs of AAV8 and AAVv128. Full (bright spheres) and empty (spheres with darker spot). **c**, rAAV samples were analyzed by using Sedimentation velocity analytical Ultracentrifugation (SV-AUC) to determine the AAV8 or AAVv128 empty-to-full capsid ratio. Scale bars= 200 μ m

Note: we examined the results obtained from the ddPCR assay for genomic titer. To ensure accuracy and reliability, we conducted three repetitions for each sample, with each assay being operated independently. The genomic titers of the AAV8-anti-VEGF vector were determined to be 2.36×10^{13} vg/mL, 2.23×10^{13} vg/mL, and 2.17×10^{13} vg/mL, respectively. Similarly, the genomic titers of the AAVv128-anti-VEGF vector were measured to be 2.18×10^{13} vg/mL, 2.07×10^{13} vg/mL, and 2.16×10^{13} vg/mL, respectively.

Comment 4: The robust but peripheral transduction (Fig 4) in the cynomolgus macaque eye is unique, and aligns more with the peripheral transduction reported by PMID 32055646 than PMID 31408444 in rhesus macaques, or PMID 32948113 in rats. This should be elaborated upon in the Discussion section, where the authors should address why the biodistribution of suprachoroidal AAV in macaques may differ from rodents, and how v128 may provide better translatability. The authors

should also compare the vector dose used in this study with prior studies to support its superior performance with lower viral
titers.

**Response:** We are grateful for the suggestion. It was revealed that the suprachoroidal delivery of AAV8 in Sprague Dawley
rats mainly transduced photoreceptors and retinal pigment epithelium (RPE) [1]. This suprachoroidal delivery resulted in a
larger average transduction diameter and an abundance of transduced cells within the outer nuclear layer (ONL). However,
no transduction of choroidal endothelial cells or other choroidal cell types occurred regardless of the route of administration,
including suprachoroidal injection [1]. In monkeys, suprachoroidal delivery of AAV8 caused diffuse distribution with
widespread expression in the peripheral RPE. Suprachoroidal delivery also transduced photoreceptors and outer retinal layers,
including the photoreceptor inner segment/outer segment junctions (IS/OS) and the external limiting membrane (ELM) in
monkeys [2]. There may be three possible reasons for the differences between them.

First, there are significant differences between species, whose implications reverberate through the species selection and
the interpretation of drug-related ocular changes. The human fundus most closely resembles that of the non-human primate
species commonly used in the laboratory, including specialized structures not present in other laboratory species (rat, mouse),
such as the macula and fovea [3].

Second, a significant barrier to ocular penetration appears to be Bruch's membrane (BrM) and the vascular choroidal
layer which can easily carry away a drug (for example, AAV) in the circulation [4]. The dense proteoglycan content of BrM
plays a crucial role in this process, as it can sequester viral-based vectors and repel non-viral formulations through electrostatic
interactions [4]. In the eyes of Sprague Dawley rats, BrM has a structure similar to that of human and monkey eyes, albeit with
a significantly thinner profile. The membrane thickness in rats ranges from 0.29 μ m to 1.14 μ m, accounting for approximately
40% of the corresponding area in the human eye [5]. In monkeys, the width of BrM varies from 0.39 μ m to 1.62 μ m [5]. The
relation of the pigment epithelial cell to its basement membrane, and the relation of the basement membrane to the collagenous
zones in rat eye, was found to be the same as in the monkey except for the smaller dimensions [5].

Third, the thickness or number of cells within the retinal layers can vary significantly depending on the species. A
comparison between the retinas of a cynomolgus monkey and a Sprague Dawley rat revealed distinct differences. In monkey
retina, the nerve fiber layer, ganglion cell layer, inner nuclear layer and outer plexiform layer were found to be thicker
compared to rat retina. Conversely, the inner plexiform layer and the outer nuclear layer in the monkey retina were observed
to be thinner than those in the rat retina [3]. These variations in retinal layer thickness and cellular density highlight species-
specific features of retinal structure.

In a study, it was observed that the administration of a lower dose of AAV8 (7×10^{11} vg/eye, rhesus 01) did not result in
any detectable expression at any time point. However, when a higher dose of 7×10^{12} vg/eye (rhesus 02) was injected, scattered
areas of punctate eGFP fluorescence were observed bilaterally in the peripheral fundus [2]. In contrast, the use of the AAVv128
vector resulted in a significant increase in eGFP fluorescence in the peripheral fundus compared to AAV8 (dose: 3.5×10^{12}
vg/eye). Additionally, it was reported that the delivery of AAV via the suprachoroidal route in monkeys tends to distribute
circumferentially along the periphery rather than posteriorly towards the macula [6,7]. This suggests that suprachoroidal AAV
delivery may be more suitable for gene therapies targeting peripheral or RPE diseases, such as the production of anti-
angiogenesis agents for neovascular retinal conditions, rather than macular diseases [2]. Furthermore, protein level distribution
was assessed in monkey retinal tissue using a dose of 1.0×10^{12} vg/eye. The NHP retinal tissue and other tissues treated with
AAVv128 had a higher concentration of the anti-VEGF protein compared to the prototype AAV8. Specifically, there was a
56-fold increase in the retinal tissue of the left eye (OS, oculus sinister) and a 76-fold increase in the retinal tissue of the right
eye (OD, oculus dextrus) compared to AAV8 (Fig.7a, Extended Data Table.4). These findings suggest that the novel AAVv128
capsid holds promise as a potential gene delivery vehicle for the treatment of ocular fundus diseases of the retina.

To better illustrate the results, the corresponding discussions were added in the revised manuscript.

The transduction cell types of AAVv128 in these eyes were mainly photoreceptors and RPE cells (Extended Data Fig.8). It
was shown that suprachoroidal delivery of AAV8 in Sprague-Dawley rats mainly resulted a larger average transduction
diameter and an abundance of transduced cells within the outer nuclear layer (ONL) [72]. This could be due to the structural
differences between species that result in different transduction cell types, including Bruch's membrane (BrM) thickness, the
thickness or number of cells within the retinal layers [73,74]. (**Section Discussion Line 392-397, Page 14**)

In a study, it was observed that the administration of a lower dose of AAV8 (7×10^{11} vg/eye, rhesus 01) did not result in
any detectable expression at any time points. However, when a higher dose of 7×10^{12} vg/eye (rhesus 02) was injected, scattered

areas of punctate eGFP fluorescence were observed bilaterally in the peripheral fundus³³. In contrast, the use of the AAVv128 vector led to a significant increase in eGFP fluorescence in the peripheral fundus compared to AAV8 (dose: 3.5×10^{12} vg/eye). Additionally, it was reported that suprachoroidal delivery of AAV in monkey might have a greater tendency to distribute circumferentially along the periphery rather than posteriorly toward the macula^{70,71}. Suprachoroidal AAV delivery might be better suited for gene therapies that target peripheral or RPE diseases, for example the production of anti-angiogenesis agents for neovascular retinal conditions, rather than macular diseases³³. (**Section Discussion Line 384-392, Page 17**)

Extended Data Table.4 | Concentrations of the transgene product (anti-VEGF protein) in different tissues of rhesus monkeys after a single suprachoroidal injection in bilateral eyes (ng/g for solid tissues or ng/mL for liquid tissues, n=1)

No.	Aqueous humor*		Choroid		Conjunctiva		Iris/Ciliary body		Retina		Sclera		Vitreous body*	
	OS	OD	OS	OD	OS	OD	OS	OD	OS	OD	OS	OD	OS	OD
AAV8	BLQ	BLQ	31.38	BLQ	BLQ	BLQ	41.22	55.32	37.68	50.82	BLQ	BLQ	BLQ	BLQ
AAVv128	9.95	12.1	1032	1638	BLQ	48.48	502.2	134.4	2112	4056	84	48.42	36	44.9

Note: “*” means liquid tissues, with the unit of ng/mL; “-” means no tissue; BLQ means below the lower limit of quantification.

Reference:

- [1] Han IC., Cheng JL., Burnight ER., et al. Retinal Tropism and Transduction of Adeno-Associated Virus Varies by Serotype and Route of Delivery (Intravitreal, Subretinal, or Suprachoroidal) in Rats. *Hum Gene Ther.* 2020 Dec;31(23-24):1288-1299.
- [2] Yiu G., Chung SH., Mollhoff IN., et al. Suprachoroidal and Subretinal Injections of AAV Using Transscleral Microneedles for Retinal Gene Delivery in Nonhuman Primates. *Mol Ther Methods Clin Dev.* 2020 Jan 21;16:179-191.
- [3] Andrea B Weir, Margaret Collins. Comparative ocular anatomy in commonly used laboratory animals. In *Assessing Ocular Toxicology in Laboratory Animals*, 2013, pp. 1-21, Humana Press, New York.
- [4] Hammadi S., Tzoumas N., Ferrara M., et al. Bruch’s Membrane: A Key Consideration with Complement-Based Therapies for Age-Related Macular Degeneration. *J. Clin. Med.* 2023, 12, 2870.
- [5] Nakaizumi Y. The Ultrastructure of Bruch's Membrane: II. Eyes With a Tapetum. *Arch Ophthalmol.* 1964;72(3):388–394.
- [6] Moisseiev E, Loewenstein A, Yiu G. The suprachoroidal space: from potential space to a space with potential. *Clin Ophthalmol.* 2016 Jan 25;10:173-8.
- [7] Emami-Naeini P, Yiu G. Medical and Surgical Applications for the Suprachoroidal Space. *Int Ophthalmol Clin.* 2019 Winter;59(1):195-207.

Comment 5: Given the challenge of suprachoroidal delivery in comparison to subretinal or intravitreal injections, the custom device or needle used should be clearly explained. The use of the infrared thermal camera is strongly supportive, although the video was not available in the review packet download.

Response: We express our gratitude to the reviewer for your valuable recommendations. The device utilized for suprachoroidal injection was a customized combination of an adapter, a 1 mL syringe, and a needle. The adapter incorporated a unique internal design that enabled the syringe and needle to lock after insertion, allowing control over the exposed length of the needle. The length of the needle tip was adjusted by turning the fitting at the top of the adapter. For suprachoroidal injection in monkeys, a needle tip length of 0.7 mm was used. Due to the pending patent status of this device, no further details can be disclosed at this time.

To better illustrate the results, the corresponding descriptions were added in the revised manuscript.

For the suprachoroidal injection, the device utilized for suprachoroidal injection was a customized combination of an adapter, a 1 mL syringe, and a needle. The adapter incorporated a unique internal design that enabled the syringe and needle to lock after insertion, allowing control over the exposed length of the needle. The length of the needle tip was adjusted by turning the fitting at the top of the adapter. For suprachoroidal injection in monkeys, a needle tip length of 0.7 mm was used. Additionally, we have uploaded a video capturing the process of suprachoroidal injection in a monkey, recorded using an infrared thermal camera (Extended Data Video.1). (**Section Method Line 838-843, Page 25**)

Comment 6: The identity of the anti-VEGF agent in the AAV vector is not clear. The Methods suggest it’s a decoy VEGFR1

like sFlt but this was never clarified. If this is similar to the anti-VEGF FAb fragment similar to ranibizumab, or a VEGF-
TRAP similar to aflibercept, the comparator pharmacologic agent can be a good positive control.

**Response:** We appreciate the suggestion and in response to the reviewer's concerns, we have provided a more comprehensive
interpretation of the information regarding the anti-VEGF protein. The coding region of the anti-VEGF protein includes
domain 2 of human VEGFR1, domain 3 of human VEGFR2, and the Fc portion of human IgG1. (Section Method Line
754-755, Page 24)

**Comment 7:** The measurement of blood VEGF levels is highly irregular, as intraocular delivery should presumably reduce
intraocular VEGF levels without affecting systemic VEGF levels, which could be dangerous. Aqueous or vitreous sampling
to quantify levels of VEGF could be considered, although suprachoroidal delivery may not impact aqueous/vitreous VEGF
levels as strongly as intravitreal delivery.

**Response:** We sincerely appreciate the reviewer for bringing this issue to our attention. As the reviewer mentioned, in
subsequent experiments we also found that the concentration of VEGF in the blood varies significantly between individuals
and at different time points. Therefore, we acknowledge that these data may not accurately reflect the efficacy of our AAVv128
capsid. To demonstrate the superiority of our AAVv128 capsid, we replaced these data by measuring anti-VEGF protein levels
in the aqueous humor of NHPs. As shown in Extended Data Table 5, we detected the expression of anti-VEGF protein in all
8 eyes treated with AAVv128, while protein expression was not observed in any of the 8 eyes in the AAV8-treated group.
Although we did not specifically examine the level of VEGF in the aqueous humor or vitreous, the detection of anti-VEGF
protein expression in the aqueous humor suggests that the AAVv128-treated group may indeed reduce the level of VEGF in
the aqueous humor.

To better illustrate the results, the corresponding descriptions were added in the revised manuscript.

The anti-VEGF protein level of AAV8 and AAVv128 in the aqueous humor were measured on day 35. A higher concentration
of the anti-VEGF protein was found in the AAVv128-treated NHP aqueous humor, compared to the prototype AAV8 (n=8,
Extended Date Table.5). (Section Results Line 338-341, Page 15)

**Extended Date Table.5** | Concentrations of the transgene product (anti-VEGF protein) in aqueous humor of rhesus monkeys after a single
suprachoroidal injection in bilateral eyes (ng/mL)

No.	1F001				1F002				1M001				1M002				4F001				4F002				4M001				4M002			
	AAV8								AAVv128																							
	OS	OD	OS	OD	OS	OD	OS	OD	OS	OD	OS	OD	OS	OD	OS	OD	OS	OD	OS	OD	OS	OD	OS	OD	OS	OD						
	BLQ	BLQ	BLQ	BLQ	BLQ	BLQ	BLQ	BLQ	BLQ	BLQ	BLQ	BLQ	BLQ	BLQ	BLQ	BLQ	BLQ	BLQ	BLQ	BLQ	88.2	39.3	172.1	189.7	119.5	57.8	24.6	26.9				

Note: BLQ means below the lower limit of quantification.

**Comment 8:** Why is the anti-VEGF effective on laser CNV when the expression is largely peripheral? If possible, local
measurements of anti-VEGF protein levels from retinal and choroidal tissues in the macula vs. periphery would be extremely
insightful.

**Response:** We thank the reviewer for this remark. Previous studies have suggested that suprachoroidal delivery of AAV in
monkeys tends to distribute more circumferentially along the periphery rather than posteriorly towards the macula ^[1,2]. This
suggests that suprachoroidal AAV delivery may be more suitable for gene therapies targeting peripheral or RPE diseases, such
as the production of anti-angiogenesis agents for neovascular retinal conditions, rather than macular diseases ^[3]. The anti-
VEGF protein contains a secretion signal peptide that enables effective extracellular secretion. Therefore, its distribution in
the periphery should have no impact on its effectiveness. We appreciate the reviewer's suggestion and acknowledge that we
are currently unable to measure the anti-VEGF protein levels in retinal and choroidal tissues during the laser CNV efficacy
study. We are continuously monitoring the monkeys for long-term expression of anti-VEGF protein levels (n=8). We will
consider isolating retinal and choroidal tissues in the macula and periphery to locally measure anti-VEGF protein levels at the
end of the trial in the future.

Additionally, we measured the levels of anti-VEGF proteins in other tissues of the cynomolgus macaque eye (n=1), as
listed in Extended Data Table 4. In the AAVv128-treated group, the anti-VEGF protein level was mainly distributed in the
retina and choroid. In contrast, the AAV8 group showed generally low protein expression of anti-VEGF, which was only
detected in the retina and choroid.

**Extended Data Table.4** | Concentrations of the transgene product (anti-VEGF protein) in different tissues of rhesus monkeys after a single
suprachoroidal injection in bilateral eyes (ng/g for solid tissues or ng/mL for liquid tissues, n=1)

No.	Aqueous humor*		Choroid		Conjunctiva		Iris/Ciliary body		Retina		Sclera		Vitreous body*	
	OS	OD	OS	OD	OS	OD	OS	OD	OS	OD	OS	OD	OS	OD
AAV8	BLQ	BLQ	31.38	BLQ	BLQ	BLQ	41.22	55.32	37.68	50.82	BLQ	BLQ	BLQ	BLQ
AAVv128	9.95	12.1	1032	1638	BLQ	48.48	502.2	134.4	2112	4056	84	48.42	36	44.9

Note: “**” means liquid tissues, with the unit of ng/mL; “-” means no tissue; BLQ means below the lower limit of quantification.

Reference:

- [1] Moisseiev E, Loewenstein A, Yiu G. The suprachoroidal space: from potential space to a space with potential. *Clin Ophthalmol.* 2016 Jan
25;10:173-8.
[2] Emami-Naeini P, Yiu G. Medical and Surgical Applications for the Suprachoroidal Space. *Int Ophthalmol Clin.* 2019 Winter;59(1):195-207.
[3] Yiu G., Chung SH., Mollhoff IN., et al. Suprachoroidal and Subretinal Injections of AAV Using Transscleral Microneedles for Retinal Gene
Delivery in Nonhuman Primates. *Mol Ther Methods Clin Dev.* 2020 Jan 21;16:179-191.

I have more specific comments on the text below:

**Comment 9:** Lines 59-61, the authors state, “However, this treatment approach requires frequent injections every 2 to 3
312 months, posing inconvenience and potential risks to patients.” Please elaborate on the potential risks such as endophthalmitis
or retinal tears.

**Response:** We appreciate the reviewer's suggestion and have incorporated additional details regarding the potential risks
associated with this treatment approach.

However, it is important to note that this approach necessitates frequent injections every 2 to 3 months, which can be
inconvenient for patients and may pose certain risks. These risks include subconjunctival and vitreal hemorrhages, corneal
edema, conjunctival scars, retinal tears and detachment, lens damage, development of cataracts, choroidal rupture, ocular
hypertension, and endophthalmitis²⁵⁻²⁸. (**Section Introduction Line 65-68, Page 4**)

**Comment 10:** Lines 116-117, what viral dose was used, and why choose only one dosage? Did the authors see a dosage
dependent effect in the eGFP mRNA level when they increase or decrease the dosage? This could lay the foundation for the
vector dose for rabbit and NHP studies.

**Response:** We sincerely appreciate the reviewer's valuable suggestion. AAV variants were administered at 6×10^8 vg/eye. Our
aim was to perform a preliminary screening of the 18 AAV variants that we rationally designed, with the aim of identifying
those with higher mRNA expression levels. Therefore, we chose a single dose of 6×10^8 vg/eye, which is commonly used in
the literature^[1], based on the current virus titer. We did not employ multiple doses to measure eGFP mRNA levels. After
obtaining the results for eGFP mRNA expression, we repeated the intravitreal injection of AAVv121, AAVv123, AAVv124,
AAVv125, AAVv128, AAVv129, and AAVv1213, which exhibited higher mRNA levels. The results of these two experiments
were consistent. After selecting AAVv128, we conducted a detailed assessment of its transduction efficiency and the cell types
it transduced in mice. Only after confirming the validity of the mouse tests did we proceed to test it in rabbits. The injection
dose in rabbits was determined based on the dose reported in the literature^[2]. Thanks to the reviewer's suggestions, we will
evaluate the transduction efficiency of the AAV variants in the future screening procedure using multiple doses.

To better illustrate the results, the corresponding descriptions were added in the revised manuscript.

Four weeks after injection, a careful examination was conducted to assess the expression of eGFP transgene mRNAs in the
eyes of mice exposed to different vectors (6×10^8 vg/eye, n=3). This assessment was carried out using the quantitative reverse

transcription polymerase chain reaction (qRT-PCR). (*Section Results Line 120-123, Page 5*)

Reference:

[1] Nieuwenhuis B, Laperrousaz E, Tribble JR, et al. Improving adeno-associated viral (AAV) vector-mediated transgene expression in retinal ganglion cells: comparison of five promoters. *Gene Ther.* 2023 Jun;30(6):503-519.

[2] Marangoni D, Wu Z, Wiley HE, et al. Preclinical safety evaluation of a recombinant AAV8 vector for X-linked retinoschisis after intravitreal administration in rabbits. *Hum Gene Ther Clin Dev.* 2014 Dec;25(4):202-11.

Comment 11: Line 149, the authors claim that subretinal AAV8 delivered to mouse eyes reached its peak eGFP expression at Day 28. However, the data shown in Fig. 2 ended at Day 28. How did the authors know that the eGFP expression did not increase past 28 days?

Response: We apologize for the confusion caused by the description in the original article. According to the literature, the FVIII protein level reached its peak at Day 28 after intravenous delivery of AAV8-FVIII (5×10^{11} vg per mouse) in mice ($n = 4 \sim 6$)^[1]. However, other studies suggested that the maximum expression of AAV8-delivered transgenes occurs after 21 or 56 days^[2-3]. Therefore, it is not appropriate to directly state that subretinal AAV8 delivered to mouse eyes reached its highest eGFP expression on Day 28, and we revised the manuscript as follows.

It was generally assumed that AAV8 would reach peak expression after 21 days⁴⁸⁻⁵⁰. Interestingly, the mean pixel intensity per pixel area of AAVv128 increased significantly from day 0 to day 14, but only slightly between days 14 and 28. This observation suggests that the AAVv128 vector has a faster onset of transgene expression compared to AAV8 (Fig.2e-f). (*Section Results Line 142-145, Page 6*)

Reference:

[1] Nguyen GN, George LA, Siner JI, et al. Novel factor VIII variants with a modified furin cleavage site improve the efficacy of gene therapy for hemophilia A. *J Thromb Haemost.* 2017 Jan;15(1):110-121.

[2] Vandenberghe LH, Xiao R, Lock M, et al. Efficient serotype-dependent release of functional vector into the culture medium during adeno-associated virus manufacturing. *Hum Gene Ther.* 2010 Oct;21(10):1251-7.

[3] Zincarelli C, Soltys S, Rengo G, et al. Analysis of AAV serotypes 1-9 mediated gene expression and tropism in mice after systemic injection. *Mol Ther.* 2008 Jun;16(6):1073-80.

Comment 12: Fig. 3: eGFP expression was not consistent within group (v128 vs AAV8). For example, in the first row of v128, eGFP expression could be seen in the ONL but not in other rows? Similarly for AAV8, only the first row showed eGFP expression in the RPE while the other showed eGFP more in the photoreceptor layer.

Response: Thanks to the reviewers for pointing out the issue. To obtain a more comprehensive knowledge of the transduced cells after subretinal injection of AAVv128, we conducted the experiment again and captured new images using the Olympus SpinSR10 spinning disc confocal super-resolution microscope. The updated results were presented in Fig.3. AAVv128 showed almost complete ONL layer transduction and a broader range of eGFP transduction levels compared to AAV8. However, AAV8 showed a greater diversity of eGFP fluorescence signals only near the cornea. Other retinal areas had lower eGFP expression, with transduction observed exclusively in photoreceptors and RPE cells. We used different antibodies to identify the transduction cell types of AAVv128, as shown in Fig.3. AAVv128 was able to transduce rod, cone, RPE, and horizontal cells. Nevertheless, Müller cells, astrocytes, ganglion cells, and anaplastic cells did not show transduction in response to AAVv128. On the other hand, AAV8 transduced only a tiny number of RPE cells and photoreceptors at the same dosage.

Fig.3 Cross-section and immunofluorescence analyses of mice retina treated with subretinal injection of AAV8 and AAVv128 capsid.
 Note: **a** Coronal sections of mice retina transduced with rAAV-CBA-eGFP. Native eGFP expression (green) shows the positively transduced cell. Scale bar= 200 μ m. **b** IF stained sections (red) with antibodies against Rhodopsin (photoreceptors, Rod), Peanut agglutinin (PNA, photoreceptors, Cone), RPE65 (RPE cells), GS (Müller cells), GFAP (astrocytes), RBPMS (ganglion cells), PROX1 (anaplastic cells and ganglion cells) and Calbindin D28K (horizontal cells, anaplastic cells and ganglion cells) indicate the distribution of cell types across the retina. Native eGFP expression (green) that colocalize with IF staining (yellow or white) reveals the positively transduced cell type indicated. Scale bars= 20 μ m (right column) or 200 μ m (left column). INL: inner nuclear layer, ONL: outer nuclear layer

Comment 13: Fig. 4a/c: the larger view pictures do not show any DAPI staining as compared to the insets.

Response: Thank you for your careful review. We added the DAPI staining in the larger view pictures.

Fig.7 Intraocular injections in large animals to evaluate the transduction efficacy of AAV8 and AAVv128

Note: The New Zealand rabbit eyes treated by intravitreal injections or suprachoroidal injections of the AAV8 and AAVv128 confers detectable mCherry expression at days 28 post-injection (1×10^{11} vg/eye, 100 μ L volume, n=4). **a,c** Immunofluorescence analysis of the New Zealand rabbit retinas after intravitreal injection (**a**) and suprachoroidal injection (**c**). **b,d** The retina-choroid tissues of the New Zealand rabbit retinas after intravitreal injection (**b**) and suprachoroidal injection (**d**) were isolated and their viral genomic DNA content were measured by qPCR (n=4). NHPs eyes treated by suprachoroidal injection of the AAV8 and AAVv128 capsid confers detectable eGFP expression at days 14 post-injection (3.5×10^{12} vg/eye, 100 μ L volume, n=1). **e** Infrared imaging analysis of suprachoroidal injections in cynomolgus monkeys. **f** Transduction validation of NHPs eyes treated with AAV8 and AAVv128 by Scanning laser ophthalmoscopy (SLO). Scale bars= 50 μ M (down column) or 200 μ M (up column); GCL: ganglion cell layer, INL: inner nuclear layer, ONL: outer nuclear layer. Values represent mean \pm SD. p Values were determined by *t*-test. #p < 0.05, ##p < 0.01, ###p < 0.001, ####p < 0.0001.

Comment 14: Fig 4e: more magnified views of the injection area showing the spread of the injectate is preferred

Response: We thank the reviewer for this remark. We have added three magnified images to show the beginning, middle and end of the injection process (Extended Data Fig.7a-c). We have also uploaded the video of the SCS injection process recorded by an infrared thermal camera.

**Extended Date Fig.6 | Infrared imaging analysis of suprachoroidal injections in cynomolgus monkeys.**

**Comment 15:** How many eyes per group were used for the NHP blood VEGF study? Based on the paragraph and Fig. 5, only
seemed like one animal was reported.

**Response:** In response to the reviewer's question, we acknowledge that only one animal in each group was used to measure
VEGF in NHP blood. As the reviewer pointed out, we later discovered through additional experiments that the
concentration of VEGF in the blood varies significantly between individuals and at different time points. Therefore, these
data cannot accurately reflect the efficacy of our AAVv128 capsid. To address this limitation, we replaced the data with
measurements of the expression of anti-VEGF protein levels in the aqueous humor of NHPs. This approach allows us to
demonstrate the superiority of our AAVv128 capsid (Extended Date Table.5). It is important to note that the NHPs currently
being used to evaluate the efficacy of nAMD are still in the breeding stage. Therefore, we were unable to measure the
expression of anti-VEGF protein levels in other tissues of the eye.

**Extended Date Table.5 | Concentrations of the transgene product (anti-VEGF protein) in aqueous humor of rhesus monkeys after a single**
**suprachoroidal injection in bilateral eyes (ng/mL)**

No.	1F001				1F002				1M001				1M002				4F001		4F002		4M001		4M002	
	AAV8								AAVv128															
	OS	OD	OS	OD	OS	OD	OS	OD	OS	OD	OS	OD	OS	OD	OS	OD	OS	OD	OS	OD	OS	OD		
	BLQ	BLQ	BLQ	BLQ	BLQ	BLQ	BLQ	BLQ	BLQ	BLQ	BLQ	BLQ	BLQ	BLQ	BLQ	88.2	39.3	172.1	189.7	119.5	57.8	24.6	26.9	

Note: BLQ means below the lower limit of quantification.

**Comment 16:** Fig. 5b: beside measuring anti-VEGF protein from retinal tissues, measurement in choroidal tissues could be
more informative for suprachoroidal delivery

**Response:** Thanks to the reviewer's suggestions, we have conducted measurements of anti-VEGF protein levels in various
tissues of the eye. The results of these measurements have been included in Extended Data Table 4.

**Extended Date Table.4 | Concentrations of the transgene product (anti-VEGF protein) in different tissues of rhesus monkeys after a single**
**suprachoroidal injection in bilateral eyes (ng/g for solid tissues or ng/mL for liquid tissues)**

No.	Aqueous humor*		Choroid		Conjunctiva		Iris/Ciliary body		Retina		Sclera		Vitreous body*	
	OS	OD	OS	OD	OS	OD	OS	OD	OS	OD	OS	OD	OS	OD
AAV8	BLQ	BLQ	31.38	BLQ	BLQ	BLQ	41.22	55.32	37.68	50.82	BLQ	BLQ	BLQ	BLQ
AAVv128	9.95	12.1	1032	1638	BLQ	48.48	502.2	134.4	2112	4056	84	48.42	36	44.9

Note: "*" means liquid tissues, with the unit of ng/mL; "-" means no tissue; BLQ means below the lower limit of quantification.

**Comment 17:** Line 338-339: Clarify what is meant by "...in the VP3 VIII variable region, which exhibited greater spatially
flexible."

**Response:** Thank you for your careful review. We are very sorry for the mistakes in this manuscript. We have revised as:
The AAVv128 and AAV8 structures were similar, except for the 3-fold protrusions (Root-mean-square deviation (RMSD) of
0.507 Å after superimposition of 516 Cα atoms; Fig.6b-c). The 3-fold protrusions were more pronounced in AAVv128 due to
the 5-amino acid insertion, which protruded radially outward from the capsid surface, forming a larger loop that had greater

spatial flexibility (Fig.6b-c). The 3-fold region, including the protrusions, contained receptor binding, infectivity and
antigenicity controlling residues⁵⁴. (*Section Results Line 249-254, Page 10*)

**Comment 18:** Lines 476-477: this statement is false; suprachoroidal injections are rarely used in clinical settings, whereas
intravitreal injections are very common, and subretinal injections are less common; suprachoroidal delivery is a relatively
novel mode of delivery that is not yet well characterized.

**Response:** Thank you for your careful review. We apologize for the misdescription of the suprachoroidal injection. We have
revised as:

Compared with subretinal and intravitreal injections, suprachoroidal injections is a recent breakthrough in the retinal gene-
delivery landscape and may provide a unique opportunity to perform less invasive surgeries and less likely to cause side
effects such as inflammation³³. (*Section Discussion Line 371-373, Page 13*)

**Comment 19:** Lines 651-653: more details are needed to clarify what the “self-researched suprachoroidal chamber syringe”.
If this is not commercially available, the authors should include a photo and explain its design, as compared to other
suprachoroidal AAV studies.

**Response:** We appreciate the suggestion. As mentioned in **Comment 5** above, we have included a detailed description of the
injection needle.

For the suprachoroidal injection, the device utilized for suprachoroidal injection was a customized combination of an
adapter, a 1 mL syringe, and a needle. The adapter incorporated a unique internal design that enabled the syringe and needle
to lock after insertion, allowing control over the exposed length of the needle. The needle tip length was adjusted by turning
the fitting at the top of the adapter. For suprachoroidal injection in monkeys, a needle tip length of 0.7 mm was used.
Additionally, we have uploaded a video capturing the process of suprachoroidal injection in a monkey, recorded using an
infrared thermal camera (Extended Data Video.1). (*Section Method Line 838-843, Page 25*)

**Comment 20:** Line 695: Heidelberg Spectralis is not a fundus camera, it’s an SLO for near-infrared reflectance.

**Response:** Thank you for the corrections, and we have revised manuscript accordingly.

Image acquisition was performed by Spectralis HRA (Heidelberg), and monkeys were quickly injected with fluorescein
sodium (20 mg/kg) via saphenous veins of lower limb (or other suitable sites). (*Section Method Line 877-879, Page*
**26**)

**Comment 21:** Line 708: More details are needed for the laser procedure. Was a macular contact lens used? Is the Vitra a slit
lamp, rather than a “total retinoscope” which is not an appropriate term.

**Response:** We are thankful for the suggestion. In order to enhance clarity and address the concerns raised by the reviewer,
we have included a concise description as follows:

The laser (Vitra 532 nm) was mounted on a slit lamp using an adapter and the beam was directed onto the retina using a
contact lens (Mainster wildfield laser lens). Laser photocoagulations were performed in the perimacular area of the monkey
eyes. Lesions were placed in the macula with six spots. Laser lesions were placed in a circular fashion around the macula
about one disk diameter from the foveal center. Care was taken to avoid lasering the fovea. Photocoagulation used laser energy
to create controlled burns in specific areas of the retina. The laser parameters used for photocoagulation typically included a
spot diameter of 50 μm , an energy range of 0.5 to 0.7 W, and an exposure time of 0.05 seconds. (*Section Method Line*
**888-893, Page 26**)

**Comment 22:** Minor: ARPE19 was spelled incorrectly throughout manuscript.

**Response:** We sincerely apologize for the misspelling of the word, which was a result of a clerical error. We have rectified
the description "ARPE19" in the full text.

**Comment 23:** Minor: grammatical errors throughout (Lines 376, 383)

**Response:** We apologize for the language issues in the original manuscript and have made the necessary revisions.

To assess the stability of the AAVv128, we conducted experiments comparing the temperature dependence of AAVv128

genome release with that of AAV8 at different pH values. (***Section Results Line 177-179, Page 7***)
Compared to AAV8, AAVv128 exhibited more evident DNA accessibility at pH 6.0. (***Section Results Line 185, Page 7***)

**Comment 24**: Minor: line 442: did the authors mean "superior nuclear fraction"?

**Response:** Thank you for your thorough review. We apologize for the incorrect description of the superiority of AAVv128. We have made the necessary revisions.

Additionally, we found that the relative anti-VEGF protein expression levels of AAVv128 were significantly higher than AAV8 at different time points when transduced with HeLaRC32 and ARPE19 cells. This further supports the higher nuclear fraction ratio of AAVv128 compared to AAV8 in HeLaRC32 cells. (***Section Results Line 228-231, Page 10***)

**Comment 25**: Ext Fig. 1: Why are the CHX10 and PROX1 expression different between v128 and AAV8 groups?

**Response:** Thank you for your careful review. During the immunohistochemical process, we stained each section individually with different antibodies, such as CHX10, PROX1, and RHO, among others. However, due to the potential variability of viral transduction in retinal tissue, it is possible that not every section will be identical after staining.

**Comment 26**: Ext Fig. 2: Some additional eyes had local hemorrhages (e.g. last OS eye of vehicle) that should be excluded also.

**Response:** We appreciate this suggestion. We excluded additional eyes that had local hemorrhages and added the Grade IV spots in Extended Table.6.

Extended Date Fig.10 | A model of laser-induced CNV in NHPs is being used to evaluate the efficacy of nAMD. Fluorescein fundus angiograph (FFA) was used to determine the number of Grade IV lesions. Representative FFA of NHP eyes treated with suprachoroidal injection of Vehicle (a), AAV8-anti-VEGF vector (b) and AAVv128-anti VEGF vector (c) at 35-days and 49-days (2×10^{12} vg/eye, 100 μ L volume, n=8). OD, oculus dextrus; OS, oculus sinister; FFA, fundus fluorescence angiography. [1] The laser spot could not be observed due to bleeding after laser modelling. It was not included in the statistical values.

Extended Date Table.6 | Grade IV CNV lesions % of each assessable eye at day 35 and day 49 (dose: 2×10^{12} vg/eye, 100 μ L volume, n=8)

Time	Vehicle			AAV8			AAVv128		
	No.	spots	Grade IV lesions (%)	No.	spots	Grade IV lesions (%)	No.	spots	Grade IV lesions (%)
Day35	5F001-OD	2	33.33	1F001-OD	6	100.00	4F001-OD	0	0.00
	5F001-OS	3	50.00	1F001-OS	3	50.00	4F001-OS*	N/A	N/A
	5F002-OD	5	83.33	1F002-OD	1	16.67	4F002-OD	0	0.00
	5F002-OS	5	83.33	1F002-OS	3	50.00	4F002-OS	0	0.00
	5M001-OD*	3	60.00	1M001-OD	0	0.00	4M001-OD	0	0.00
	5M001-OS	1	16.67	1M001-OS	0	0.00	4M001-OS	0	0.00
	5M002-OD	2	33.33	1M002-OD	0	0.00	4M002-OD	0	0.00
	5M002-OS*	0	0.00	1M002-OS	2	33.33	4M002-OS	0	0.00
Day49	5F001-OD	6	100.00	1F001-OD	5	83.33	4F001-OD	0	0.00
	5F001-OS	4	66.67	1F001-OS	4	66.67	4F001-OS*	N/A	N/A
	5F002-OD	6	100.00	1F002-OD	2	33.33	4F002-OD	0	0.00
	5F002-OS	5	83.33	1F002-OS	3	50.00	4F002-OS	0	0.00
	5M001-OD	5	83.33	1M001-OD	0	0.00	4M001-OD	0	0.00
	5M001-OS	5	83.33	1M001-OS	1	16.67	4M001-OS	0	0.00
	5M002-OD	3	50.00	1M002-OD	1	16.67	4M002-OD	0	0.00
	5M002-OS	0	0.00	1M002-OS	4	66.67	4M002-OS	0	0.00

Note: Six laser spots were applied for each eye. Grade IV CNV lesions % of each assessable eye was calculated as follow: Grade IV CNV lesions % = the absolute number of grade IV lesions \div the total number of assessable lesions. N/A: The laser spot could not be observed due to bleeding after laser modelling. It was not included in the statistical values.

Reviewer #2 (Remarks to the Author):

In this paper, Luo et al exploit rational design to identify novel adeno-associated viral (AAV) vectors-based capsids, with improved retinal tropism. From an initial in vivo screening, the authors isolate the AAVv128 capsid, which they further characterize both at the structural and functional level. This capsid presents 10 amino acids of difference from the parental AAV8, which the authors propose to critically influence capsid structure and stability and, consequently, interaction with cell surface receptors. To investigate the improved transduction abilities of this novel AAV vectors, the authors explore 3 relevant intraocular delivery routes (i.e. intravitreal, subretinal and suprachoroidal), using as models both mice, rabbits, and non-human primates (NHP). To finally define the therapeutic potential of the novel v128 vector, they explore its use to deliver anti-VEGF in a laser-induced choroidal neovascularization (CNVs) NHP model of age-related macular degeneration.

Data in the CNVs NHP model presented in the manuscript appear promising, showing higher efficiency of the v128 vector, compared to AAV8, in preventing development of neovascular AMD phenotypes when delivered via suprachoroidal injection. However, despite exploitation of different injection routes and animal models, the manuscript lacks in several instances of a comprehensive characterization of the transduction properties of the vector, to support the claim of v128 being a “highly potent variant enabling efficient ocular-directed gene delivery”, to relevant cell types involved in retinal diseases, as detailed below:

Comment 1: Intravitreal injection: v128 is found to significantly outperform AAV8 in mice and rabbit; yet, AAV8 very poorly

transduces the retina from the vitreous, so it is difficult to understand whether the transduction levels achieved with v128 are
therapeutically meaningful. Comparison with AAV serotypes known to result in higher transduction levels from the vitreous
(i.e. AAV7m8 or AAV2) appears required. Also, in Fig. 4a, transduction from vitreous appears to result in mCherry expression
more efficient in the outer retina and RPE than in the INL/GCL; this is in contrast with what shown in mice (Fig. 1e), where
transduction from v128 is exclusively limited to the GCL. Can the author explain this discrepancy?

**Response:** We are extremely grateful to reviewer for pointing out this problem. In our study, we used the mCherry transgene
to perform a comprehensive comparison of three capsids, namely AAV8, AAV2.7m8, and AAVv128, through intravitreal
injection and suprachoroidal injection in New Zealand rabbits (1×10^{11} vg/eye). The results showed that AAVv128 had the
highest transduction efficiency when administered via suprachoroidal injection, while AAV2.7m8 outperformed other AAV
variants when delivered through intravitreal injection (Extended Date Fig.6). It is worth noting that the AAVv128 capsid was
developed based on the AAV8 capsid, using it as a scaffold and introducing specific amino acid substitutions in the variable
region of VP3 VIII (Q588~ A592). Therefore, we focused on the use of AAVv128 compared to AAV8 in the manuscript. We
are grateful for the reviewer's valuable suggestion and we will conduct extensive experiments in the further to investigate the
potential of AAVv128 compared to other AAV capsids (i.e. AAV2, AAV2tYF and 4D-R100) in the future.

We think the different transduction efficiencies between rabbits and mice by intravitreal injection were as follows: (1)
the different doses used in rabbits and mice. Previous studies have indicated that the cellular tropism of AAVrh10 was
significantly different in the retinas of rats receiving 2.5×10^{10} vg/eye and 2.5×10^{11} vg/eye [1]. The lower dose of AAVrh10
only transduced the GCL, while the higher dose of AAVrh10 could target all retinal layers from the GCL to the RPE cell layer.
However, the cellular tropism of AAVrh10 was transferred to the GCL, INL and RPE cell layer in the rabbit retina receiving
2.8×10^{12} vg/eye. In our study, the dose of AAVv128 was 6.8×10^8 vg/eye in mice and 1×10^{11} vg/eye in rabbit; (2) there are
different types of AAV receptors between the eyes of mice and rabbits [2]; (3) variations in the thickness and/or components
of the inner limiting membrane between rodent and rabbit eyes [2]; (4) discrepancy in retinal cell microenvironment between
rodents and rabbits [3]; (5) difference in eye size resulting in less physical damage and/or changes in intraocular pressure (IOP)
after intravitreal injection in rabbits than in rodents [3,4]. Some of these same factors may account for individual differences
within species.

To better illustrate the results, the corresponding descriptions were added in the revised manuscript.

In an attempt to assess translatability of the strong retina tropism of AAVv128 from murine to large animals, we used the
mCherry transgene to perform a comprehensive comparison of three capsids, namely AAV8, AAV2.7m8, and AAVv128,
through intravitreal injection and suprachoroidal injection in New Zealand rabbits (1×10^{11} vg/eye). The results showed that
AAVv128 exhibited the highest transduction efficiency when administered via suprachoroidal injection, while AAV2.7m8
outperformed other AAV variants when delivered through intravitreal injection (Extended Date Fig.6). It is worth noting that
the AAVv128 capsid was developed based on the AAV8 capsid, using it as a scaffold and introducing specific amino acid
substitutions in the variable region of VP3 VIII (Q588~ A592). In subsequent experiments, we focused on the use of AAVv128
compared to AAV8. (*Section Results Line 274-282, Page 10*)

**Extended Date Fig.6 | Intraocular injections in New Zealand rabbits to evaluate the transduction efficacy of AAV8, AAVv128 and**
**AAV2.7m8**

Note: The New Zealand rabbit eyes treated by intravitreal injections or suprachoroidal injections of the AAV8 and AAVv128 confers detectable mCherry expression at days 28 post-injection (1×10^{11} vg/eye, 100 μ L volume). Scale bars= 100 μ m

Reference:

- [1] Zeng Y, Qian H, Wu Z, et al. AAVrh-10 transduces outer retinal cells in rodents and rabbits following intravitreal administration. *Gene Ther.* 2019 Sep;26(9):386-398.
- [2] Dias MS, Araujo VG, Vasconcelos et al. Retina transduction by rAAV2 after intravitreal injection: comparison between mouse and rat. *Gene Ther.* 2019 Dec;26(12):479-490.
- [3] Andrea B Weir, Margaret Collins. Comparative ocular anatomy in commonly used laboratory animals. In *Assessing Ocular Toxicology in Laboratory Animals*, 2013, pp. 1-21, Humana Press, New York.
- [4] Nakaizumi Y. The Ultrastructure of Bruch's Membrane: II. Eyes With a Tapetum. *Arch Ophthalmol.* 1964;72(3):388–394.

Comment 2: Subretinal injection: Fig.2 convincingly shows overall higher levels of retinal transduction from v128 compared to AAV8; however, given the experimental setting used (inclusion of a ubiquitous promoter in the EGFP expression cassette and harvesting of the whole retina for DNA analysis) it is not clear the transduction levels achieved by v128 in relevant cell types for treatment of retinal diseases (RPE and, more importantly, photoreceptors). Also, evaluation of EGFP genome copy number in retinal tissues (panels j) shows that only half (3 out of 6) of the retinas injected with v128 clearly outperform AAV8. Data in Fig. 3 are included to support increased transduction of several different cell types in the retina from v128, including photoreceptors and ganglion cells. However, data appear not consistent enough to support the claims. Photoreceptors nuclei are clearly transduced only in 1 (first row on the left) out of 7 v128-injected eyes shown in Fig. 3. All the other eyes show green signal only at the level of OS, which is unexpected considering that the EGFP reporter protein should localize evenly in photoreceptors. Indeed, even in the presence of low levels of transduction, as in one of the AAV8-injected eyes (see fifth row of the right part of Fig. 3), EGFP positive nuclei in the ONL are clearly detectable. Eyes injected with vectors encoding for EGFP from a photoreceptor-specific promoter are required to definitively assess levels of transduction from v128 in photoreceptors. Similarly, EGFP positive GC are evident only in the last row of v128-injected eyes (Fig. 3, left part). Therefore, the claim that “v128 vector robustly transduce ganglion cells” appears not supported. The author should explain this apparent lack of consistency. Lastly, the authors should include representative pictures of PBS-injected eyes, as negative controls of the immunostainings.

Response: Thanks to the reviewers for pointing out the issue, we identified an inconsistency in the immunofluorescence sections of the manuscript. To obtain a more comprehensive knowledge of the transduced cells after subretinal injection of AAVv128, we conducted the experiment again and captured new images using the Olympus SpinSR10 spinning disc confocal super-resolution microscope. The updated results were presented in Fig.3. AAVv128 showed almost complete ONL layer transduction and a broader range of eGFP transduction levels compared to AAV8. However, AAV8 showed a greater diversity of eGFP fluorescence signals only near the cornea. Other retinal areas had lower eGFP expression, with transduction observed exclusively in photoreceptors and RPE cells. We used different antibodies to identify the transduction cell types of AAVv128, as shown in Fig.3. AAVv128 was able to transduce rod, cone, RPE, and horizontal cells. Nevertheless, Müller cells, astrocytes, ganglion cells, and anaplastic cells did not show transduction in response to AAVv128. On the other hand, AAV8 transduced only a tiny number of RPE cells and photoreceptors at the same dosage.

Furthermore, we have integrated the PBS group in Fig.3. We chose to use the mouse housekeeping gene *tfrC* to standardize the scoring in order to attain improved accuracy because individual mouse differ from one another. With the exception of one animal with an abnormally high level (0.55, 0.60, 0.63, 0.66, 1.28, and 2.44), the average viral genome content for the AAV8-treated group was 0.74 viral copies per mouse genome, as shown in Fig.2j. In the AAVv128-treated group, the average viral genome content was 3.98 copies/mouse genome (0.91, 1.56, 2.01, 5.59, 6.00, and 7.86). We therefore concluded that the AAVv128-treated group had significantly greater viral genome content in their mouse retina than the AAV8-treated group. Furthermore, we completely concur with the reviewer's recommendation to continue using specific promoters such RPE65 instead of the CBA promoter to further investigate the cellular transduction mechanism of AAVv128 in subsequent research.

Fig.3 Cross-section and immunofluorescence analyses of mice retina treated with subretinal injection of AAV8 and AAVv128 capsid.

Note: **a** Coronal sections of mice retina transduced with rAAV-CBA-eGFP. Native eGFP expression (green) shows the positively transduced cell. Scale bar= 200 μ m. **b** IF stained sections (red) with antibodies against Rhodopsin (photoreceptors, Rod), Peanut agglutinin (PNA, photoreceptors, Cone), RPE65 (RPE cells), GS (Müller cells), GFAP (astrocytes), RBPMS (ganglion cells), PROX1 (anaplastic cells and ganglion cells) and Calbindin D28K (horizontal cells, anaplastic cells and ganglion cells) indicate the distribution of cell types across the retina. Native eGFP expression (green) that colocalize with IF staining (yellow or white) reveals the positively transduced cell type indicated. Scale bars= 20 μ m (right column) or 200 μ m (left column). INL: inner nuclear layer, ONL: outer nuclear layer

Comment 3: Suprachoroidal injection: as for subretinal injections, more accurate dissection of which cell layers are efficiently transduced from v128 is required. In Extended Data Fig. 1 no clear transduction is seen in both the ONL and IS. In Fig. 4c (and a), the upper panels with low magnification images of the retina are barely visible and, as such, not enough informative on the extent of retinal transduction (including ONL) achieved. The authors comment “v128 predominantly transduced the photoreceptor inner and outer segments (...) after both intravitreal and suprachoroidal injections”, however it is unclear why expression of reporter proteins (as EGFP and mCherry) should be limited to IS/OS in photoreceptors, and not visible in the whole photoreceptor body. Representative pictures of PBS-injected eyes should be included here too. Also, it would be highly informative to add Scanning laser ophthalmoscopy (SLO) images, as in Fig. 4f, for all the NHP eyes injected with EGFP. Lastly, as a note, reading of the manuscript has been complicated by the not always accurate use of the English language, as well as several typos (as an examples, ARPE19 cells are misspelled as either APRE19 or AREP19 cell throughout the text) and non-sequential order of citation of the figure panels (see Fig.6b-c, cited in line 324, before Fig. 6a, cited in line 330; or Fig. 8f, g, cited in lines 432, before Fig.8d, e, cited in line 435). Also, in this reviewer opinion, the manuscript would be easier to be followed if the characterization of the novel AAV vector (fig.6, 7 and 8) is presented before investigation of its

retinal transduction properties and therapeutic efficacy.

**Response:** Thanks to the reviewer's suggestions, we have corrected a mistake in our description of the transduced cell type
after suprachoroidal injection in monkeys. As shown in Extended Data Fig.8, the images revealed that the transduction of
AAVv128 was significantly higher, exhibiting stronger eGFP fluorescence compared to AAV8. Regarding AAVv128
transduction, it was predominantly observed in photoreceptors and RPE cells. Conversely, AAVv128 showed limited
transduction in astrocytes, horizontal cells, bipolar cells, Müller cells and ganglion cells (Extended Data Fig.8). In contrast,
AAV8 mainly targeted photoreceptors and RPE cells, albeit with lower efficiencies compared to AAVv128. Under a 60x field
of view, the homogeneous distribution of eGFP in cone and rod cells was shown in Fig.3. In subsequent research, we will use
a specific promoter to investigate the transduction of cell types after suprachoroidal injection of AAVv128 in monkeys.

We apologized for only having two monkeys for the immunofluorescence test, which were injected with AAV8 and
AAVv128, respectively. We were unable to provide images of the PBS group injections because we did not perform the
injections.

Meanwhile, we have included Scanning laser ophthalmoscopy (SLO) images of the remaining monkey eyes in Extended
Data Fig.9. We only captured one photo of the OS eye in the AAV8 group due to the weak fluorescence observed. We are
thankful to the reviewer for the valuable suggestions on language standardization, which we have incorporated into the full
text. In addition, we have optimized the organization of the original text by repositioning the description of experimental data
pertaining to the novel AAV vector (Fig.6, Fig.7, and Fig.8) ahead of the discussion on the retinal transduction properties and
therapeutic efficacy of large animals.

To better illustrate the results, the corresponding descriptions were added in the revised manuscript.

The images revealed that the transduction of AAVv128 was much more efficient, showing stronger eGFP fluorescence
compared to AAV8. In the case of AAVv128, the transduction was predominantly observed in photoreceptors and RPE cells.
On the other hand, AAVv128 showed limited transduction in astrocytes, horizontal cells, bipolar cells, Müller cells and
ganglion cells (Extended Data Fig.8). In contrast, AAV8 primarily targeted photoreceptors and RPE cells, but with lower
efficiencies compared to AAVv128. (*Section Results Line 310-315, Page 12*)

Additionally, it was reported that suprachoroidal delivery of AAV in monkey might have a greater tendency to distribute
circumferentially along the periphery rather than posteriorly toward the macula^{70,71}. Suprachoroidal AAV delivery might be
better suited for gene therapies that target peripheral or RPE diseases, for example the production of anti-angiogenesis agents
for neovascular retinal conditions, rather than macular diseases³³. The transduction cell types of AAVv128 in these eyes were
mainly photoreceptors and RPE cells (Extended Data Fig.8). It was shown that suprachoroidal delivery of AAV8 in Sprague-
Dawley rats mainly resulted a larger average transduction diameter and an abundance of transduced cells within the outer
nuclear layer (ONL)⁷². This could be due to the structural differences between species that result in different transduction
cell types, including Bruch's membrane (BrM) thickness, the thickness or number of cells within the retinal layers^{73,74}.
(*Section discussion Line 388-397, Page 14*)

Extended Data Fig.9 | Transduction validation of NHPs eyes treated with AAV8 and AAVv128 by Scanning laser ophthalmoscopy (SLO).

Reviewer #3 (Remarks to the Author):

The manuscript by Luo et al describes the rational design, the production, and the characterization of an AAV variant AAVv128. This variant is rationally designed based on the sequence and structure of AAV8. When compared to AAV8, AAVv128 exhibits significantly greater transduction of all ocular cells tested, independent of the mode of application, or the size of the test model ranging from mice to non-human primates. To provide an explanation for the significantly higher transduction phenotype of AAVv128 compared to AAV8, the authors present an extensive comparative biophysical characterization of the AAVv128 and AAV8 by cryo-EM, DSF, and Capillary Isoelectric focusing (cIEF). This manuscript provides valuable information for the AAV gene therapy community, specifically for the treatment of ocular genetic disorders and neovascular age related macular degeneration. The manuscript provides substantial experimental evidence to support the conclusion that AAVv128 is superior to AAV8 in targeting and transducing ocular tissue. There are however several issues that needs to be addressed:

Comment 1: There are numerous publications citing the use of different AAV2 based vectors for the treatment of different ocular disorders. One of these AAV2 based vectors (Upstaza) have been approved for the treatment of RPE, however all the comparisons have focused on the use of AAVv128 compared to AAV8. Although AAVv128 is superior to AAV8 there is no discussion of its performance compared to other established serotypes used in the treatment of ocular disorders. In the discussion, there is an extensive description of other AAV serotypes and variants that are optimized for treatment in the eye, and while this is important to the discussion it needs to be mentioned in the introduction.

Response: We are extremely grateful to reviewer for pointing out this problem. We have added an extensive description of other AAV serotypes and variants for the treatment of different ocular disorders in introduction and discussion. The description as follow:

In addition to the naturally occurring AAVs, the engineered AAV variants were also investigated in many studies.

Multiple engineered AAV variants have gone to therapeutic applications. Among the ongoing clinical trials for retinal diseases,
34% used engineered AAV serotypes¹³. These include AAV2tYF (generated by rational design, clinical trials include
NCT02599922, NCT02935517, NCT03316560 and NCT02416622), AAV2.7m8 (generated by directed evolution, clinical
trials include NCT04645212, NCT03748784, NCT04418427 and NCT03326336) and 4D-R100 (generated by directed
evolution, clinical trials include NCT04483440 and NCT04517149). Encouragingly, more and more engineered AAV variants
are explored in clinical applications. (*Section Introduction Line 48-55, Page 3*)

In an attempt to assess translatability of the strong retina tropism of AAVv128 from murine to large animals, we used the
mCherry transgene to perform a comprehensive comparison of three capsids, namely AAV8, AAV2.7m8, and AAVv128,
through intravitreal injection and suprachoroidal injection in New Zealand rabbits (1×10^{11} vg/eye). The results showed that
AAVv128 exhibited the highest transduction efficiency when administered via suprachoroidal injection, while AAV2.7m8
outperformed other AAV variants when delivered through intravitreal injection (Extended Date Fig.6). It is worth noting that
the AAVv128 capsid was developed based on the AAV8 capsid, using it as a scaffold and introducing specific amino acid
substitutions in the variable region of VP3 VIII (Q588~A592). In subsequent experiments, we focused on the use of AAVv128
compared to AAV8. (*Section Results Line 274-282, Page 10*)

Rational design approaches, based on knowledge about potential receptor interactions, have been employed to create
capsid variants with desired features and to reduce the size of the AAV capsid variants library⁵⁹. In addition to the naturally
occurring AAVs, multiple engineered AAV variants have gone to clinical trials for retinal diseases, including AAV2tYF,
AAV2.7m8 and 4D-R100¹³. (*Section Discussion Line 355-358, Page 13*)

**Extended Date Fig.6 | Intraocular injections in New Zealand rabbits to evaluate the transduction efficacy of AAV8, AAVv128 and**
**AAV2.7m8**

Note: The New Zealand rabbit eyes treated by intravitreal injections or suprachoroidal injections of the AAV8 and AAVv128 confers detectable
mCherry expression at days 28 post-injection (1×10^{11} vg/eye, 100 μL volume). Scale bars= 100 μm

**Comment 2:** Section 3 .5 The description of the icosahedral 2-fold, 3-fold and 5-fold axis should be labeled on figure 6A.
The secondary structural elements of the model should also be described or referenced. The variable regions should be
introduced and rmsd values shown for the superposition of the monomers in 6C. The colors selected for the AAV8 and
AAVv128 models are too close and need to be more distinct. The stick models in 6E and 6F should have all the residues
labeled and the broken lines representing the interactions removed. The interactions can be shown either in a table or as a
table in the figure. It is also not clear from the figure which residues are the insertion and the 'pocket loop'.

**Response:** With utmost gratitude, we wholeheartedly acknowledge the invaluable suggestion provided, which has led us to
refine the description of the structure-related data. As elucidated in the illustrious Fig.6, we have modified the aforementioned
description, ensuring a comprehensive and nuanced portrayal of the intricate details. We have labeled the residues that the
insertion and the 'pocket loop' in a resplendent shade of red, as depicted in the illustrious Fig.6e. We have also unraveled the
intricate tapestry of hydrogen bonding interactions within the VR-III loop in the Extended data Table.8.

To better illustrate the results, the corresponding descriptions were added in the revised manuscript.
The AAVv128 and AAV8 structures were similar, except for the 3-fold protrusions (Root-mean-square deviation (RMSD) of

0.507 Å after superimposition of 516 Cα atoms; Fig.6b-c). The 3-fold protrusions were more pronounced in AAVv128 due to the 5-amino acid insertion, which protruded radially outward from the capsid surface, forming a larger loop that had greater spatial flexibility (Fig.6b-c). The 3-fold region, including the protrusions, contained receptor binding, infectivity and antigenicity controlling residues⁵⁴. (Section Results Line 249-254, Page 10)

Fig.6 Cryo-EM reveals differences between AAVv128 and AAV8

Note: **a** The rendered images of the empty AAVv128 was shown and labeled with the particle dimensions. Depth cueing with colors was used to indicate radius (< 100 Å, blue; 100–120 Å, from cyan to green; > 140 Å, red). **b** Comparison of the capsid surfaces of AAV8 and AAVv128. Gray surface representation of the AAV8 (PDB, 6V12) and AAVv128 (PDB, 8JRE) generated from 60 VP monomers. The amino acid differences between with AAV8 and AAVv128 on the capsid surface were colored red. **c** The Structural superposition of AAV8 and AAVv128 VP shown as ribbon diagrams. The conserved β-barrel core motif and the αA helix were indicated, the positions of VR-I to VR-IX were labeled. **d** The residues in a featured fragment from AAVv128 VP3 VIII variable region (D584-N604) were shown as colored sticks and cartoon surrounded by density. **e,f** The residues of main difference between AAV8 (**f**) and AAVv128 (**e**) lied in the VP3 VIII variable region were shown as colored sticks. The residues that the insertion and the 'pocket loop' were labeled in a resplendent shade of red. **g** The identical or conserved residues were shown as letters with red or yellow backgrounds, while the non-conserved residues have a white background. The variable regions between different AAV serotypes were shown in cyan

Extended Date Table.8 | The hydrogen bonding interactions within VR-III loop.

AAVv128	
L586	R596
R587	R596
R588	T594, R596
G589	N593, A595
N590	A595
AAV8	
Q587	T591
Q588	T591

**Comment 3:** The statement in Section 3.6 line 374 - 376 is not grammatically correct and needs a reference. The same
statement is repeated in 387 – 390 which also needs a reference and proper explanation as to why these physical properties
are important for commercial applications.

**Response:** Thank you for your meticulous and thorough review. We greatly appreciate your valuable feedback, and we have
diligently incorporated the suggested revisions into the manuscript.

A good capsid variant possesses immense commercial value and holds promising prospects for application^{13, 51}. Its
significance extends beyond its high transduction efficiency, encompassing attributes such as exceptional stability, ease of
purification, and remarkable packaging efficiency. (*Section Results Line 175-177, Page 7*)

The stability of a capsid variant is crucial for its commercial value and application prospects⁵³. When moving into human
use under the auspices of an FDA Investigational New Drug (IND) application, it is necessary to demonstrate the stability of
AAV under various conditions of storage, dilution, and administration when used in humans⁵³. (*Section Results Line*
*189-152, Page 8*)

Minor issues

**Comment 4:** Several grammatical errors, and the sentences need further clarification. A few examples are listed below:

(i) Line 68, ...suspected unexpected serious adverse reaction

(ii) Line 113 – 114, Four weeks after the injection, the expression of eGFP transgene mRNAs in the mouse eyes received
different vectors were quantified by qRT-PCR.

(iii) Line 118, ... with a level 75-folds higher than that of AAV8

(iv) Line 376 – 377, To assess the stability of the v128, we conducted experiments compared the temperature 377 dependence
of v128 genome release with that of AAV8 at different pHs levels.

(v) Line 605, The study involved the use of Six- to eight-week-old male

**Response:** We are immensely grateful to the esteemed reviewer for your astute observation and invaluable contribution in
identifying this critical issue. With utmost diligence, we have meticulously addressed the grammatical errors in the sentences
and made the necessary revisions in the manuscript.

(i) In the Phase 2 INFINITY trial, one subject receiving a high dose of ADVM-022 encountered a truly unexpected and serious
adverse reaction of hypotony with panuveitis and loss of vision following a single intravitreal injection. (*Section*
*Introduction Line 74-76, Page 4*)

(ii) Four weeks after injection, a careful examination was conducted to assess the expression of eGFP transgene mRNAs in
the eyes of mice exposed to different vectors (6×10^8 vg/eye, n=3). This assessment was carried out using the quantitative
reverse transcription polymerase chain reaction (qRT-PCR). (*Section Results Line 120-123, Page 5*)

(iii) Notably, AAVv128 demonstrated the highest mRNA expression, with a level that was 75-fold higher than that of AAV8.
(*Section Results Line 125-126, Page 6*)

(iv) To assess the stability of the v128, we conducted experiments comparing the temperature dependence of v128 genome
release with that of AAV8 at different pH values. (*Section Results Line 177-179, Page 7*)

(v) Six- to eight-week-old male C57BL/6J mice, New Zealand White rabbits with weight ranging from 2.0 to 2.5 kg (Dashuo
Laboratory Animal Technology Co., Ltd., China), and Cynomolgus monkeys with weight ranging from 2.5 to 5.0 kg (West
China-Frontier PharmaTech Co., Ltd., Tianfu Drug Research Center, SYXK(Chuan)2021-238) were used in this study.
(*Section Method Line 799-801, Page 25*)

**Comment 5:** The term AAVv128 and v128 is used interchangeable, please select one term and use it consistently.

**Response:** Thanks to the reviewer's comments, we have replaced all v128 with AAVv128.

**Comment 6:** It is not clear why the Capillary isoelectric focusing (cIEF) was done. The high resolution Cryo-EM structure
show the location of the positive residues in VRIII. It should also be noted that the 588-RGNQQ-592 motif in AAVv128 is
most similar to 585-RGNRQ-589 motif in AAV2. The AAV2 residues R585 and R588 have been shown to be critical for the
heparin binding phenotype of AAV2. Consequently, a heparin binding assay would potentially provide more informative than

a cIEF assay.

**Response:** We are really grateful for this suggestion. We included the positively charged amino acids R588 and R596 in our
rational design. Capillary isoelectric focusing (cIEF) was used for a comparison investigation in order to evaluate the effect
of these alterations on the charged property of AAV8. Fig.4 showed that the pI of AAVv128 and AAV8 was 7.693 and 6.859,
respectively. Additionally, during the ion-exchange chromatography purification procedure, we noticed notable alterations in
the elution peaks for AAVv128. These changes were ascribed to AAVv128's higher positive charge in comparison to AAV8.

To further investigate the binding capabilities of AAV2, AAV2.7m8, AAV8, and AAVv128, we conducted a heparin
binding assay. The results indicated that among these variants, AAV2.7m8 exhibited the strongest ability to bind HSPG,
followed by AAV2 and AAVv128. Conversely, AAV8 demonstrated the weakest binding affinity. Consequently, AAVv128
incorporated two crucial amino acids, R588 and R596, in the VP3 VIII region. Our findings indicated that these amino acids
might improve the cellular receptor HSPG's recognition (Extended Fig.11, Extended Table.8) and raise the capsid charge
(Fig.4b). Furthermore, these amino acids significantly improved cell binding ability (Fig.5d, Fig.5f) and facilitated post-entry
cell escape (Fig.5e, Fig.5g-i), as seen in Fig.5.

**Extended Date Fig.11** | A heparin binding assay for AAV2, AAV2.7m8, AAV8 and AAVv128

**Extended Date Table.8** | The hydrogen bonding interactions within VR-III loop.

AAVv128	
L586	R596
R587	R596
R588	T594, R596
G589	N593, A595
N590	A595
AAV8	
Q587	T591
Q588	T591

**Reviewer #4 (Remarks to the Author):**

The paper describes the intraocular transduction characteristics of the AAV8-derived capsid variant v128 which outperforms
the parental vector upon intravitreal, subretinal and suprachoroidal administrations. v128 does not cross the retinal barriers
following intravitreal administration back to the outer retina, however, in general, cells that are AAV8 target upon intraocular
injections are transduced at higher levels by v128 across species up to non-human primates. This results in better therapeutic
outcome of v128 than AAV8 in a primate model of choroidal neovascularization upon suprachoroidal administration. The
cryo EM analysis of v128 highlights differences with AAV8 which could explain its higher transduction characteristics.

The experiments are overall well designed with AAV8 as constant comparison for v128. The evidence of v128 higher potency
than AAV8 is supported by an extensive and comprehensive set of data both in vitro and in the retina, leaving little doubt that
v128 should be preferred in the retina to AAV8. It would be interesting to test if this holds true also in liver, another traditional
target of AAV8. This could importantly expand v128 therapeutic applications. But of course this goes beyond the scope of the
current manuscript.

**Comment 1:** My only comment is that the authors should explain in detail the rationale for the selection of the regions Q588~
A592 or N263~T274 at AAV8 VP3 VIII or VP3 VI variable regions for the mutagenesis. The current explanation is quite
vague and provides references other papers but since the whole work starts from there it would be nice to expand this part.

**Response:** We appreciate the reviewer's suggestion, we have included more information about the changes made to the VP3
VIII region. Our design approach took into account the importance of R585 and R588, which are key amino acids for the

AAV2 cellular receptor heparan sulfate proteoglycan (HSPG)^[1]. These amino acids play a role in interacting with the laminin
receptor (LamR)^[2]. Additionally, we consider the role of positively charged amino acids in promoting cell binding and
improving immune evasion after cell entry^[3-5]. These factors are carefully considered in our rational design strategy.

To better illustrate the results, the corresponding descriptions were added in the revised manuscript.

Briefly, we used the AAV8 capsid as a scaffold and replaced its amino acid sequences between Q588~ A592 or
N263~T274 at the VP3 VIII or VP3 VI variable region. To construct these capsid variants, we designed a set of 18 peptides,
ranging from 5 to 14 amino acids in length, based on the known structural biology and cellular receptorology of AAV8³⁵⁻⁴²
(Extended Data Table.1). Our design approach considered the significance of R585 and R588 as crucial amino acids for the
AAV2 cellular receptor heparan sulfate proteoglycan (HSPG)⁴³. These amino acids also played a role in interacting with the
laminin receptor (LamR)⁴⁴. Furthermore, we acknowledged the importance of positively charged amino acids in enhancing
cell binding and facilitating immune evasion post-cell entry⁴⁵⁻⁴⁷. These rationally engineered capsid variants were generated
by separately packaging a vector genome containing an enhanced green fluorescent protein (eGFP) expression cassette
flanked with AAV2 inverted terminal repeats (ITRs) by using HEK293 cell-triple transfection method (Fig.1a). (**Section**
**Results Line 106-116, Page 5**)

864 865 866 Reference:

- [1] Crosson SM, Bennett A, Fajardo D, et al. Effects of Altering HSPG Binding and Capsid Hydrophilicity on Retinal Transduction by AAV. *J*
*Virol.* 2021 Apr 26;95(10):e02440-20.
- [2] Akache B, Grimm D, Pandey K, et al. The 37/67-kilodalton laminin receptor is a receptor for adeno-associated virus serotypes 8, 2, 3, and
9. *J Virol.* 2006 Oct;80(19):9831-6.
- [3] Patel SG, Sayers EJ, He L, et al. Cell-penetrating peptide sequence and modification dependent uptake and subcellular distribution of green
fluorescent protein in different cell lines. *Sci Rep.* 2019 Apr 18;9(1):6298.
- [4] Lussi C, de Martin E, Schweizer M. Positively Charged Amino Acids in the Pestiviral Erns Control Cell Entry, Endoribonuclease Activity
and Innate Immune Evasion. *Viruses.* 2021 Aug 10;13(8):1581.
- [5] Saadat M, Zahednezhad F, Zakeri-Milani P, et al. Drug Targeting Strategies Based on Charge Dependent Uptake of Nanoparticles into Cancer
Cells. *J Pharm Pharm Sci.* 2019;22(1):191-220.

REVIEWERS' COMMENTS

Reviewer #1 (Remarks to the Author):

The authors have largely addressed most of the prior comments in detail. Given that the suprachoroidal needle is custom and not commercially available, I would still recommend that the authors clearly note in the methods that the device is a custom-made needle with proprietary design, and if possible disclose the manufacturer so that other investigators may reach out to the vendor if necessary.

Reviewer #2 (Remarks to the Author):

The authors have adequately revised the manuscript and the new data convincingly support the conclusions.

As a minor comment, in Fig. 7C, in the higher magnification image from AAVv128-transduced eyes, the authors use dashed line to indicate IS/OS region, however, it appears also the RPE has been included in the marked region. Please double check.

Reviewer #3 (Remarks to the Author):

The authors have addressed all my concerns and made all the necessary modifications to the manuscript.

Responses to Reviewers

Reviewer #1 (Remarks to the Author):

Comment 1: The authors have largely addressed most of the prior comments in detail. Given that the suprachoroidal needle is custom and not commercially available, I would still recommend that the authors clearly note in the methods that the device is a custom-made needle with proprietary design, and if possible disclose the manufacturer so that other investigators may reach out to the vendor if necessary.

Response: We appreciate the reviewer's suggestion, we have included more information about the manufacturer in the revised manuscript.

For the suprachoroidal injection, the device utilized for suprachoroidal injection was a customized combination of an adapter, a 1 mL syringe, and a needle. The adapter incorporated a unique internal design that enabled the syringe and needle to lock after insertion, allowing control over the exposed length of the needle. The length of the needle tip was adjusted by turning the fitting at the top of the adapter. The device was manufactured by Chengdu Origen Biotechnology and currently not commercially available. (*Section Methods Line 357-360, Page 11*)

Reviewer #2 (Remarks to the Author):

Comment 1: The authors have adequately revised the manuscript and the new data convincingly support the conclusions. As a minor comment, in Fig. 7C, in the higher magnification image from AAVv128-transduced eyes, the authors use dashed line to indicate IS/OS region, however, it appears also the RPE has been included in the marked region. Please double check.

Response: We are extremely grateful to reviewer for pointing out this problem. We have corrected the marked region in the Fig.7c.

Fig.7 Intraocular injections in large animals to evaluate the transduction efficacy of AAV8 and AAVv128

Note: The New Zealand rabbit eyes treated by intravitreal injections or suprachoroidal injections of the AAV8 and AAVv128 confers detectable mCherry

expression at days 28 post-injection (1×10^{11} vg/eye, 100 μ L volume, n=4). **a,c** Immunofluorescence analysis of the New Zealand rabbit retinas after intravitreal
injection (**a**) and suprachoroidal injection (**c**). **b,d** The retina-choroid tissues of the New Zealand rabbit retinas after intravitreal injection (**b**) and suprachoroidal
injection (**d**) were isolated and their viral genomic DNA content were measured by qPCR (n=4). NHPs eyes treated by suprachoroidal injection of the AAV8 and
AAVv128 capsid confers detectable eGFP expression at days 14 post-injection (3.5×10^{12} vg/eye, 100 μ L volume, n=1). Scale bars: 50 μ m (down column) or 2
30 mm (up column). **e** Infrared imaging analysis of suprachoroidal injections in cynomolgus monkeys. **f** Transduction validation of NHPs eyes treated with AAV8
and AAVv128 by Scanning laser ophthalmoscopy (SLO); scale bars: 200 μ m. GCL: ganglion cell layer, INL: inner nuclear layer, ONL: outer nuclear layer. Values
represent mean \pm SD. *P* Values were determined by a two-tailed Student t Test. #*P* < 0.05, ##*P* < 0.01, ###*P* < 0.001, ####*P* < 0.0001. Source data are provided
as a Source Data file.